# $C^2$-methyladenosine in tRNA promotes protein translation by facilitating the decoding of tandem m²A-tRNA-dependent codons

Hong-Chao Duan [1,3], Chi Zhang[1,3], Peizhe Song[1], Junbo Yang[1], Ye Wang[1] & Guifang Jia [1,2] ✉

RNA modification $C^2$-methyladenosine (m²A) exists in both rRNA and tRNA of *Escherichia coli* (*E. coli*), installed by the methyltransferase RlmN using a radical-*S*-adenosylmethionine (SAM) mechanism. However, the precise function of m²A in tRNA and its ubiquity in plants have remained unclear. Here we discover the presence of m²A in chloroplast rRNA and tRNA, as well as cytosolic tRNA, in multiple plant species. We identify six m²A-modified chloroplast tRNAs and two m²A-modified cytosolic tRNAs across different plants. Furthermore, we characterize three Arabidopsis m²A methyltransferases—RLMNL1, RLMNL2, and RLMNL3—which methylate chloroplast rRNA, chloroplast tRNA, and cytosolic tRNA, respectively. Our findings demonstrate that m²A37 promotes a relaxed conformation of tRNA, enhancing translation efficiency in chloroplast and cytosol by facilitating decoding of tandem m²A-tRNA-dependent codons. This study provides insights into the molecular function and biological significance of m²A, uncovering a layer of translation regulation in plants.

RNA modifications play a crucial role in the structure, function, and fate of RNA. More than a hundred types of post-transcriptional modifications have been identified in various RNA molecules, with a majority found in tRNA[1]. Particularly, tRNA modifications within the anticodon loop often serve structural functions by influencing nucleoside conformation equilibrium, preventing intra-loop base pairing, and restricting loop dynamics[2]. These modifications contribute to codon recognition[3], stabilization of codon-anticodon interaction[4,5], and prevention of frameshifting[6,7]. Ultimately, tRNA modifications regulate the fidelity and efficiency of protein translation[8,9]. Additionally, mRNA modifications have the ability to influence pre-mRNA processing[10–12], mRNA localization and transport[13], translation initiation[14], and mRNA degradation[15].

Similarly, rRNA modifications are involved in optimizing protein translation speed and accuracy[16]. Despite their prevalence, the specific molecular functions and biological significance of most RNA modifications remain largely unknown.

$C^2$-methyladenosine (m²A) was initially found in *Escherichia coli* (*E. coli*) at position 37 (m²A37) of specific tRNAs (tRNA$^{Arg}_{ACG}$, tRNA$^{Asp}_{GUC}$, tRNA$^{Gln}_{UUG}$, tRNA$^{Gln}_{CUG}$, and tRNA$^{Glu}_{UUC}$, and tRNA$^{His}_{GUG}$)[17,18], as well as at position 2503 (m²A2503) of 23S rRNA[19]. In plants, m²A37 has been identified in certain tRNAs (*Scenedesmus obliquus* chloroplast elongator tRNA$^{Met}_{CAU}$[20], *Triticum aestivum* cytosolic tRNA$^{Arg}_{ACG}$[21], and *Nicotiana rustica* cytosolic tRNA$^{Gln}_{UUG}$[22]). While m²A is typically considered a bacteria-specific RNA modification, its prevalence in plants has not been

[1]Synthetic and Functional Biomolecules Center, Key Laboratory of Bioorganic Chemistry and Molecular Engineering of Ministry of Education, Beijing National Laboratory for Molecular Sciences, Key Laboratory of Bioorganic Chemistry and Molecular Engineering of Ministry of Education, College of Chemistry and Molecular Engineering, Peking University, Beijing 100871, China. [2]Peking-Tsinghua Center for Life Sciences, Beijing 100871, China. [3]These authors contributed equally: Hong-Chao Duan, Chi Zhang. ✉e-mail: guifangjia@pku.edu.cn

thoroughly investigated. In *E. coli*, the dual-specificity methyltransferase RlmN is responsible for m²A installation in both rRNA[23] and tRNA[24], utilizing a radical-SAM mechanism due to the inert nature of adenosine's *C2* position[25]. Extensive studies have elucidated the catalytic features of RlmN in the last decade[26-28]. Another radical-SAM methyltransferase, Cfr, modifies A2503 of 23S rRNA as *C8*-methyladenosine (m⁸A) via a similar mechanism to RlmN[29-31]. m²A2503 in 23S rRNA enhances stacking between A2059 and A2503 in the *syn* conformation, thereby maintaining the structure of the peptide exit tunnel wall[32,33]. Loss of m²A from 23S rRNA increases translation readthrough in *Δrlmn E. coli* strain[24], suggesting a role of m²A in enhancing translational accuracy. However, the precise functions of m²A in tRNA are still largely unclear.

Here, we found that m²A modification is widespread throughout various phyla of the Plantae kingdom, present in chloroplast rRNA, chloroplast tRNA, and cytosolic tRNA of Arabidopsis and other representative plant species. Through precise mapping and quantification, we identified six chloroplast tRNAs and two cytosolic tRNAs that undergo m²A modification at specific sites and frequencies. Furthermore, we characterized three RlmN-like proteins, namely RLMNL1, RLMNL2, and RLMNL3, as the methyltransferases responsible for m²A modification in chloroplast rRNA, chloroplast tRNA, and cytosolic tRNA, respectively. Function investigation revealed that m²A37 plays a crucial role in maintaining the relaxed conformation of tRNA and reducing its melting temperature. Interestingly, m²A in tRNA, but not rRNA, promotes protein translation by facilitating the decoding of corresponding codons. Ribosome footprinting sequencing results provided further evidence that tRNA m²A37 modifications enhance translation efficiency by decoding tandem m²A-tRNA-dependent codons. Overall, our findings unveil the widespread presence of m²A in plants and elucidate the role of tRNA m²A37 in facilitating protein translation.

## Results

### m²A exists in various plant species

We initially examined the presence of m²A (Fig. 1a) in Arabidopsis. Total RNA and purified fractions of different RNA types (rRNA, tRNA, and poly(A) RNA) were enzymatically digested into nucleosides and subjected to LC-MS/MS analysis. To distinguish m²A from other methylated adenosine forms such as *N¹*-methyladenosine (m¹A), *N⁶*-methyladenosine (m⁶A), and *C⁸*-methyladenosine (m⁸A), we compared the retention time of ion chromatography with commercially available standards of methylated adenosine nucleosides. This enabled us to differentiate and identify m²A in the samples, (Fig. 1b, c and Supplementary Fig. 1a). The results showed that m²A is indeed present in chloroplast rRNA, chloroplast tRNA, and cytosolic tRNA, albeit with varying modification ratios. However, we did not detect the presence of m²A in cytosolic rRNA or poly(A) RNA (Fig. 1c, d). Importantly, we confirmed that the m²A nucleoside was not detectable in the enzymes used during the digestion process (Supplementary Fig. 2), providing strong evidence that the m²A identified in Arabidopsis is of endogenous origin. Conversely, we did not observe the presence of m⁸A in examined Arabidopsis RNA samples (Supplementary Fig. 1b).

Subsequently, we selected a range of representative species to investigate the prevalence of m²A across different plants. Various types of RNAs extracted from *Spinacia oleracea* (dicots plant), *Oryza sativa* (monocot plant), *Physcomitrella patens* (moss), and *Chlamydomonas reinhardtii* (green algae) were enzymatically digested and subjected to LC-MS/MS analysis. The results showed all these plant species have m²A in chloroplast rRNA and total tRNA with different modification fractions, but not in poly(A) RNA and cytosolic rRNA (Supplementary Fig. 3). Furthermore, we examined RNA extracted from human HeLa cells and found the absence of m²A in both total RNA and the isolated RNA types

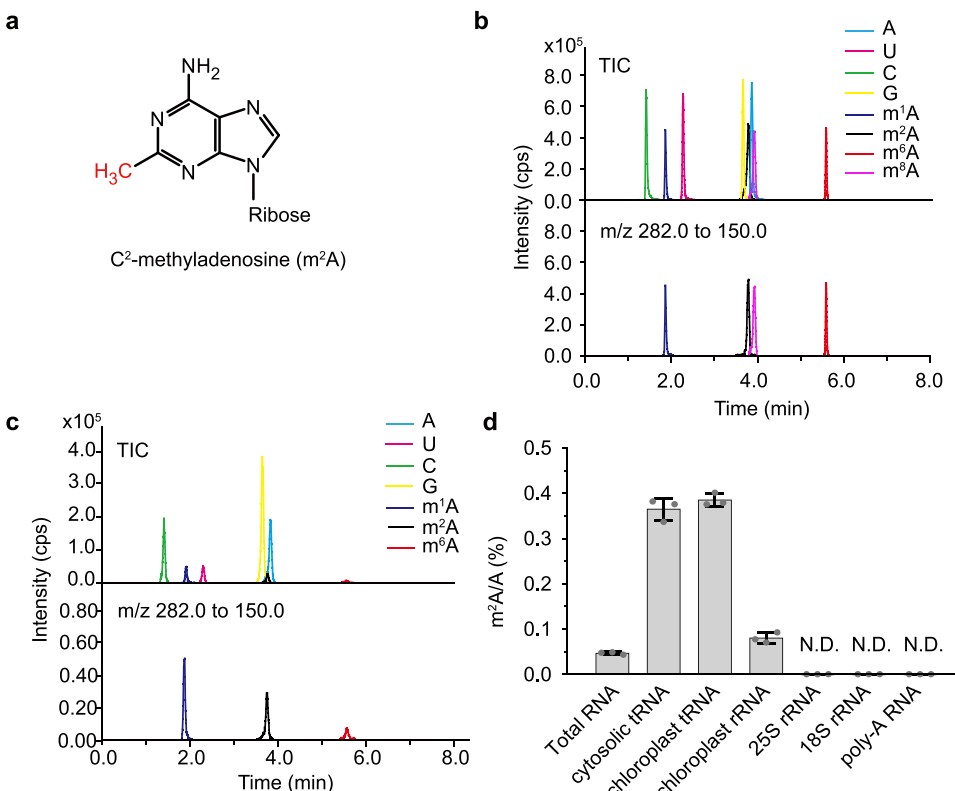

**Fig. 1 | RNA modification m²A is present in Arabidopsis. a** The chemical structure of m²A. **b** Representative LC–MS/MS chromatograms of the indicated commercial nucleoside standards. TIC, total ion chromatogram. m/z, mass to charge ratio. **c** LC–MS/MS chromatograms of the indicated nucleosides digested from Arabidopsis total tRNA. **d** Quantification of m²A levels in different RNA types of Arabidopsis. m²A was not detectable (indicated as N.D.) in Arabidopsis 25S rRNA, 18S rRNA, and poly(A) RNA. Data in (**d**) are represented as means ± SD (*n* = 3 biological replicates).

(Supplementary Fig. 3). These results strongly indicated that m²A may represent a widespread RNA modification across plant species.

## Eight m²A-modified tRNA species are identified in plants

To determine the specific tRNA species harboring m²A modifications in Arabidopsis, we focused on tRNAs with an adenosine 3'-adjacent to the anticodon (position 37), based on the known m²A location at position 37 of tRNA in *E. coli*. A comprehensive analysis revealed a total of 57 cytosolic or chloroplast tRNAs meeting this criterion Supplementary Fig. 4). To isolate individual tRNA species, we designed specific biotinylated antisense probes for each of the identified tRNAs (Supplementary Table 1). The isolated RNAs were subjected to LC-MS/MS analysis and the results showed that two cytosolic tRNAs (tRNA$^{Arg}_{ACG}$ and tRNA$^{Gln}_{UUG}$) and four chloroplast tRNAs (elongator tRNA$^{Met}_{CAU}$, tRNA$^{Arg}_{ACG}$, tRNA$^{Ser}_{GGA}$, and tRNA$^{His}_{GUG}$) contain m²A in Arabidopsis (Table 1 and Supplementary Fig. 4). Comparing these findings with m²A-modified tRNAs in *E. coli*, we noticed the absence of m²A in tRNA$^{Asp}_{GUC}$ and tRNA$^{Glu}_{UUC}$ from both cytosol and chloroplast of Arabidopsis. Additionally, as previous report in tobacco[22], cytosolic tRNA$^{Gln}_{CUG}$ in Arabidopsis was found to lack m²A modification. Intriguingly, Arabidopsis chloroplast tRNA$^{Ser}_{GGA}$ and elongator tRNA$^{Met}_{CAU}$ contain m²A modification, while tRNAs with anticodon ending in A or U typically possess a bulky modification (such as i⁶A, t⁶A, or their derivatives) at position 37 to enhance the stability of weak A-U interaction in bacteria and mammals[34].

Expanding our understanding of m²A distribution and conservation in diverse plant species, we isolated individual tRNA from various plants and analyzed digested nucleosides using LC-MS/MS. Thirteen selected tRNA species were tested, taking into account the m²A-modified tRNAs found in Arabidopsis and *E. coli* (Table 1). We identified two additional tRNA species (chloroplast tRNA$^{Gln}_{UUG}$ and chloroplast tRNA$^{Asp}_{GUC}$) carrying m²A in specific plants. Among all identified plant m²A-modified tRNAs, m²A in cytosolic tRNA$^{Gln}_{UUG}$ and chloroplast tRNA$^{Arg}_{ACG}$ is conserved across all examined species. In contrast, m²A was exclusively detected in chloroplast tRNA$^{Asp}_{GUC}$ from *C. reinhardtii*, while cytosolic tRNA$^{Gln}_{CUG}$, tRNA$^{Asp}_{GUC}$, tRNA$^{Glu}_{UUC}$, and chloroplast tRNA$^{Glu}_{UUC}$ lacked m²A across all investigated plants (Table 1). Collectively, we identified a total of eight m²A-modified tRNAs across different plant species.

## Precise m²A position and fraction are detected via the Malc method

To accurately determine the precise position and fraction of m²A within specific RNA molecules, we developed a quantification method called Malc (Mung Bean Nuclease (MBN)-assisted LC−MS/MS). This method utilized a modified version of the conventional MBN protection assay, enabling the detection of modifications at single-nucleotide resolution. As depicted in Fig. 2a, we synthesized a set of biotinylated antisense DNA probes (Probe X−1, Probe X, and Probe X + 1) with identical 5′ ends and one-nucleotide variations at the 3′ end, targeting the predicted position X of the RNA modification. These probes were individually incubated with total tRNA (or total RNA used for rRNA detection) and hybridized to the target RNA molecule, thus shielding specific RNA regions from MBN digestion. The retained biotinylated hybrids were subsequently isolated by streptavidin beads. The expected RNA fragments were eluted, digested and quantified by LC−MS/MS. Notably, the three protected fragments differ by one nucleotide at the 5′ end in a sequential manner. Through LC−MS/MS quantification, the fragments protected by Probe X−1 ($F_{X-1}$) and Probe X ($F_X$) exhibited signals corresponding to the RNA modification, whereas the fragments protected by Probe X + 1 ($F_{X+1}$) did not show such signals. This process confirmed the precise position of the modification. Ideally, the quantity of modified nucleotide in $F_{X-1}$ and $F_X$ should be equivalent. However, MBN displays partially double-stranded endonuclease activity when the RNA fragment possesses an A/U-rich end, resulting in a lower detection of modified nucleotide in $F_X$ (Fig. 2b, c). Therefore, we utilized the LC-MS/MS results of $F_{X-1}$ to calculate the m²A modification fraction in most RNA species, with the exception of two tRNA ($F_{X-2}$ for tRNA$^{Ser}_{GGA}$ and $F_{X-3}$ for elongator tRNA$^{Met}_{CAU}$).

Using the aforementioned Malc method, we successfully determined the precise location and fraction of m²A in six Arabidopsis tRNA species, Arabidopsis chloroplast 23S rRNA, spinach chloroplast tRNA$^{Gln}_{UUG}$, and *Chlamydomonas* chloroplast tRNA$^{Asp}_{GUC}$. Notably, in all the analyzed tRNAs, m²A was consistently found at position 37, with an average modification fraction of approximately 80%. Furthermore, we found that m²A is located at position 2521 of Arabidopsis chloroplast 23S rRNA with a 78% modification ratio, which is corresponds to

**Table 1 | Quantification of the m²A/A ratio of different tRNA species in the indicated plants**

|  | Arabidopsis thaliana | Oryza sativa | Spinacia oleracea | Physcomitrella patens | Chlamydomonas reinhardtii |
|---|---|---|---|---|---|
| Cytosolic tRNAs |  |  |  |  |  |
| ACG (Arg) | 3.75 ± 0.10[a] | (G)[b] | 3.69 ± 0.12 | (G) | (G) |
| UUG (Gln) | 5.47 ± 0.53 | 4.38 ± 0.14 | 5.70 ± 0.30 | 2.37 ± 0.084 | 2.27 ± 0.14 |
| CUG (Gln) | ND[c] | ND | ND | ND | ND |
| GUG (His) | (G) | (G) | (G) | (G) | (G) |
| UUC (Glu) | ND | ND | ND | ND | ND |
| GUC (Asp) | ND | ND | ND | ND | ND |
| Chloroplast tRNAs |  |  |  |  |  |
| ACG (Arg) | 2.70 ± 0.091 | 3.79 ± 0.15 | 2.82 ± 0.087 | 3.18 ± 0.16 | 2.61 ± 0.041 |
| UUG (Gln) | (G) | (G) | 5.67 ± 0.21 | 1.10 ± 0.048 | (G) |
| GUG (His) | 2.34 ± 0.12 | 4.11 ± 0.054 | 3.02 ± 0.12 | (G) | 1.86 ± 0.081 |
| UUC (Glu) | ND | ND | ND | ND | ND |
| GUC (Asp) | ND | ND | ND | ND | 2.56 ± 0.089 |
| **GGA (Ser)**[e] | 1.85 ± 0.029 | ND | 1.98 ± 0.19 | ND | NE[d] |
| **CAU (eMet)**[e] | 2.28 ± 0.024 | ND | 0.262 ± 0.012 | 2.22 ± 0.11 | 2.18 ± 0.13 |

[a]The data are presented in m²A/A (%) ± SD, *n* = 3 biological replicates.
[b](G) stands that the tRNA has guanosine at position 37.
[c]ND stands that the tRNA has adenosine at position 37 but m²A is not detectable.
[d]NE stands that the tRNA does not exist in the species.
[e]The tRNA species that do not bear m²A in *E. coli* are present in bold font. "eMet" stands for the elongator tRNA of methionine.

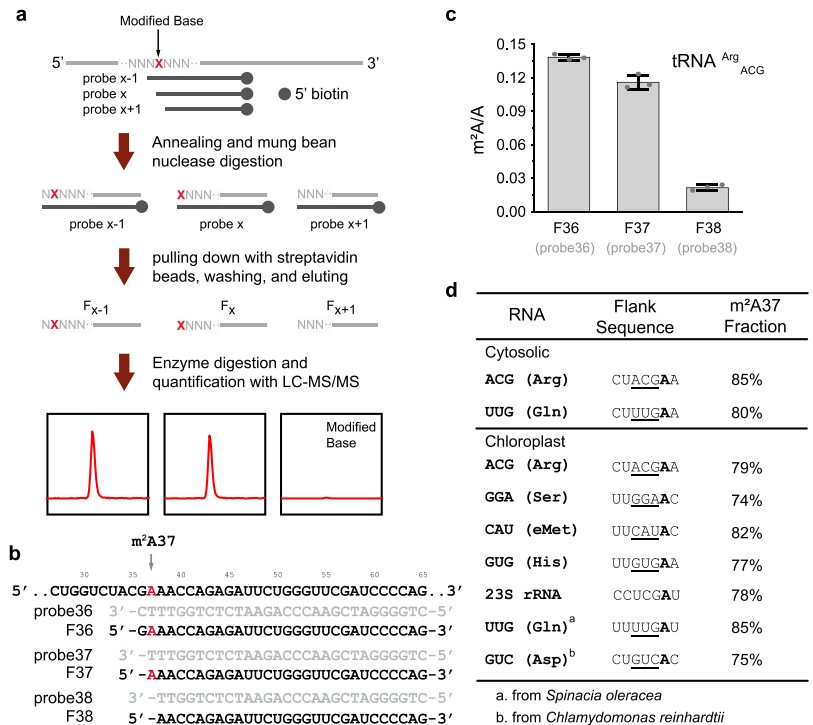

**Fig. 2 | Development of the "Malc" method to determine precise m²A position and fraction. a** The procedure of the "Malc" method. RNA is illustrated using light grey lines, and the DNA probes are illustrated using dark grey lines. The modified base in RNA is shown as a red "X" letter, and the unmodified bases around are shown as grey "N" letters. The 5′ biotin of probes is shown as grey circles. **b** DNA probes used in the case of cytosolic tRNA$^{Arg}_{ACG}$ and corresponding RNA fragments.

**c** Quantification of m²A levels in cytosolic tRNA$^{Arg}_{ACG}$ RNA fragments pulled down using representative probes. **d** Flank sequence and calculated m²A modification ratio in different RNA species. The adenosine modified as m²A is shown in bold font. Anti-codon of tRNA is indicated with an underline. Data are represented as means ± SD ($n = 3$ biological replicates) for (**c**).

the position of m²A2503 in *E. coli* 23S rRNA (Supplementary Fig. 5). Overall, these results indicate that m²A is situated at evolutionarily conserved positions and represents a high stoichiometry RNA modification.

### RlmN-like proteins of different clades are widespread in eukaryotes and expressed in Arabidopsis

Next we sought to identify RlmN homologs in Arabidopsis. BLAST analysis revealed three RlmN-like proteins, designated as RLMNL1 (At2g39670), RLMNL2 (At1g60230), and RLMNL3 (At3g19630), encoded in the Arabidopsis genome (Supplementary Fig. 6a). These proteins displayed high sequence similarity to *E. coli* RlmN (with BLAST *E*-value of 8e-65, 1e-76, and 3e-61 for RLMNL1, RLMNL2, and RLMNL3, respectively) and relatively lower similarity to *S. aureus* Cfr (with BLAST E-value of 6e-55, 3e-47, and 2e-52 for RLMNL1, RLMNL2, and RLMNL3, respectively). Conserved residues characteristic of RlmN, but distinct from Cfr were identified in RLMNL1, RLMNL2, and RLMNL3[35] (Supplementary Fig. 6b), corroborating the absence of m⁸A in Arabidopsis RNA samples. Sequence alignment further confirmed the conservation of crucial residues including C355 (corresponding to C398, C430, and C345 in RLMNL1, RLMNL2, and RLMNL3, respectively), which forms a covalent intermediate with substrate adenosine, C118 (C170, C197, and C118 in RLMNL1, RLMNL2, and RLMNL3, respectively), involved in resolving the cross-linked intermediate, the canonical CxxxCxxC iron-sulfur cluster binding motif, and the MGMGE motif responsible for electron transfers to the enzyme's iron-sulfur cluster (Supplementary Fig. 6b).

Further searches conduced against the NCBI protein database revealed the presence of RlmN-like proteins in various eukaryotes, including green plants, red algae, heterokonts, cryptomonads, haptophytes, and a limited number of fungi and chytrids species. Notably, RlmN-like proteins were found to be absent in animals and

the majority of fungi. Phylogenetic analyses illustrated that eukaryotic RlmN-like proteins can be categorized into six distinct clades (Fig. 3). Clade I, represented by Arabidopsis RLMNL1, exhibited the closest evolutionary relationship with RlmN from blue-green algae. Clade II, represented by Arabidopsis RLMNL2, shared an evolutionary branch with RlmN from deltaproteobacteria and was also found in red algae. Clade III, represented by Arabidopsis RLMNL3, appeared to have no close prokaryote relatives and was identified in green plants, heterokonts, and several chytrid species. Green plants did not possess clade IV, V, and VI RlmN-like proteins. Clade IV and V exhibited a similar distribution in heterokonts, cryptomonads, and haptophytes. Clade IV RlmN-like proteins displayed a close phylogenetic relationship with RlmN from chloroflexi, while clade V proteins lacked close prokaryote relatives. Clade VI proteins were exclusively found in a range of fungi and were closely related to RlmN from alphaproteobacterial. The intricate phylogeny and wide distribution of eukaryotic RlmN-like proteins suggest complex origins involving horizontal gene transfer (HGT) and conserved functional roles (Fig. 3).

To examine the expression patterns of *RLMNL1*, *RLMNL2*, and *RLMNL3*, total RNA was isolated from various organs of Arabidopsis, and their transcript levels were measured using qPCR (Supplementary Fig. 7a). The expression of *RLMNL1*, *RLMNL2*, and *RLMNL3* showed minimal or no variation across different organs, except for lower transcript levels observed in roots of seedlings for *RLMNL1* and *RLMNL2*. To investigate the subcellular location of these proteins, RLMNL1-GFP, RLMNL2-GFP, and RLMNL3-GFP fusion proteins were transiently expressed in wild-type Col-0 protoplasts. The results showed that RLMNL1 and RLMNL2 were exclusively located in chloroplasts, while RLMNL3 was excluded from chloroplast (Supplementary Fig. 7b), suggesting potential differences in substrate specificity among these three methyltransferases.

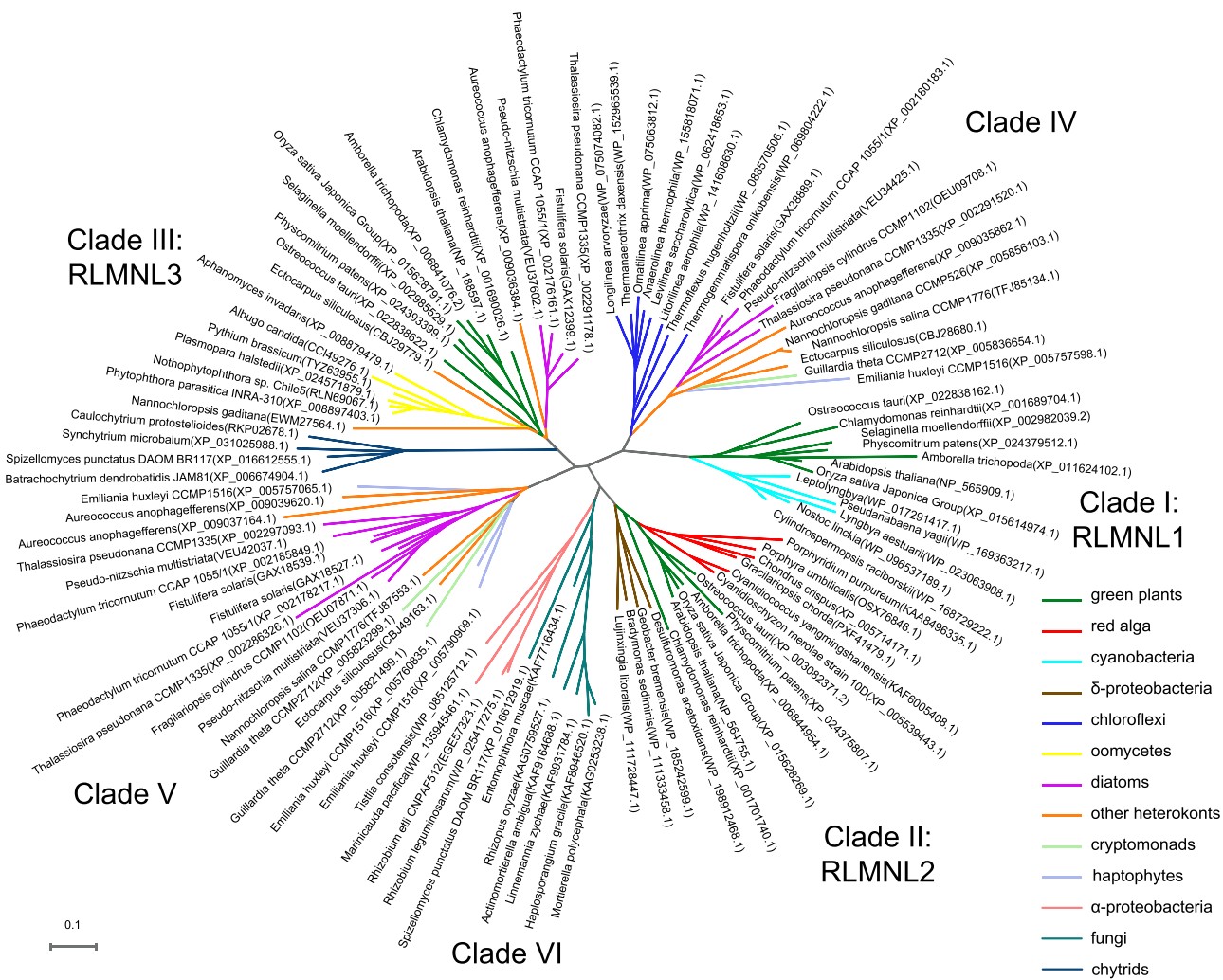

**Fig. 3 | The tree was generated from NCBI-BLAST.** RlmN-like protein sequences are labelled as their taxonomic name (with NCBI accession number). The branches are coloured according to the species classification.

## Characterization of RLMNL1-3 as m²A methyltransferases for chloroplast rRNA, chloroplast tRNA, and cytosolic tRNA, respectively

To investigate the in vivo enzymatic functions of RLMNL1-3, we characterized homozygous T-DNA insertion mutants *rlmnl1*, *rlmnl2*, and *rlmnl3*, respectively (Supplementary Fig. 8a). The T-DNA insertion in these mutants were positioned upstream of at least one active site of the proteins (Supplementary Figs. 6b and 8a). qPCR analysis confirmed the absence of the full-length transcripts of *RLMNL1*, *RLMNL2*, and *RLMNL3* in the respective mutants (Supplementary Fig. 8b). We then isolated chloroplast rRNA and tRNA, as well as cytosolic tRNA, from Col-0 and the mutant lines, and measured the m²A/A ratio using LC-MS/MS. The m²A level in chloroplast rRNA was drastically reduced to 1% in *rlmnl1* compared to Col-0, while no significantly change was observed in *rlmnl2* and *rlmnl3* (Fig. 4a), indicating that RLMNL1 is responsible for m²A methylation in chloroplast rRNA. In contrast, the m²A level in chloroplast tRNA showed a 99% decrease in *rlmnl2* compared to Col-0, while remaining unchanged in *rlmnl1* and *rlmnl3* (Fig. 4a), demonstrating the role of RLMNL2 in writing m²A in chloroplast tRNA. Additionally, the m²A modification in cytosolic tRNA was reduced to less than 2% in *rlmnl3* compared to Col-0, whereas no significant change was observed in *rlmnl1* or *rlmnl2* (Fig. 4a), revealing that RLMNL3 acts as a cytosolic tRNA m²A methyltransferase. These findings align with the subcellular localization of these proteins (Supplementary Fig. 7b).

To further validate the enzymatic activity of RLMNL1-3, we performed genetic complementation experiments in *rlmnl1*, *rlmnl2*, and *rlmnl3* mutants, generating two complemented lines for each mutant. The first set of lines, *RLMNL1:RLMNL1/rlmnl1*, *RLMNL2:RLMNL2/rlmnl2*, and *RLMNL3:RLMNL3/rlmnl3*, expressed wild-type methyltransferases, while the second set of lines, *RLMNL1:RLMNL1m/rlmnl1*, *RLMNL2:RLMNL2m/rlmnl2*, and *RLMNL3:RLMNL3m/rlmnl3*, expressed the catalytically inactive mutant proteins (RLMNL1 C398A, RLMNL2 C430A, and RLMNL3 C345A were termed as RLMNL1m, RLMNL2m, and RLMNL3m, respectively) (Supplementary Fig. 8b). The results demonstrated that the deficiency of m²A modification in chloroplast rRNA observed in *rlmnl1* was restored by the expression of wild-type RLMNL1 but not by the catalytically inactive mutant RLMNL1m (Fig. 4b). Similarly, the expression of functional *RLMNL2* in *rlmnl2* fully restored the m²A levels in the four identified m²A-modified chloroplast tRNAs compared to Col-0, whereas the m²A levels in *RLMNL2:RLMNL2m/rlmnl2* plants remained unchanged, similar to *rlmnl2* (Fig. 4c). Furthermore, the extent of m²A modification in the two m²A-methylated cytosolic tRNA species was rescued in *RLMNL3:RLMNL3/rlmnl3* plants but not in *RLMNL3:RLMNL3m/rlmnl3* plants (Fig. 4d). Overexpressed these three RlmN-like proteins individually in Col-0 did not alter the m²A levels compared to Col-0 (Supplementary Fig. 8b and Supplementary Fig. 9), providing further evidence that m²A is a constitutive modification. Collectively, these findings conclusively establish RLMNL1, RLMNL2, and RLMNL3 as m²A

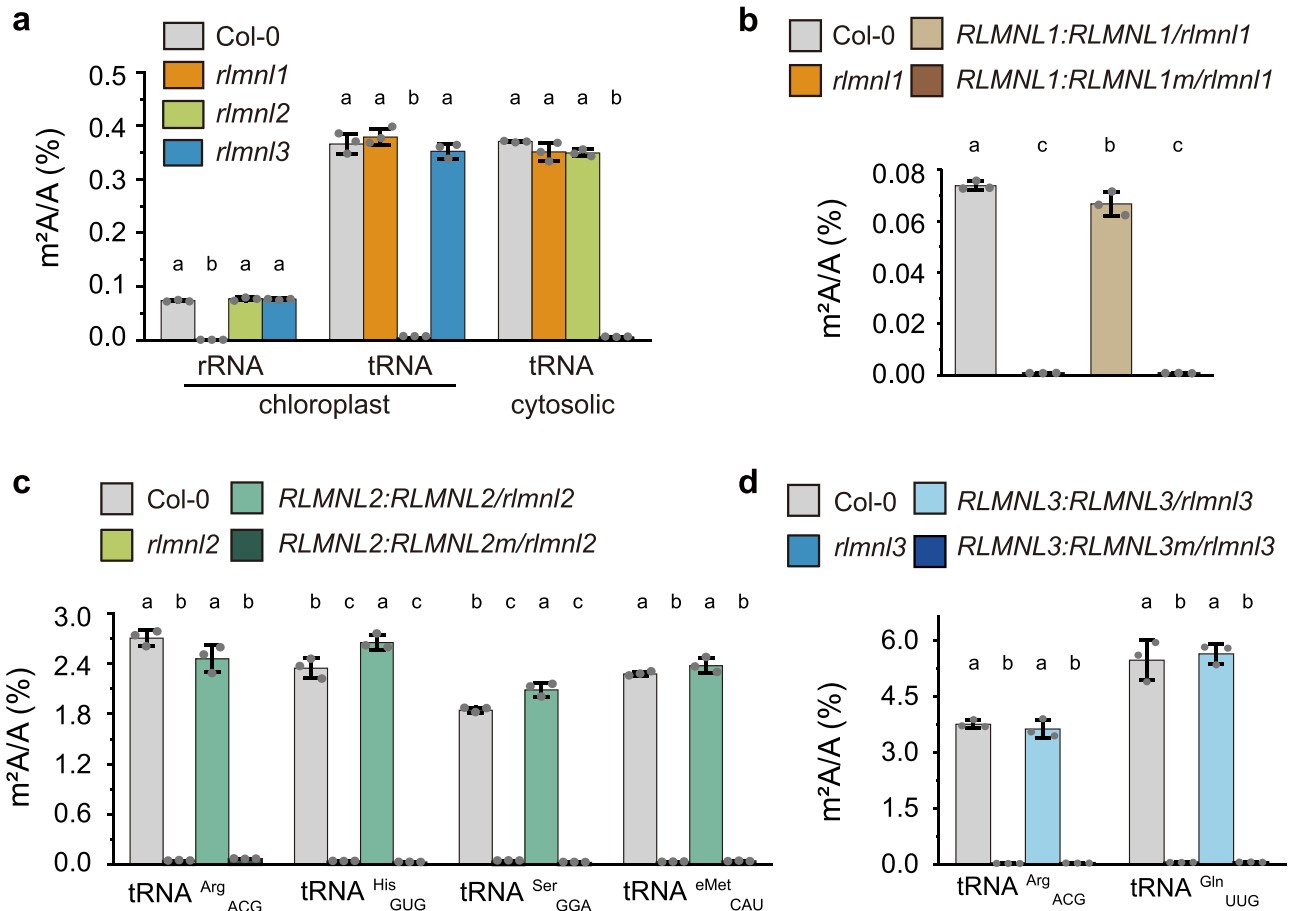

**Fig. 4 | RLMNL1, RLMNL2, and RLMNL3 install m²A modification in chloroplast rRNA, chloroplast tRNA, and cytosolic tRNA, respectively. a** Quantification of the m²A/A ratio of total chloroplast rRNA, chloroplast tRNA, and cytosolic tRNA in the indicated plant lines by LC–MS/MS. Quantification of the m²A/A ratio in total chloroplast rRNA (**b**), four specific m²A-modified chloroplast tRNAs (**c**), and two specific m²A-modified cytosolic tRNAs (**d**) in the indicated plant lines by LC–MS/MS. "eMet" stands for the elongator tRNA of methionine. Data are represented as means ± SD ($n = 3$ biological replicates). Different letters indicate significant differences at $p < 0.05$ (one-way ANOVA followed by Bonferroni post hoc test), and exact $p$-values are provided in Source Data.

methyltransferases in Arabidopsis, responsible for modifying chloroplast rRNA, chloroplast tRNA, and cytosolic tRNA, respectively.

**tRNA m²A37 does not affect tRNA stability and aminoacylation**

Despite the long-standing identification of m²A2503 in 23S rRNA and m²A37 in six tRNA species in *E. coli*, the molecular functions of m²A, particularly in tRNA, remain largely elusive. To address this, we conducted an investigation into the potential roles of m²A in RNA post-transcriptional fate by quantifying m²A-modified RNAs in both Col-0 and mutant lines lacking m²A methyltransferases. Initial analysis using an Agilent 2100 bioanalyzer revealed no significant changes in the amount of chloroplast rRNA between *rlmnl1* and Col-0 (Supplementary Fig. 10), indicating that m²A likely does not influence chloroplast rRNA transcription or stability. Subsequently, we employed RNA gel blot analysis to assess the levels of m²A37-modified tRNA, with U6 snRNA serving as the loading control. The results revealed that the absence of tRNA m²A37 modification in both *rlmnl2* and *rlmnl3* mutants had no discernible impact on the levels of any of the m²A-modified tRNAs compared to Col-0 (Supplementary Fig. 11). These results strongly indicate that the absence of m²A has minimal effects on RNA steady-state levels.

Given the structural observation of a hydrogen bond interaction between m²A37 and Asn370 in the *E. coli* glutaminyl-tRNA synthetase with tRNA^Gln,[36], we proceeded to investigate whether m²A37 could influence aminoacyl-tRNA synthesis. To assess this, we separated aminoacyl-tRNA and uncharged tRNA using an acidic urea-TBE

polyacrylamide gel. Deacylated tRNAs, obtained from total RNA treated with mild alkaline hydrolysis, were included as an indicator of uncharged tRNA. The results showed that the aminoacylation levels of m²A-modified tRNAs remained unaltered in *rlmnl2* and *rlmnl3* mutant plants compared to Col-0 (Supplementary Fig. 12). These findings suggest that m²A does not have an impact on aminoacyl-tRNA synthesis in Arabidopsis.

**m²A37 induces a relaxed conformation and decreases the melting temperature of tRNA**

Previous studies have indicated that RNA modifications at position 37 are crucial to the functional conformation of the anticodon loop in certain tRNAs[2,37,38]. Based on this, we hypothesized that m²A37 might play a role in regulating tRNA conformation. To investigate this, total RNA from Col-0 and mutant plants was subjected to electrophoresis on a native polyacrylamide gel in Tris-HEPES buffer. The RNA was subsequently transferred to a nylon membrane and hybridized with biotin-labelled probes specific to particular tRNA species[39]. Remarkably, the results revealed that m²A-modified chloroplast tRNAs (tRNA^Arg_ACG, tRNA^His_GUG, elongator tRNA^Met_CAU, and tRNA^Ser_GGA) from Col-0 exhibited slower migration compared to their m²A-deficient counterparts from *rlmnl2* (Fig. 5a). Similarly, cytosolic m²A-modified tRNAs (tRNA^Arg_ACG and tRNA^Gln_UUG) from Col-0 displayed a slower migration in native PAGE relative to the m²A-deficient counterparts from *rlmnl3* (Fig. 5b). As negative controls, the non-m²A-modified chloroplast tRNA^Gln_UUG and cytosolic tRNA^His_GUG from Col-0 and *rlmnl2*

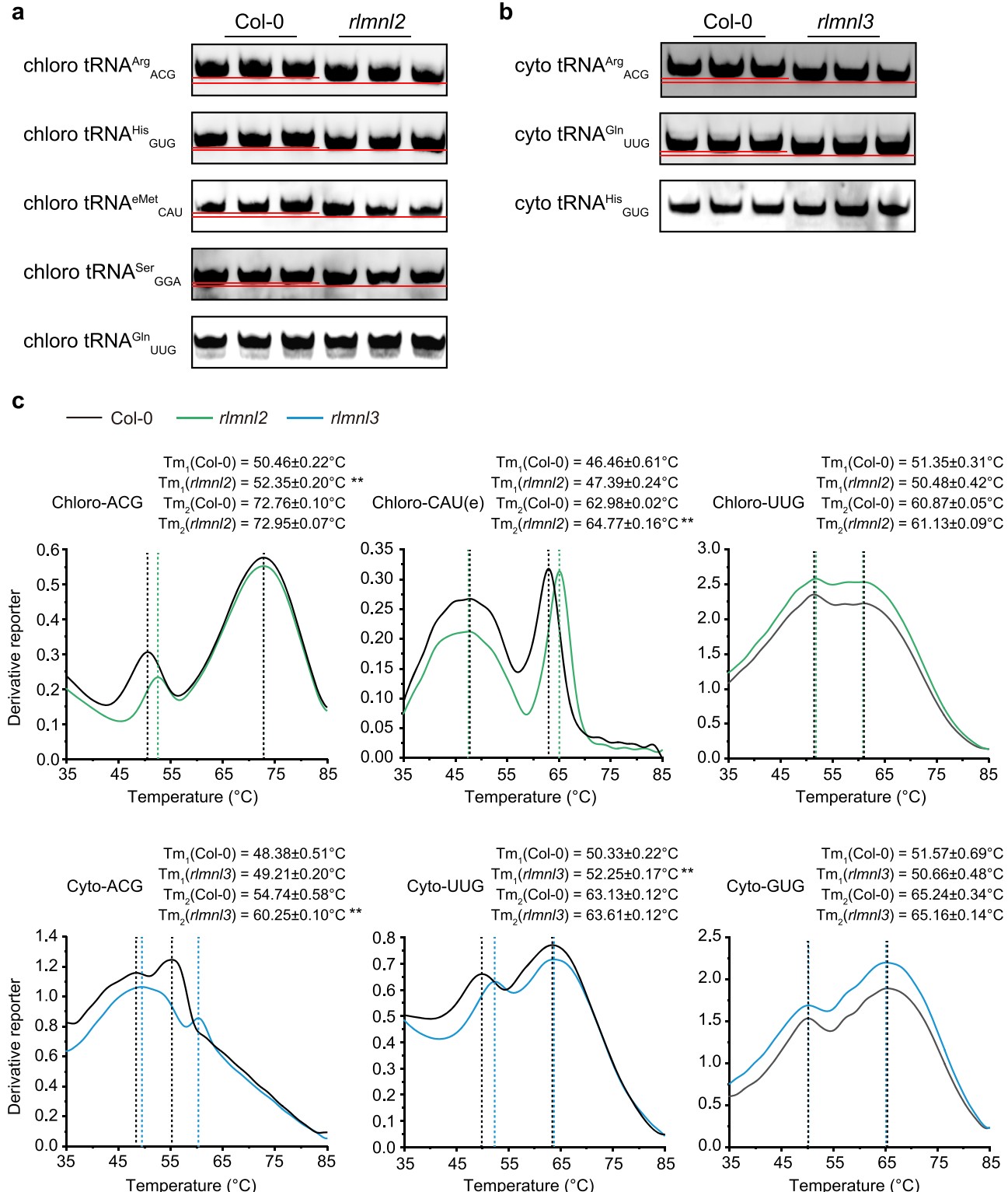

**Fig. 5 | tRNA m²A modification helps regulate tRNA conformation. a** The native-PAGE results of chloroplast m²A-modified tRNAs from Col-0 or *rlmnl2*. Chloroplast tRNA$^{Gln}_{UUG}$ was used as the negative control. **b** The native-PAGE results of cytosolic m²A-modified tRNAs from Col-0 or *rlmnl3*. Cytosolic tRNA$^{His}_{GUG}$ was used as negative control. **c** The derivative melting curve plot of m²A-modified tRNAs from Col-0 or mutant plants. Chloroplast tRNA$^{Gln}_{UUG}$ and cytosolic tRNA$^{His}_{GUG}$ were used as negative control. The peak on each curve indicates the transition point (melting temperature). Three biological replicates were run parallelly in (**a**, **b**). Data in (**c**) are represented as mean ± SEM (*n* = 3 biological replicates). The double asterisk indicates significant differences at *p* < 0.01 (paired two-sided student's t-test), and exact *p*-values are provided in Source Data.

or *rlmnl3* mutant exhibited similar migration velocities in the native PAGE (Fig. 5a, b). It is important to note that compactly folded RNA migrates faster than loosely folded RNA with the same sequence in native PAGE. These results indicate that the m$^2$A-deficient tRNAs adopt a more compactly folded architecture compared to their m$^2$A-modified counterparts.

Conformation changes in tRNA are often associated with alterations in thermostability. Therefore, we measured melting temperature (T$_m$) of m$^2$A-modified tRNAs from Col-0 and their m$^2$A-deficient counterparts from the mutant plants. Typically, the melting curves of tRNA exhibit two transition points: a lower (T$_{m1}$) and a higher transition point (T$_{m2}$), which are indicative of the dissolution temperatures of tertiary and secondary structure, respectively[40]. The results showed that deficiency of m$^2$A significantly elevated T$_{m1}$ in chloroplast tRNA$^{Arg}_{ACG}$ and cytosolic tRNA$^{Gln}_{UUG}$, while T$_{m2}$ was increased in chloroplast elongator tRNA$^{Met}_{CAU}$ and cytosolic tRNA$^{Arg}_{ACG}$ in response to m$^2$A defects (Fig. 5c). As negative controls, the non-m$^2$A-modified chloroplast tRNA$^{Gln}_{UUG}$ and cytosolic tRNA$^{His}_{GUG}$ from both Col-0 and *rlmnl2* or *rlmnl3* mutant exhibited similar melting temperature (Fig. 5c). Since modifications in the anticodon loop are unlikely to affect the L-shape tertiary structure of tRNA, a plausible explanation is that m$^2$A impacts intra-loop base stacking (potentially contributing to T$_{m1}$) in chloroplast tRNA$^{Arg}_{ACG}$ and cytosolic tRNA$^{Gln}_{UUG}$, or intra-loop base pairing (possibly contributing to T$_{m2}$) in chloroplast elongator tRNA$^{Met}_{CAU}$ and cytosolic tRNA$^{Arg}_{ACG}$. These findings are consistent with the gel migration results, as changes in intra-loop base interaction influence RNA folding. Intriguingly, the melting curves of chloroplast tRNA$^{His}_{GUG}$ and tRNA$^{Ser}_{GGA}$ did not show distinct transition points, and their melting temperatures were not significantly changed in *rlmnl2* (Supplementary Fig. 13). The detailed mechanism by which m$^2$A influences the structure of these tRNAs warrants further investigation.

## m$^2$A in tRNA enhances protein translation by enabling decoding of m$^2$A-tRNA-dependent codons

Due to the dual-specificity of bacterial RlmN, previous studies using *Δrlmn E. coli* stain were unable to determine the distinct biological roles of rRNA m$^2$A and tRNA m$^2$A[24]. However, in Arabidopsis, the substrates of RlmN-like proteins are specialized, allowing us to investigate the individual functions of rRNA and tRNA m$^2$A in protein translation. To accomplish this, we performed the puromycin labelling assay, which measures the incorporation of puromycin into newly synthesized proteins. By immunoblotting with an anti-puromycin antibody, we assessed the rate of new protein synthesis in Col-0 and the mutant lines lacking the three m$^2$A methyltransferases. Specifically, we treated cytosolic and chloroplast nascent peptides with puromycin under different conditions (see methods). The results demonstrated a decrease in chloroplast protein synthesis in *rlmnl2* (Fig. 6a) and a reduction in cytosol protein synthesis in *rlmnl3* (Fig. 6b). Conversely, the nascent protein level remained unaffected in *rlmnl1* under both conditions (Fig. 6a, b), indicating that tRNA m$^2$A37, but not rRNA m$^2$A, plays a crucial role in facilitating translation.

To further investigate the impact of tRNA m$^2$A37 on translation through the decoding function of individual tRNA rather than the disruption of protein translation machinery, we conducted a dual-luciferase assay using protoplasts derived from 4-week-old leaves of wild-type Col-0, *rlmnl3*, *RLMNL3:RLMNL3/rlmnl3*, and *RLMNL3:RLMNL3m/rlmnl3*. As mentioned previously, cytosolic tRNA$^{Arg}_{ACG}$ and tRNA$^{Gln}_{UUG}$ are modified with m$^2$A by RLMNL3 (Fig. 4d). According to the wobble hypothesis, tRNA$^{Arg}_{A(I)CG}$ can recognize CGA, CGC, and CGU codons, while tRNA$^{Gln}_{UUG}$ can recognize CAA and CAG codons. To assess the decoding efficiency, we introduced six repeated codon sequences (6 × CGU, 6 × CGC, and 6 × CGA for tRNA$^{Arg}_{ACG}$, 6 × CAA and 6 × CAG for tRNA$^{Gln}_{UUG}$, and 6 × CGG for non-m$^2$A-modified tRNA$^{Arg}_{CCG}$ as a negative control) upstream of the firefly luciferase (F-luc) gene in a dual-luciferase reporter vector (Fig. 6c).

Renilla luciferase (R-luc) encoded in the same vector was used to normalize transformation efficiency. The empty control reporter lacking the 6 × codon sequence was employed to normalize transcription levels across protoplasts of different genotypes (Fig. 6c).

The results showed translation levels of the empty control reporter and the negative control reporter with 6 × CGG for non-m$^2$A-modified tRNA$^{Arg}_{CCG}$ have no significant differences among protoplasts of different genotypes (Fig. 6d). However, for tRNA$^{Gln}_{UUG}$, translation levels of both 6 × CAA and 6 × CAG reporters decreased by more than 70% in *rlmnl3* protoplasts compared to Col-0. The introduction of wild-type *RLMNL3*, but not the inactive mutant *RLMNL3m* with C345A mutation, rescues the translation effects in the *rlmnl3* mutant background (Fig. 6d). These findings indicate that the decoding of CAA and CAG codons is impaired by the deficiency of m$^2$A in tRNA$^{Gln}_{UUG}$. Similarly, we observed decreased translation levels of 6 × CGT and 6 × CGC reporters for tRNA$^{Arg}_{A(I)CG}$ in *rlmnl3* protoplasts compared to Col-0. This reduction in translation was recovered in *RLMNL3:RLMNL3/rlmnl3* but not in *RLMNL3:RLMNL3m/rlmnl3* (Fig. 6d). However, no significant differences were observed in the translation of 6 × CGA reporter among protoplasts of different genotypes (Fig. 6d), suggesting that the CGA codon might not be decoded by tRNA$^{Arg}_{A(I)CG}$. This is consistent with previous studies showing that the presence of m$^2$A at position 37 in the anticodon stem-loop of *E. coli* tRNA$^{Arg}_{A(I)CG}$ prevents binding to the CGA codon in vitro[41]. We speculate that the CGA codon could be decoded by tRNA$^{Arg}_{UCG}$, which is present in plants. Based on the fact that the codons CAA, CAG, CGU, and CGC are decoded by m$^2$A-modified tRNAs, we refer to them as m$^2$A-tRNA-dependent codons in the cytosol (Supplementary Fig. 14).

According to the wobble hypothesis and the "two out of three" hypothesis, which suggest that a tRNA pairing with only the first two codon bases is sufficient for translation[42], the m$^2$A-tRNA-dependent codons in chloroplast can be deduced (Supplementary Fig. 14). Inspired by the dual-luciferase assay, we examined all chloroplast-encoded genes to identify those containing tandem m$^2$A-tRNA-dependent codons. We found that genes with five or more tandem m$^2$A-tRNA-dependent codons do not exist in the chloroplast. However, 14 chloroplast-encoded genes have four or three m$^2$A-tRNA-dependent codons in tandem (Supplementary Fig. 15a). From these genes, we selected *NDHH*, *PSBA*, and *RPOC2* to investigate whether their translation is affected by by m$^2$A deficiency (Supplementary Fig. 15b). Protein gel blotting results showed that two out of the three genes (*NDHH* and *PSBA*) exhibited decreased protein levels by about 30% in *rlmnl2*, but not in *rlmnl1* and *rlmnl3* (Fig. 6e and Supplementary Fig. 16a). However, the protein level of RPOC2 was not significantly affected in the mutant plants (Supplementary Fig. 16b, c). Interestingly, the transcription of *NDHH* and *RPOC2* were significantly elevated in *rlmnl2* compared to Col-0, while the mRNA level of *PSBA* remained unchanged in the mutant plants (Supplementary Fig. 16d). To further assess the effects of m$^2$A deficiency on protein translation, we calculated the apparent translation efficiency (TE) by dividing the relative protein amount by the mRNA level of the corresponding gene. The results showed that the TE of all three genes was reduced in *rlmnl2* (Supplementary Fig. 16e). Collectively, our results demonstrate that tRNA m$^2$A37 facilitates protein translation through the decoding of m$^2$A-tRNA-dependent codons in cytosol and chloroplast. Additionally, it raised the hypothesis that m$^2$A37 may enhance translation efficiency by decoding tandem m$^2$A-tRNA-dependent codons.

## tRNA m$^2$A37 promotes translation efficiency by decoding tandem m$^2$A-tRNA-dependent codons in the cytosol

To gain further insights into the mechanism underlying the impact of m$^2$A on protein translation, we performed three independent polysome profiling experiments in both Col-0 and *rlmnl3*. The results revealed a slight increase in 80 S monosome peaks and a reduction in polysome fractions upon disruption of *RLMNL3* (Fig. 7a and

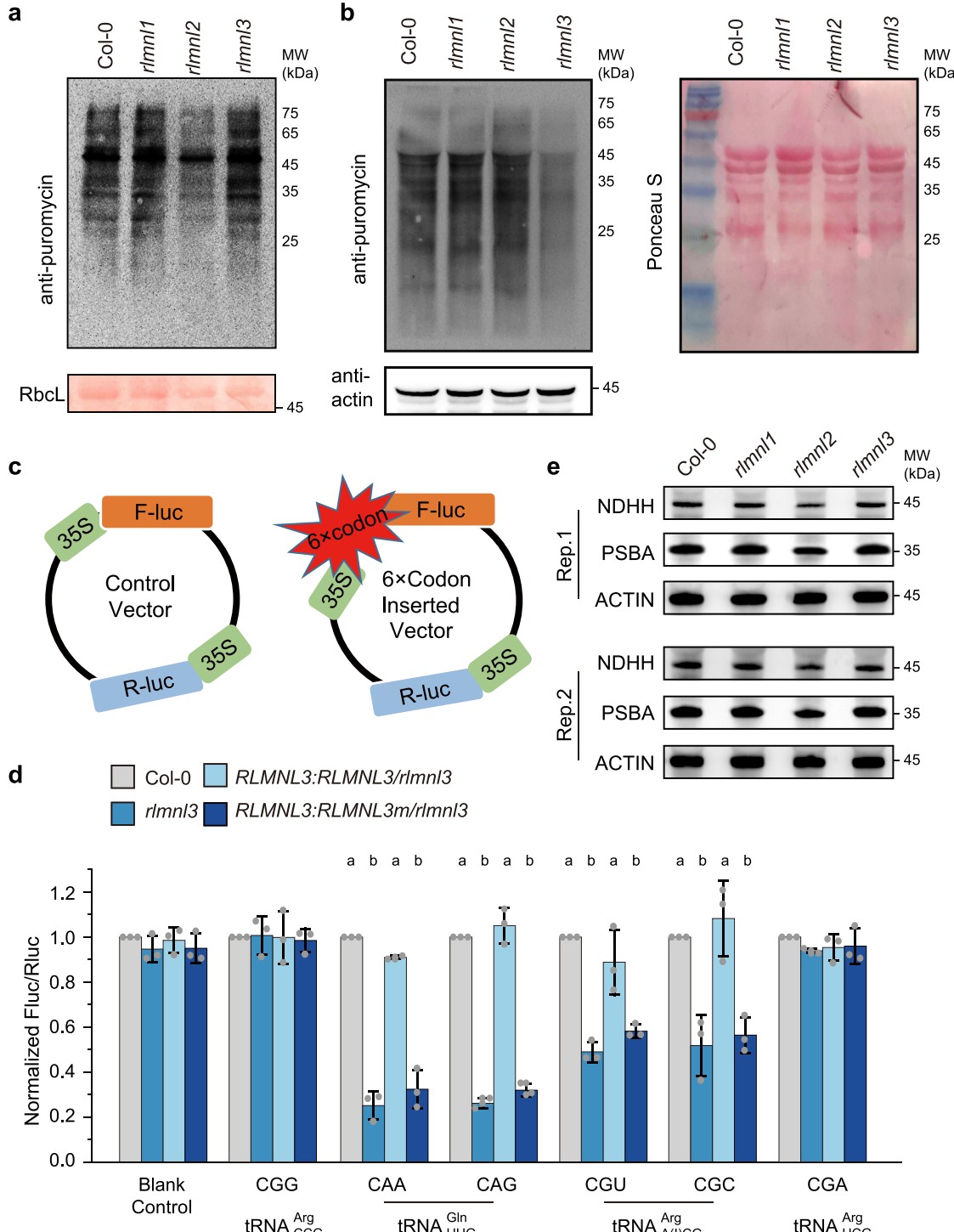

**Fig. 6 | tRNA m²A37 facilitates protein translation through the decoding of m²A-tRNA-dependent codons. a** Puromycin-labelling showing the newly translated chloroplast proteins in Col-0, *rlmnl1*, *rlmnl2*, and *rlmnl3*. Rubisco large subunit (RbcL) stained by Ponceau S was used as a loading control. Experiments were repeated for three times with similar results. **b** Puromycin-labelling showing the newly translated nuclear-encoded proteins in Col-0, *rlmnl1*, *rlmnl2*, and *rlmnl3*. Actin and total proteins visualized by Ponceau S were used as loading controls. Experiments were repeated for three times with similar results. **c** Scheme of the translation reporter assay. Both vectors encode dual luciferases—firefly luciferase (F-luc) as the primary translation reporter and *Renilla* luciferase (R-luc) as internal

transfection control. 6 × codon was inserted after the 35 S promoter of F-luc. The control reporter lacking 6 × codon insertion was used to normalize the translation differences between different samples. **d** Translation level reflected by normalized luciferase activity (F-luc/R-luc) of the control vector and 6 × codon inserted reporters transfected in protoplasts of the indicated plant lines. **e** Two biological replicates of protein immunoblotting reflecting NDHH and PSBA protein levels in Col-0, *rlmnl1*, *rlmnl2*, and *rlmnl3*. Actin was used as a loading control. Data in (**d**) are represented as means ± SD (*n* = 3 biological replicates). Different letters indicate significant differences at *p* < 0.05 (one-way ANOVA followed by Bonferroni post hoc test), and exact *p*-values are provided in Source Data.

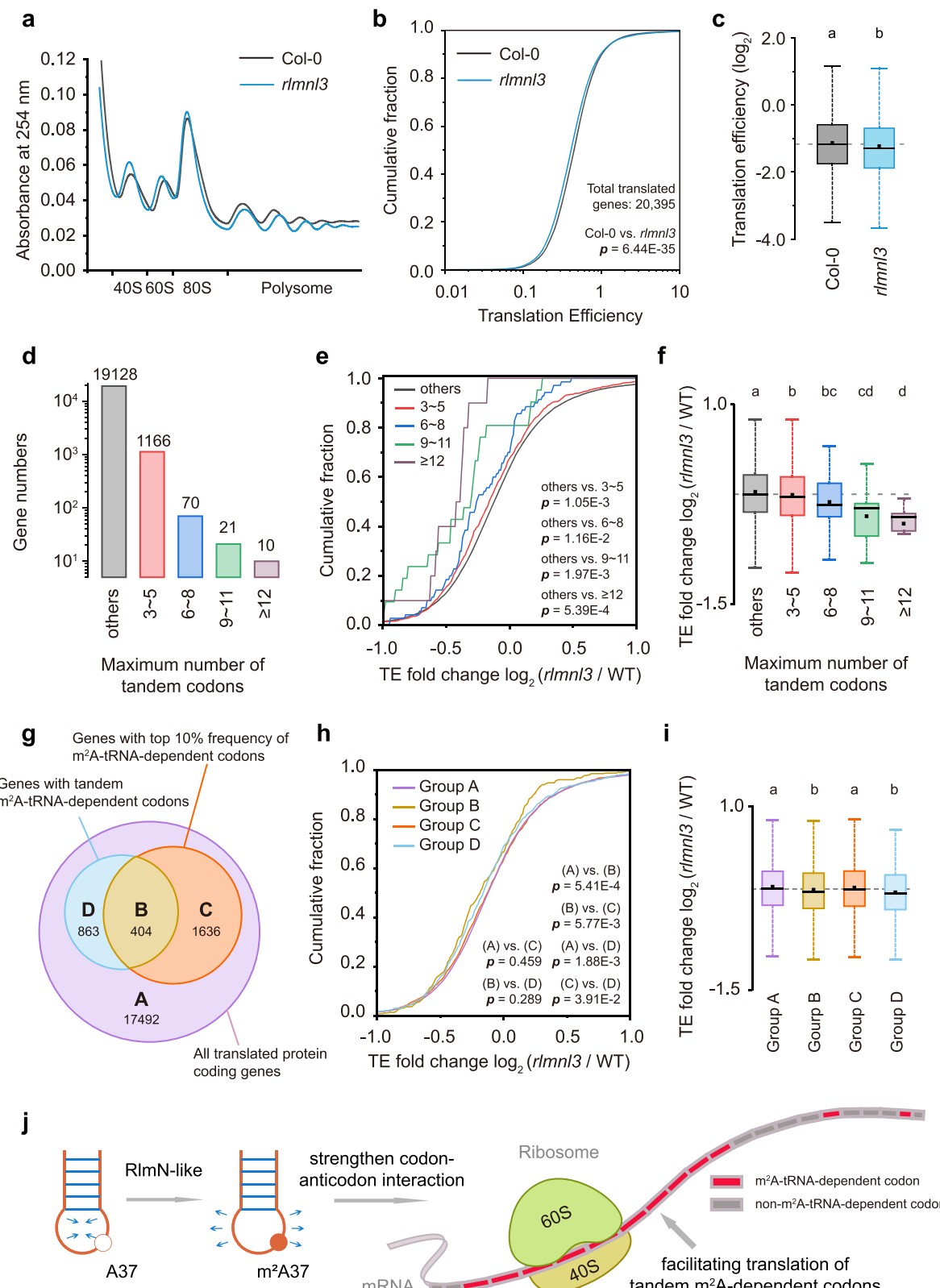

**Fig. 7 | tRNA m²A promotes the translation of tandem m²A-tRNA-dependent codons. a** Polysome profiling of Col-0 and *rlmnl3*. **b, c**, Cumulative frequency plots (**b**) and box plots (**c**) showing the overall translation efficiency of Col-0 and *rlmnl3*. **d** All nuclear-encoded genes grouped by the maximum number of tandem cytosolic m²A-tRNA-dependent codons. Cumulative frequency plots (**e**) and box plots (**f**) showing TE fold change (*rlmnl3*/WT) in genes grouped by the maximum number of tandem m²A-tRNA-dependent codons. Gene numbers of each group are shown in (**d**). **g** All nuclear-encoded genes grouped by whether having high m²A-tRNA-dependent codon frequency (top 10% in ranking) or containing no less than three tandem m²A-tRNA-dependent codons. Cumulative frequency plots (**h**) and box plots (**i**) showing TE fold change (*rlmnl3*/WT) in genes grouped as (**g**) shown. Gene numbers of each group are shown in (**g**). **j** A brief model of m²A37 modification facilitating translation. *p*-values were obtained from Kruskal–Wallis test followed by paired comparison. Box plots show the data distribution through median (center line), mean (center dot), first and third quartiles (filled box), and 1.5× interquartile range (whiskers). Different letters indicate significant differences at *p* < 0.05, and exact *p*-values are provided in Source Data.

Supplementary Fig. 17). Additionally, we conducted ribosome footprinting sequencing (Ribo-seq) and mRNA sequencing in Col-0 and *rlmnl3*. Translation efficiency (TE) of nuclear-encoded genes was calculated by dividing ribosome protected fragments by mRNA-seq FPKM (fragments per kilobase per million fragments) of the corresponding genes (Supplementary Fig. 18 and Supplementary Dataset 1). We observed a significantly decreased in overall TE in *rlmnl3* compared to Col-0 (Fig. 7b, c, and Supplementary Dataset 2), which is consistent with the findings from the puromycin labelling and polysome profiling assays (Figs. 6b and 7a). In contrast, the defects in *RLMNL3* led to marginal differences in mRNA levels between Col-0 and *rlmnl3* (cutoff criteria: FPKM fold change > 2; adjusted $p$-value < 0.05; $SNR$ > 1, see "Methods") (Supplementary Fig. 19 and Supplementary Dataset 3), indicating that m²A has minimal effect on gene expression.

Following the observations from the dual-luciferase assay, where tandem m²A-tRNA-dependent codons were associated with reduced protein translation in *rlmnl3*, we sought to investigate whether the global TE decrease in *rlmnl3* could be attributed to the impaired translation of genes containing tandem m²A-tRNA-dependent codons. We grouped nuclear-encoded genes based on the maximum number of tandem cytosolic m²A-tRNA-dependent codons (CAA, CAG, CGU, and CGC) they possessed, while the remaining genes (containing less than three aforementioned tandem codons) were categorized as "others" (Fig. 7d and Supplementary Fig. 20). Overall, the translation efficiency of the genes with at least three tandem m²A-tRNA-dependent codons was significantly decreased in *rlmnl3* compared to the "others" group (Fig. 7e). Moreover, the extent of TE reduction in *rlmnl3* showed an increasing trend as the number of tandem codons in genes increased (Fig. 7f). As negative controls, we selected six sets of non-m²A-dependent codons (four codons per set) and grouped all nuclear-encoded genes based on the maximum number of these codons in tandem. Analysis of their translation efficiency variation revealed no obvious difference between genes with three or more tandem non-m²A-dependent codons and other genes upon disruption of *RLMNL3* (Supplementary Fig. 21). These results suggest that m²A promotes translation efficiency by facilitating the decoding of tandem m²A-tRNA-dependent codons.

To provide further evidence supporting the impact of m²A deficiency on the decoding of tandem m²A-tRNA-dependent codons, we employed dual-luciferase assays using protoplasts isolated from 4-week-old leaves of wild-type Col-0, *rlmnl3*, *RLMNL3:RLMNL3/rlmnl3*, and *RLMNL3:RLMNL3m/rlmnl3*. We inserted three naturally existing tandem m²A-tRNA-dependent codon sequences found in protein-coding genes (CAA-CGU-CGC-CGU from *AT2G18500*, CAG-CGU-CAA-CAA from *AT2G33640*, and CAA-CAG-CGU-CAA-CAA-CAG from *AT3G03460*) into the dual-luciferase reporter construct as shown in Fig. 6c (termed as S1, S2, and S3, respectively). As a negative control, a sequence without m²A-tRNA-dependent codons (GUG-GUG-ACA-ACA, termed as S4) was included. The results showed a significant decrease in translation levels of all three reporters containing S1, S2, and S3 in *rlmnl3* protoplasts compared to Col-0, which was recovered in *RLMNL3:RLMNL3/rlmnl3* but not in *RLMNL3:RLMNL3m/rlmnl3* (Supplementary Fig. 22). No notable differences were observed in the translation of the empty control reporter and the negative control reporter among different genotypic protoplasts. These results provide compelling evidence that the translation of tandem m²A-tRNA-dependent codons is diminished in the absence of tRNA m²A37 modification.

Previous studies have demonstrated a correlation between the frequency of m⁷G-tRNA-dependent codons in genes and alterations in TE upon depletion of m⁷G methyltransferase METTL1[43]. Similarly, in our investigation, we observed that genes with a high frequency of m²A-tRNA-dependent codons (top 10% ranking) exhibited a significant overall decrease in TE compared to genes with a low m²A-tRNA-dependent codon frequency (bottom 10% ranking) in *rlmnl3* mutant

plants (Supplementary Fig. 23). To address whether the reduced overall TE in *rlmnl3* for genes containing tandem m²A-tRNA-dependent codons might be attributed solely to their higher codon frequency, we grouped all translated nuclear-encoded genes based on whether they had a high m²A-tRNA-dependent codon frequency (top 10% ranking) or containing no less than three tandem m²A-tRNA-dependent codons, and analyzed their TE alteration (Fig. 7g). The results showed that irrespective of having a high m²A-tRNA-dependent codon frequency, genes with tandem m²A-tRNA-dependent codons (Group B and D in Fig. 7h, i) exhibited significantly reduced translation efficiencies in *rlmnl3* compared to genes without tandem codons (Group A and C in Fig. 7h, i). However, no significant TE variation in *rlmnl3* was observed between genes with a high m²A codon frequency but without tandem m²A codons (Group C in Fig. 7h, i) and genes with neither tandem m²A codons nor high m²A codon frequency (Group A in Fig. 7h, i). These results indicate that tandem m²A-tRNA-dependent codons affect protein translation independently of codon frequency, and further support the hypothesis that tRNA m²A37 promotes translation efficiency by decoding tandem m²A-tRNA-dependent codons (Fig. 7j).

### Defects of *RLMNL1-3* lead to hypersensitivity to aminoglycoside antibiotics and slight enhancement of osmotic stress resistance

To assess the physiological roles of m²A methyltransferase, we carried out a series of phenotypic analyses on both wild-type and mutant plants. Disruption of *RLMNL1*, *RLMNL2*, or *RLMNL3* led to elevated sensitivity to aminoglycoside antibiotics, specifically paromomycin and G418, which are known to disturb protein translation. All three mutant lines showed delayed germination (Supplementary Fig. 24a) and suppressed early-stage vegetative growth (Supplementary Fig. 24b) under antibiotics treatment. These phenotypes can be recovered in *RLMNL1:RLMNL1/rlmnl1*, *RLMNL2:RLMNL2/rlmnl2*, and *RLMNL3:RLMNL3/rlmnl3* lines, but not in *RLMNL1:RLMNL1m/rlmnl1*, *RLMNL2:RLMNL2m/rlmnl2*, and *RLMNL3:RLMNL3m/rlmnl3* lines expressing enzymatically inactive mutant protein. Importantly, in the absence of antibiotics, no significant differences were observed in germination and root length among the various genotypes.

We further examined the response of the mutants to abiotic stresses and observed a slight increase in resistance to mannitol-induced osmotic stress in *rlmnl1*, *rlmnl2*, and *rlmnl3* mutants (Supplementary Fig. 24c). The complementary lines *RLMNL1:RLMNL1/rlmnl1*, *RLMNL2:RLMNL2/rlmnl2*, and *RLMNL3:RLMNL3/rlmnl3* had no significant difference in growth inhibition compared to Col-0, while *RLMNL1:RLMNL1m/rlmnl1*, *RLMNL2:RLMNL2m/rlmnl2*, and *RLMNL3:RLMNL3m/rlmnl3* lines exhibited similar resistance to *rlmnl1*, *rlmnl2*, and *rlmnl3* mutants (Supplementary Fig. 24c). Notably, no significant differences were observed in germination under salt stress (125 mM NaCl) or survival ratio after heat shock (45 °C for 30 min) (Supplementary Fig. 24d, e). It is worth mentioning that although the three m²A methyltransferases have different substrates, *rlmnl1*, *rlmnl2*, and *rlmnl3* exhibited similar phenotypes under the tested conditions. However, the precise mechanisms by which RLMNL1, RLMNL2, and RLMNL3 respond to aminoglycoside antibiotics and osmotic stress require further investigated in future studies.

## Discussion
The functions of various tRNA modifications at position 37, such as m¹G, yW, t⁶A, and i⁶A, have been extensively studied. However, the molecular function of tRNA m²A37 has received comparatively less attention. In our study, using a native-PAGE assay, we discovered that m²A-modified tRNAs exhibit a more relaxed architecture compared to their m²A-deficient counterparts. Interestingly, our findings differ from a previous study showing similar migration velocities of synthesized *E. coli* tRNA^Arg anticodon stem-loops with or without m²A using Tris-borate as the buffering agent, suggesting no significant structural alteration[41]. We hypothesize that the disparity in our results may be the

different electrophoresis conditions employed. Tris-borate used in previous study was reported to destabilize RNA folding. Instead, we utilized Tris-HEPES buffer in and optimized the $Mg^{2+}$ concentration based on the method of RNA folding analysis[39] (Supplementary Fig. 25). Corresponding to the gel migration results, four out of the six $m^2A$-modified tRNA in Arabidopsis exhibited increased melting temperature upon $m^2A$ defects. Additional investigation such as NMR experiments or molecular dynamics simulations could gain further insights into the specific structure alteration of $m^2A$-deficient tRNAs.

Unlike the dual substrates of *E. coli* RlmN, the substrate specialization of plant $m^2A$ methyltransferases allows for separate investigations into the biological roles of $m^2A$ in tRNA and rRNA. The $m^2A2503$ of 23 S rRNA controls translational accuracy in *E. coli*[24]. Indeed, we found removal of $m^2A$ in the position 2521 of Arabidopsis chloroplast 23 S rRNA does not affect the translation efficiency of chloroplast proteins (Fig. 6a, b). Through puromycin labelling experiments, we found that $m^2A37$ in both chloroplast tRNA and cytosolic tRNA promotes protein translation. This finding was further validated by dual-luciferase assays (Fig. 6d), which demonstrated that $m^2A37$ in cytosolic tRNA enhances protein translation by decoding $m^2A$-tRNA-dependent codons in cytoplasm. We also found 14 chloroplast-encoded genes have four or three tandem $m^2A$-tRNA-dependent codons and the translation efficiencies of three randomly selected genes among them, *NDHH*, *PSBA*, and *RPOC2*, were indeed decreased in *rlmnl2* (Supplementary Fig. 16e). Our ribosome footprinting sequencing results also showed that tRNA $m^2A37$ modification also promote the translation of tandem $m^2A$-tRNA-dependent codons in cytoplasm.

Previous studies have shown that rare or unfavourable codons in tandem can influence translation, as exemplified by the case of the two tandem UGG codons in the Trp operon[44,45]. Here we demonstrate that $m^2A37$ in both chloroplast tRNA and cytosolic tRNA enhances protein translation efficiency by facilitating the decoding of tandem $m^2A$-tRNA-dependent codons. We further investigated the impact of $m^2A$ on tRNA structure and observed that $m^2A37$ promotes a more relaxed conformation in the anticodon stem-loop of tRNA. Combining these findings, we proposed a mechanism for $m^2A37$ in chloroplast tRNA and cytosolic tRNA that promotes protein translation. In this mechanism, $m^2A$ strengthens the codon-anticodon interaction by maintaining an open-loop conformation in the anticodon stem-loop, thereby accelerating translation elongation and preventing translation pauses specifically at tandem $m^2A$-tRNA-dependent codons (Fig. 7j). It is conceivable that other tRNA modifications may exert a similar influence on translation. Exploring these possibilities warrants further investigation and may provide valuable insights into the broader regulatory mechanisms governing protein synthesis.

We developed a quantification method called MBN-assisted LC-MS/MS (Malc) to accurately determine the location and fraction of $m^2A$ in rRNA and tRNA at single-nucleotide resolution. The Malc method can be applied to other modifications in various RNA types, although it requires priori knowledge of the modified nucleotide. Previous studies on the mechanism of *E. coli* RlmN have revealed that it can form a covalent intermediate between the protein and RNA during the reaction. Mutant variants of RlmN with C118A or C118S mutation are unable to resolve this intermediate, leading to the formation of a stable cross-linked protein-RNA complex[27,28]. Exploiting this property, RlmN C118A mutant protein was used to solve the structure of RlmN-tRNA complex[46] and identify substrate RNAs using high-throughput sequencing (termed miCLIP-MaPseq)[47]. The miCLIP-MaPseq approach provides a paradigm to investigate the potential substrates of other RlmN homologs. It is important to note that the possibility of eukaryotic RlmN-like proteins, particularly those belonging to clade IV, V, and VI, methylating other RNA types cannot be ruled out. Combining miCLIP-MaPseq with the Malc method holds promise for further exploring the presence of $m^2A$ in different RNA types in eukaryotes.

Although the radical-SAM superfamily is one of the largest enzyme groups, this superfamily is relatively scarce in eukaryotes and is limited to a small number of subgroups[48]. RlmN and Cfr are the only two RNA methyltransferases employing the radical-SAM mechanism. The unusual mechanism makes them more consuming than other methyltransferases. Intriguingly, we identified six distinct clades of RlmN-like proteins in eukaryotes, particularly within the Diaphoretickes group. Four of these clades show clear phylogenic relationships with prokaryotic counterparts from various taxa. However, the origin of clade I, which exhibits the highest similarity to RlmN in blue-green algae, can be explained through plastid symbiogenesis. The horizontal gene transfer (HGT) events giving rise to RlmN-like proteins in eukaryotes likely occurred several times independently, indicating the universal benefits of RlmN and/or $m^2A$ gaining. Although deficiency of RlmN-like proteins in Arabidopsis only results in limited phenotypic effects under certain conditions, the underlying physiological roles of $m^2A$, potentially including its metabolites, remain to be investigated.

We demonstrated that $m^2A37$ modification in tRNA enhances protein translation efficiency on the genes with tandem $m^2A37$-tRNA-dependent codons. However, removal of $m^2A$ from tRNAs in *rlmnl2* or *rlmnl3* did not affect the growth and development of *Arabidopsis*. A similar result was also observed in *rlmnl1* mutant, lacking $m^2A2521$ in chloroplast 23S rRNA, although the function of $m^2A$ in rRNA controls translational accuracy[24]. Disruption of *RLMNL1*, *RLMNL2*, or *RLMNL3* only leads to hypersensitivity to aminoglycoside antibiotics, paromomycin and G418, along with a slight enhancement of osmotic stress resistance (Supplementary Fig. 24). Driven by curiosity as to why the loss of $m^2A$ in tRNA and rRNA did not affect plant growth and development, we checked the published results for other tRNA and rRNA modifications. Three methylation modifications ($m^1G$, $m^2G$, and $m^7G$) are well-conserved tRNA modifications, but only the loss of $m^2G$ exhibited an early-flowering phenotype[49]. One interesting observation is that $m^2G$ modification is present at multiple positions of tRNA. Depletion of 2'-*O*-methylation modifications from position 32 and 34 of tRNA suppresses resistance to *Pseudomonas syringae* DC3000 in Arabidopsis[50]. These results indicate that RNA modification at single site of tRNA might not be necessary for plant growth and development. The rRNA modification $m^{6,6}_2A$ affects the translation of a subset of proteins and is involved in antibiotic resistance[51,52]. However, the removal of $m^{6,6}_2A$ from 18S rRNA of Arabidopsis mitochondria by depletion of its writer Dim1B was not reported to affect plant growth and development[53]. Thus, we speculate that the loss of a single modification on tRNA and rRNA usually does not lead to severe defects in plant growth and development.

In summary, our study has demonstrated the significance of $m^2A37$ in maintaining a relaxed conformation of tRNA and facilitating protein translation in both chloroplast and cytosol through the decoding of tandem $m^2A$-tRNA-dependent codons. We have also confirmed the widespread presence of $m^2A$ modification in green plants. Furthermore, our investigation of RlmN homologs has expanded our understanding of their function and phylogenetic relationships, providing insights into the processes of horizontal gene transfer and evolution. Overall, this study uncovers a layer of translation regulation in plants mediated by $m^2A$ modifications and highlights the importance of studying RlmN homologs in various biological contexts.

## Methods

### Plant materials and growth conditions

All *Arabidopsis thaliana* genotypes used in this study were of the Columbia-0 (Col-0) ecotype. Three T-DNA insertion mutant lines: *rlmnl1* (Salk_022971c), *rlmnl2* (Salk_101658c), and *rlmnl3* (Salk_112040c) were obtained from the ABRC, and were confirmed as homozygotes (Supplementary Fig. 8a). Complementation lines

(*RLMNL1:RLMNL1/rlmnl1*, *RLMNL1:RLMNL1m/rlmnl1*, *RLMNL2:RLMNL2/ rlmnl2*, *RLMNL2:RLMNL2m /rlmnl2*, *RLMNL3:RLMNL3/rlmnl3*, and *RLMNL3:RLMNL3m/rlmnl3*) were obtained by Agrobacterium tumefaciens mediated transformation. All *Arabidopsis thaliana* were grown at 22 °C under long-day conditions (16 h light/8 h dark, white fluorescent tubes, 150–200 μmol m$^{-2}$ s$^{-1}$). Seedlings were collected from 14-day-old plants grown on half-strength MS nutrient agar plates (Phyto-Technology Laboratories). Rosette leaves and cauline leaves were harvested at flowering time. Inflorescence samples were collected randomly from 6-week-old plants and divided into three biological replicates. Except where otherwise indicated, 14-day-old Arabidopsis seedlings were used in experiments.

For other plant species, *Oryza sativa* (japonica group) cv. Nipponbare was cultivated in Hoagland solution at 30 °C under a 12 h light/12 h dark cycle. Shoots of 15-day-old hydroponic seedlings were used for RNA extraction. Spinach was purchased from the local market. For surface sterilization, the material was first dipped in 70% alcohol for 30 s, then in 0.5% SDS for 5 min. After that, the material was washed 5 times with sterilized distilled water and wiped dry. Leaf edge was used for DNA and RNA extraction. Two pairs of species-specific primers on mitochondrial gene CytB were used to verify spinach material (Supplementary Fig. 26 and Supplementary Table 1). *Physcomitrella patens* Gransden (gifts from Prof. Haodong Chen's lab at Tsinghua University) was cultivated on BCD medium with 1% sucrose at 22 °C under long-day conditions (16 h light/8 h dark, white fluorescent tubes, 100 μmol m$^{-2}$ s$^{-1}$), and whole plants were used for RNA extraction. *C. reinhardtii* (FACHB-265) was obtained from Freshwater Algae Culture Collection at the Institute of Hydrobiology (FACHB, Wuhan, China) and cultivated in SE medium (Bristol's solution with A5 trace metal and soil extract) at 25 °C under a 12 h light/12 h dark cycle. Algal cells were harvested at the exponential phase for RNA extraction. All plant materials were frozen in liquid nitrogen immediately after harvest and stored at −80 °C.

## Phenotypic analysis
Seeds of all genotypes are harvested from plants grown under the same condition at the same time. To measure the germination rate under different conditions, 180 seeds of each genotype (divided into 3 groups for replicates) were prepared for each treatment. The seeds were surface-sterilized by 70% alcohol for 1 min then 50% commercial bleach solution for 25 min and rinsed 5 times with sterile water. After 3 days of incubation at 4 °C, the seeds were dispersed onto half-strength MS nutrient agar plates with 1 μg/ml G418, 1 μg/ml paromomycin, 125 mM NaCl, or with no additive (mock) and grown at 22 °C under long-day conditions (16 h light/8 h dark, white fluorescent tubes, 150 to 200 μmol m$^{-2}$ s$^{-1}$). The germination rates under antibiotics treatment are measured on the third day, and the germination rates under salt stress are measured on the third, fifth, and seventh days. To measure the root length under antibiotics treatment, seedlings were grown under the same condition except placed vertically. The root length was measured on the fifth day by ImageJ software (Photographs were presented in the Source Data file). To examine the resistance to mannitol-induced osmotic stress of plants of different genotypes, 5-day-old seedlings were transferred to half-strength MS nutrient agar plates with 240 mM mannitol, and the root length was measured on the seventh day after transferring by ImageJ software (Photographs were presented in the Source Data file). To examine the resistance to heat shock of plants of different genotypes, plates with 7-day-old seedlings were placed to a 45 °C light chamber for 30 min and then recovered at 22 °C conditions. The survival rate was monitored on the third, fourth, and fifth days after treatment.

## Cell line
HeLa cells (ATCC CCL-2) were obtained from ATCC and cultivated at 37 °C in DMEM with 10% fetal bovine serum and 1% penicillin/streptomycin in 5% $CO_2$.

## Plasmid construction
The full-length cDNAs of *RLMNL1*, *RLMNL2*, and *RLMNL3* were obtained from the ABRC. pCAMBIA1307 containing a 2 × 35S promoter, an N-terminal 6 × Myc-tag, a C-terminal FLAG-tag, and CaMV poly-A signal at the multiple cloning sites were used for plasmid construction. Full-length *RLMNL1-3* cDNAs were cloned between the EcoRI and BamHI restriction sites of pCAMBIA1307 to generate 35 S:6 × Myc-RLMNL1-3-Flag, respectively. The catalytically inactive mutant variants RLMNL1 C398A (RLMNL1m), RLMNL2 C430A (RLMNL2m), and RLMNL3 C345A (RLMNL3m) were generated using the QuikChange II XL site-directed mutagenesis kit (Agilent), constructing the plasmids of 35 S:6 × Myc-RLMNL1m-3m-Flag. The 2 × 35S promoter sequence and CaMV poly-A signal in the six plasmids of 35 S:6 × Myc-RLMNL1-3-Flag and 35 S:6 × Myc-RLMNL1m-3m-Flag were replaced with the native promoter (2000 bp upstream of the start codon) and the native terminator (400 bp downstream of stop codon) of *RLMNL1-3*, respectively, generating the plasmids (*RLMNL1-3:RLMNL1-3* and *RLMNL1-3:RLMNL1m-3m*) for complementation lines. All primers were ordered from RuiBiotech and are listed in Supplementary Table 1. Restriction endonucleases and DNA polymerases were ordered from New England Biolabs.

## Plant transformation
*Agrobacterium tumefaciens* strain GV3101 was used for plant transformation by the floral dip method. T0 seeds were screened for Hygromycin B resistance, and T1 plants were confirmed by PCR analysis of genomic DNA. Complementation lines, including *RLMNL1:RLMNL1/rlmnl1*, *RLMNL1:RLMNL1m/rlmnl1*, *RLMNL2:RLMNL2/ rlmnl2*, *RLMNL2:RLMNL2m/rlmnl2*, *RLMNL3:RLMNL3/rlmnl3*, and *RLMNL3:RLMNL3m/rlmnl3* were obtained by transforming the *rlmnl1*, *rlmnl2*, and *rlmnl3* mutant plants with the plasmids of *RLMNL1:RLMNL1/rlmnl1*, *RLMNL1:RLMNL1m/rlmnl1*, *RLMNL2:RLMNL2/ rlmnl2*, *RLMNL2:RLMNL2m /rlmnl2*, *RLMNL3:RLMNL3/rlmnl3*, and *RLMNL3:RLMNL3m/rlmnl3*, respectively. Overexpression lines *35S: RLMNL1-3* were generated by transforming the 35 S:6 × Myc-RLMNL1-3-Flag into wild-type Col-0 plants.

## Gene expression analysis by RT-qPCR
First-strand cDNA was synthesized from 1 μg of DNase I-treated total RNA using Reverse Transcriptase M-MLV (TaKaRa). RT-qPCR was performed using UltraSYBR Mixture with ROX (CWBIO) on a ViiA 7 Dx Real-time PCR instrument (Applied Biosystems). All samples were analyzed in triplicate. *ACTIN2* (AT3G18780) and *TUBULIN4* (AT5G44340) were used as reference genes for measuring the expression of *RLMNL1-3*. The relative expression levels were calculated via $2^{-\triangle\triangle CT}$. All primers used were listed in Supplementary Table 1.

## The isolation of total RNA, mRNA, total tRNA, and specific tRNA species
Total RNA was isolated from ground plant tissues with TRIzol reagent (Thermo Fisher Scientific). Poly(A) RNA was isolated from total RNA using oligo(dT)$_{25}$ Dynabeads (Thermo Fisher Scientific). Removal of contaminating rRNA was achieved using the Ribominus Plant Kit (Thermo Fisher Scientific). The mRNA was qualified using a 2100 Bioanalyzer (Agilent) ensuring no degradation or rRNA/tRNA contamination. RNA species smaller than 200 nucleotides were extracted using mirVana miRNA Isolation Kit (Thermo Fisher Scientific). The small RNAs were further loaded onto a 15% TBE-Urea gel and the total tRNA bands were sliced and recovered from the gel. Specific tRNAs were isolated using biotinylated probes (Supplementary Table 1). Briefly, 10 μl Dynabeads MyOne Streptavidin C1 (Thermo Fisher Scientific) were generated according to the manufacturer's instructions, washed once with buffer A (10 mM Tris-HCl pH 7.5, 2 mM EDTA, 2 M NaCl), and finally resuspended in 10 μl of 3 × SSC. Subsequently, 25 pmol of biotinylated probes were mixed with 50 μg of total tRNA in 50 μl of 3 × SSC and heated for 10 min at 75 °C. Thereafter, the mixture

was gently rotated at room temperature for 1 h to allow the annealing of the tRNAs to specific probes. Prepared Dynabeads were then mixed and the suspension was rotated for another 30 min. The RNA-coated Dynabeads were washed three times with 3× SSC, twice with 1× SSC, and several times with 0.1× SSC until the absorbance of the wash solution at 260 nm was close to zero. The specific tRNA retained on the beads was eluted three times using RNase-free water.

## Quantitative analysis of RNA modification by LC-MS/MS

For analysis of internal RNA modifications, 50–100 ng RNA was digested with 1 unit of Nuclease P1 (Wako) in 50 μL buffer containing 10 mM ammonium acetate (pH 5.3) at 60 °C for 2 h and then at 42 °C overnight, followed by the addition of 5.5 μL of 1 M fresh $NH_4HCO_3$ and 1 unit of shrimp alkaline phosphatase (rSAP, New England Biolab). The mixture was incubated at 37 °C for an additional 3 h. Digested samples were filtered through 0.22-mm syringe filters before UPLC-MS/MS analysis. The nucleosides were separated by UPLC (Shimadzu) equipped with a ZORBAX SB-Aq column (Agilent) and detected by MS/MS using a Triple Quad 5500 (AB SCIEX) mass spectrometer in positive ion mode by multiple reaction monitoring. The MS parameters were optimized for $m^2A$ detection. Nucleosides were quantified using the nucleoside-to-base ion mass transitions of $m/z$ 268.0 to 136.0 (A), $m/z$ 282.0 to 150.0 ($m^6A$), $m/z$ 282.0 to 150.0 ($m^1A$), $m/z$ 282.0 to 150.0 ($m^2A$), $m/z$ 282.0 to 150.0 ($m^8A$), $m/z$ 244.0 to 112.0 (C), and $m/z$ 284.0 to 152.0 (G). Standard curves were generated by running a concentration series of pure commercial nucleosides ($m^2A$ and $m^8A$ from Carbosynth while others from Sigma-Aldrich). Concentrations of nucleosides and the $m^2A/A$ ratio in samples were calculated by fitting the signal intensities to the standard curves.

## Mung bean nuclease assisted LC–MS/MS quantification ("Malc")

Dynabeads were prepared as described in specific tRNA species isolation. 25 pmol of biotinylated probe36, probe37, and probe38 were mixed with 50 μg of total tRNA in 50 μl of annealing buffer (10 mM HEPES pH 6.0, 500 mM KCl, 0.5% DMSO) separately. Subsequently, the mixture was heated for 10 min at 75 °C and gently rotated at room temperature for 1 h to allow the annealing of the tRNAs to specific probes. Thereafter, 40 unit Mung bean nuclease (NEB), 5 unit RNase I (Thermo Fisher Scientific), and 0.25 μg of RNase A (Thermo Fisher Scientific) were added. The mixture volume was adjusted to 100 μl with a final concentration of 1× Mung bean nuclease buffer (Zinc ion required), 10 mM HEPES pH 6.0, 500 mM KCl, 0.5% DMSO, and incubated at 30 °C for 30 min. The incubation time and KCl concentration may be adapted for different tRNA species. A final concentration of 0.01% SDS was then added to stop the reaction. Prepared Dynabeads were then mixed and the suspension was rotated for another 30 min at room temperature. The RNA-coated Dynabeads were washed three times with 3× SSC, twice with 1× SSC, and three times with 0.1× SSC. The specific tRNA fragments retained on the beads were eluted three times using RNase-free water for LC-MS/MS quantification. The exact modification ratio of $m^2A$ ($R_m$) was calculated through equation (1):

$$R_m = \frac{AR}{1+R} \qquad (1)$$

In equation (1), A stands for the number of adenosine in Fx-1, and R stands for the $m^2A/A$ ratio measured by LC-MS/MS. Note that the stoichiometric modification $m^1A58$ of cytosolic tRNA should be excluded when counting the number of adenosine as it is included in Fx-1.

## RNA gel blot analysis

Five micrograms of total RNA isolated from 14-day-old seedlings of Col-0, *rlmnl2*, and *rlmnl3* plants were run in a 15% TBE-urea gel for

electrophoresis separation. After electrophoresis, RNA was transferred to the Hybond-N⁺ membrane (GE Healthcare) and immobilized by 150 mJ UV light (wavelength 254 nm). The membrane was prehybridized with ULTRAhyb-Oligo hybridization buffer (Thermo Fisher Scientific) for 4 h before overnight incubation with 500 pmol of biotinylated tRNA detection probes. The membrane was washed five times with washing buffer (2× SSC, 0.5% SDS) and then incubated with HRP-labeled streptavidin (Thermo Fisher Scientific) for 1 h. The membrane was then washed three times with washing buffer and signals were visualized using a 5200 chemiluminescence imaging system (Tanon).

## Analysis of tRNA conformation by native polyacrylamide gel electrophoresis

Considering that magnesium ion is necessary for RNA folding, we first screened different $Mg^{2+}$ concentrations for the most discriminative condition. Five micrograms total RNA isolated from 14-day-old seedlings of wild-type or mutant plants were incubated at room temperature for 30 min in folding buffer with a final concentration of 34 mM Tris, 66 mM HEPES, and various concentrations of $MgCl_2$ ranging from 0 mM to 24 mM. 15% Tris-HEPES polyacrylamide gel was pre-run in running buffer (34 mM Tris, 66 mM HEPES, 6 mM $MgCl_2$, 0.1 mM EDTA) at 15 W for 30 min. The samples were then mixed with loading buffer (34 mM Tris, 66 mM HEPES, 20% glycerol, 0.02% xylene cyanol, and various concentrations of $MgCl_2$ the same with folding buffer) and loaded to pre-run gel immediately. The gel was then run at 15 W for 2 h in the ice bath. After electrophoresis, the blot assay was then performed as mentioned above. The $Mg^{2+}$ concentration screen results were shown in Supplementary Fig. 24. For chloroplast tRNA$^{Ser}_{GGA}$ and elongator tRNA$^{Met}_{CAU}$, 6 mM $MgCl_2$ was chosen for final migration assays. For other tRNA species, 3 mM $MgCl_2$ was chosen.

## Analysis of tRNA melting temperature

Individual tRNAs were isolated from Col-0 or mutant plants as described above. 10 pmol tRNA were incubated at 25 °C for 1 h away from light in folding buffer with a final concentration of 17 mM Tris, 33 mM HEPES, 5 mM $MgCl_2$, and 1× SYBR Green (Invitrogen). The melting curve was measured on ViiA 7 Dx (Applied Biosystem) using the following gradients: 25 °C for 5 min, 25–85 °C at 0.01 °C s-1, and 85 °C for 5 min. FFT filter was applied to smooth the melting curves in OriginPro 2019.

## Aminoacyl-tRNA analysis

Total RNA was extracted under the acidic condition and dissolved in 10 mM NaAc (pH 5.0). Half of the isolated RNA was deacylated by incubated in 0.1 M Tris-HCl (pH 9.0) at 60 °C for 10 min, then neutralized, precipitated, and re-dissolved in 10 mM NaAc (pH 5.0). Five micrograms of untreated or treated RNA was then run through 15% urea PAGE in acidic buffer at 15 W for at least 3 h. After electrophoresis, the blot assay was then performed as mentioned in the RNA gel blot analysis part.

## Protein immunoblotting analysis

Protein samples were run in a 12% SDS-PAGE for electrophoresis separation and then transferred to a PVDF membrane. The membrane was blocked with TBST (20 mM Tris-HCl pH 7.4, 150 mM NaCl, and 0.1% Tween-20) containing 5% skimmed milk powder and incubated with the specific antibody overnight at 4 °C, followed by incubation with anti-rabbit IgG HRP-linked antibody (Cell Signaling Technology). The membrane was then washed by TBST, and signals were visualized using a 5200 chemiluminescence imaging system (Tanon). After visualizing, the membrane was stripped and incubated with the specific antibody for another protein or stained with Ponceau S as user-manual described for loading control. The specific antibodies used in this study were anti-puromycin mouse monoclonal antibody (Millipore, MABE343), anti-NDHH rabbit polyclonal antibody (PhytoAB, PHY2292S), anti-

PSBA rabbit monoclonal antibody (Abcam, ab65579), anti-RPOC2 rabbit polyclonal antibody (PhytoAB, PHY0693A), and anti-actin rabbit polyclonal antibody (Abcam, ab197345). Protein bands were quantified by ImageJ software (https://imagej.net/).

## Protoplast transient expression

Well expanded leaves from 3-4 weeks old plants before flowering were used for protoplast isolation. 10 leaves were cut into 0.5–1 mm strips with fresh razor blades without wounding and mixed with 10 ml cellulase/macerozyme solution (1.5% cellulase R10 (Yakult), 0.4% macerozyme R10 (Yakult), 0.4 M mannitol, 20 mM KCl, 20 mM MES pH 5.7, 10 mM CaCl$_2$, 5 mM β-mercaptoethanol) in a 10-cm petri dish followed by 15 min of vacuum infiltration. The suspension was gently shaken for 3 h, filtered with a 70 μm nylon mesh, and centrifuged at 100 × $g$ for 2 min to pellet the protoplasts. The protoplasts were washed once in W5 solution (154 mM NaCl, 125 mM CaCl$_2$, 5 mM KCl, 2 mM MES pH 5.7) and resuspended in MMg solution (0.4 M mannitol, 15 mM MgCl$_2$, 4 mM MES pH 5.7) before PEG transfection. PEG Transfection was carried out at 23 °C. One hundred microliters protoplasts were first added to 10 μl DNA (10–20 μg of plasmid DNA), then 110 μl of PEG/Ca solution (4 g PEG4000, 3 ml H$_2$O, 2.5 ml 0.8 M mannitol, 1 ml 1 M CaCl$_2$) were added and gently mixed. The mixture was incubated at 23 °C for 15 min, diluted with 0.44 ml W5 solution and mixed well gently. The protoplasts were then harvested via centrifuged at 100 × $g$ for 2 min, resuspended in 1 ml W5 solution, and placed in 6-well plates for overnight protein expression.

## Translation reporter assay

pGreenII0800 vector which contains a firefly luciferase region (F-luc) and a *Renilla* luciferase region (R-luc) was used for constructing the translation reporter plasmids. The reporter plasmid was obtained by inserting 35 S promoter together with 6 × CGT, 6 × CGC, 6 × CGG, 6 × CGA, 6 × CAA, or 6 × CAG before the F-luc coding region. Ten micrograms of plasmid was transfected into protoplasts of Col-0, *rlmnl3*, *RLMNL3:RLMNL3/rlmnl3*, and *RLMNL3:RLMNL3m/rlmnl3* in 6-well plates. After 16 h, protoplasts in 6-well plates were proceeded to determine luciferase activity by Dual-Luciferase Assay Kit (Vigorous). R-luc was used to normalize F-luc activity to evaluate the translation efficiency of reporters.

## Puromycin labelling assay

One gram 10-day-old Arabidopsis seedlings were transferred to MS liquid medium containing 25 μg/ml puromycin, vacuum infiltrated, and incubated for 2 h at the growing condition for chloroplast nascent protein labelling. The Arabidopsis samples were dried and placed in a mortar, grounded on ice with 10 ml of plastid extraction buffer A (300 mM sucrose, 10 mM KCl, 1 mM MgCl$_2$, 1 mM EDTA, 5 mM DTT, freshly add 1:100 protease inhibitor cocktail) added. The samples were filtered with Miracloth and the residue was discarded. After centrifuging at 800 × $g$ for 2 min, the supernatant was transferred to a clean tube. Spin at 2000 × $g$ for 1 min, and discard the supernatant. Plastid extraction buffer B (300 mM sucrose, 10 mM KCl, 1 mM MgCl$_2$, 1 mM EDTA) was used to wash the precipitated chloroplasts several times, and the white precipitates (nuclei) were discarded. For cytosolic nascent protein labelling, 1 g 10-day-old Arabidopsis seedlings were transferred to MS liquid medium containing 1 μg/ml puromycin, vacuum infiltrated, and incubated for exact 30 min at growing condition. The Arabidopsis samples were wiped dried with bibulous paper, frozen in liquid nitrogen, and ground into fine powders. Equal amounts of extracted total protein or chloroplast protein for each sample were resolved by SDS-PAGE, electrotransferred to PVDF membrane, and immunoblotted by the anti-puromycin antibody. Ponceau S staining was employed to ensure the equal loading of proteins.

## Polysome profiling and ribosome footprinting sequencing

One 15-cm plate of 10-day-old Col-0 and *rlmnl3* seedlings were prepared for each sample. 4 ml lysis buffer (10 mM Tris pH 7.4, 150 mM KCl, 5 mM MgCl$_2$, 100 μg/ml cycloheximide (CHX), 0.5% Triton-X-100, freshly add 1:100 protease inhibitor cocktail (Roche), 40 U/ml SUPERasin (Thermo Scientific)) was added to the seedling powder and then kept on ice for 15 min with occasional rotating. After centrifugation at 15,000 × $g$ for 10 min, the supernatant was collected. Eight microliters of Turbo DNase (Thermo Scientific) was added to the lysate. For polysome profiling, the lysate was directly used for sucrose gradient centrifuging. For the ribosome footprinting assay, 20% of the lysate was saved as the input portion (saved for RNA-seq to enable quantification of translational efficiency). Four microliters SUPERasin was added to the input portion and 40 μl MNase buffer and 3 μl MNase (6000 gel units, NEB) were added to the remaining sample portion. Both portions were kept at room temperature for 15 min, and then 8 μl SUPERasin was added to the sample portion to stop the reaction. The input portion was mixed with 5 ml TRIzol to further RNA extraction, mRNA purification, and fragmentation (-100 nt). The sample portion was used for ribosome footprinting following the published method[14]. Simply, a 10/50% w/v sucrose gradient was prepared in a lysis buffer without Triton-X-100. The lysate was loaded onto the sucrose gradient and centrifuged at 4 °C for 4 h at 134,500 × $g$. (Beckman, rotor SW40Ti). The sample was then fractioned and analyzed by Gradient Station (BioComp). The fractions corresponding to the 80S monosome were collected and mixed with an equal volume of TRIzol to purify RNA. The RNA was loaded and separated on a 15% TBE-urea gel for isolating the RNA between 21-nt and 42-nt. The purified RNA fraction was subjected to library construction. The RNAs from both portions were end-repaired by T4 PNK (NEB) before the library construction. The sequencing library was constructed using NEBNext Small RNA Library Prep Kit (NEB). Sequencing was performed on Illumina HiSeq 2500.

## Sequencing data analysis

Adaptor sequences were trimmed and low-quality sequences were discarded using cutadapt (v1.18)[54]. Sequencing reads were aligned to the reference genome (TAIR10) using HISAT2. The mapping results were summarized in Supplementary Dataset 1. FPKM of input and ribosome footprinting samples were calculated by Cufflinks using default parameters. Ribosome protected fragment (RPF) of a gene is defined as FPKM of ribosome footprinting samples adjusted by its reads abundance on CDS. The comparison of independent replicates was summarized in Supplementary Fig. 18. Translation efficiency (TE) of each replicate was defined as the ratio of RPF and corresponding input FPKM. TE$_{wild-type}$ and TE$_{rlmnl3}$ were defined as the geometric mean of the translation efficiency of two biological replicates. Alternative expressed genes were determined via three criteria: (1) with more than twofold change in FPKM between wild-type and *rlmnl3*; (2) adjusted $p$-value < 0.05; (3) $SNR > 1$; $SNR$ (signal-noise-ratio, calculated through equation (2)) was used to filter the genes with unacceptable varied FPKMs in two biological replicates.

$$\text{SNR} = \frac{|X - Y|}{|x_1 - x_2| + |y_1 - y_2|} \tag{2}$$

In equation (2), $X$ and $Y$ stand for the averaged FPKM of wild-type and *rlmnl3*; $x_1$, $x_2$, and $y_1$, $y_2$ stand for the FPKM of two replicates of wild-type and *rlmnl3*, respectively.

## Statistical Information

Student's *t*-test in Fig. 5c, Pearson correlation coefficient calculation in Supplementary Fig. 18, and One-way ANOVA followed by post hoc tests in Fig. 4, Fig. 6, Supplementary Fig. 9, Supplementary Fig. 16, Supplementary Fig. 22, and Supplementary Fig. 24 were carried out on

OriginPro 2019. Nonparametric tests (Mann–Whitney test or Kruskal–Wallis test) in Fig. 7, Supplementary Fig. 21, and Supplementary Fig. 23 were carried out on IBM SPSS Statistics 26. The sample size and determined statistical significance were mentioned in the figure legends. The original data, parameters of statistical analysis, and exact *p*-values are provided in Source Data

## Accession numbers
Sequence data for the genes in this study can be found in The Arabidopsis Information Resource (www.arabidopsis.org) under the following accession numbers: RLMNL1, At2g39670; RLMNL2, At1g60230; RLMNL3, At3g19630.

## Reporting summary
Further information on research design is available in the Nature Portfolio Reporting Summary linked to this article.

## Data availability
Sequencing data have been deposited into the Gene Expression Omnibus (GEO) under the accession code GSE127146. Source data are provided with this paper.

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

## Acknowledgements

This work was supported by the National Natural Science Foundation of China (nos. 22225704, 21820102008, and 92053109), the National Basic Research Program of China (2019YFA0802201), and China National Postdoctoral Program for Innovative Talent (no. BX20180007 for H.-C. D.). We thank Arabidopsis Biological Resource Center (ABRC) for RLMNL1-3 cDNA and *rlmnl1, rlmnl2,* and *rlmnl3* seeds.

## Author contributions

G.J. and H.-C.D. conceived the project and designed the experiments; H.-C.D. and C.Z. performed the experiments with the help of P.S., J.Y. and Y.W.; G.J., H.-C.D. and C.Z. wrote the manuscript.

## Competing interests

The authors declare no competing interests.
