## [Peer Review File · Nature Communications]

C2-methyladenosine in tRNA promotes protein translation by facilitating the decoding of tandem m2A-tRNA-dependent codonsReviewer #1 (Remarks to the Author):

In the manuscript "C2-methyladenosine is a plant specific RNA modification in eukaryotes facilitating protein translation" the authors identify the presence of C2-methyladenosine (m²C) in the Arabidopsis transcriptome. Specifically, they find that this modification is added to chloroplast rRNA, 6 chloroplast tRNAs, and 4 cytoplasmic tRNAs. They then go on and identify the three RNA methyltransferases responsible for this methylation, which are the plant orthologs of the enzyme that adds this modification in bacterial transcriptomes (RlmN), so they name these three enzymes RLMN-LIKE1-3 (RLMNL1-3). The authors find that RLMNL1, RLMNL2, and RLMNL3 add m²C to chloroplast rRNA, chloroplast tRNAs, and cytoplasmic tRNAs, respectively. Finally, the group finds that knocking out the function of RLMNL2 (rlmnl2 mutant plants) and RLMNL3 (rlmnl3 mutant plants) results in decreased translation output in the chloroplast and cytoplasm respectively. More specifically, the translation of transcripts encoding proteins involved in basic nucleic acid processes seem to be those whose translation is most affected in the absence of RLMNL3 function. However, the importance of this finding was not further explored or described. This is an interesting and timely study that is well done and mostly comprehensive. However, I have identified a few necessary experimental additions and manuscript edits that I have outlined below that need to be added to a revised version of the manuscript.

Necessary experimental revisions:

- 1) The protein blots done in Figure 4a with chloroplast protein lysates need to be redone and these samples from rlmnl3 mutant plants also need to be added. We need to know that RLMNL2 is truly chloroplast specific and RLMNL3 is not functioning in the chloroplast.
- 2) Relatedly, the cytoplasmic protein blots in Figure 4b should contain samples from rlmnl1 and rlmnl2 mutant plants to demonstrate that these two methyltransferases are chloroplast specific while RLMNL3 is the one that functions in the cytoplasm.
- 3) Additionally, the experiments presented in Figure 4e need to also have samples from rlmnl3 added to once again compare and contrast the effects of the three different methyltransferases. Related to this figure better discussion should be added to the manuscript to describe why PSBA shows a protein reduction in rlmnl2 mutants while PSAF does not. To me, it looks like chloroplasts in rlmnl2 mutant plants are upregulating the transcript of PSAF to increase the needed protein output for this locus. Maybe the authors should do a ratio of protein signal to RNA signal to demonstrate that this type of gene regulation is occurring due to the loss of m²C of tRNAs in the rlmnl2 mutant plants
- 4) The experiments presented in Figures 4c and 4d that were performed for RLMNL3, also need to be done for the codons of the tRNAs that are methylated by RLMNL2. The readers need to see that m²C is necessary for proper translation efficiency for both chloroplast and nuclear-encoded transcripts.

Necessary text revisions:

- 1) I don't like the use of plant-specific in reference to this modification or the enzymes that add it to plant transcripts since it also occurs in bacteria and likely other organisms. I think plant-specific should be removed from this manuscript, especially from the title of the manuscript.
- 2) There are numerous grammatical errors throughout the manuscript. Please clean these up in a revised manuscript version.
- 3) Please discuss the implications and importance of translation efficiency (TE) findings that are presented in Figure 5. Furthermore, the GO analysis that accompanies the TE analyses is very briefly described but some discussion of these findings needs to be added to both Results and Discussion to describe the importance and what the implications of these findings are for the m²C field.

Reviewer #2 (Remarks to the Author):

This manuscript describes the detection and superficial analysis of m²A modifications in eukaryotes, in particular in plants. LCMS was used to quantify the m²A content in isolated RNA species, but not further mapping or experimental assignment of positions of m²A modifications was performed. A translation phenotype in Ko lines was characterized on the molecular level. The

overall results of the characterization of ko lines appear arbitrary and nondescript as might result from disruption generic gene peripherally involved in translation. The GO results do not promote a hypothesis for a molecular mechanism. The fact that a ko affects translation is trivial and very much expected when looking at these classes of RNA; again there are several components missing to construct even the beginnings of a molecular mechanism of some novelty. Often, statements on results lack tangible content. For example, the statement that „the eukaryotic system is more complex than bacteria“ (line 301) is naïve. First, this is true for essential all types of biomolecules, including DNA, Proteins, RNA and every single type of RNA modification. Second, chloroplast and mitos are of bacterial origin, and together with the cytosolic system it is perfectly normal to have multiple homologues of an enzyme to address different cellular compartments. Indeed this is the observed rule, so RLMNL is just another example of a standard case, rather than anything extraordinary.

In addition, the authors do not really have any novel sites to report; m2A appears in tRNA and rRNA, which was known before. The authors are not capable of precisely defining the position of m2A modification within tRNA or rRNA by themselves, instead they infer sites by homology from literature on other organisms.

Beyond this, the study raises, but then leaves unanswered, some questions that might eventually become interesting. Can they confirm by methods other than LCMS, that the tRNAs are only partially modified with m2A at A37? If this were accurate (which was not convincingly shown), then the question would be, if there is another modification at A37 instead, resulting in populations of mixed A37 modifications, which is doubtful. If the remainder of A37 were instead unmodified, should that not cause massive problems with frameshifting, as observed for various ko of A37 modifications? Again, a molecular mechanism is missing.

In summary, the study is largely descriptive, parts of it are immature, and it lacks the kind of conceptual novelty appropriate for this journal. I recommend rejection.

Reviewer #3 (Remarks to the Author):

In this manuscript, a specific type of adenosine methylation (the m2A modification) in different classes of RNA was investigated in Arabidopsis, 10 tRNA genes that harbor these methyl marks in the anticodon loop were identified, the RLMNL enzymes responsible for depositing the m2A modifications were knocked out, and phenotypic consequences of the mutations were examined. The authors provide convincing evidence to suggest that the three RLMNL genes in Arabidopsis play specialized roles, with RLMNL1 modifying chloroplast rRNA, RLMNL2 methylating chloroplast tRNA, and RLMNL3 targeting cytoplasmic tRNA. Loss of function T-DNA mutants (one allele per gene) were isolated, but showed no morphological defects. In contrast, at the molecular level, all three mutants had abnormal m2A profiles, that were complementable by a WT copy of the respective RLMNL gene. Two of the mutants, *rlmnl2* and *rlmnl3*, were concluded to have global defects in translation, but the data provided in support of that claim are not very robust and missing important quality control metrics (see below). With *rlmnl3* having over a thousand genes translationally downregulated and nearly 300 genes upregulated at the translational level, it does not make sense to me that the mutant displays normal plant morphology. Such a drastic and global effect on translation should lead to a sick or at least slow-growing plant. In light of that discrepancy and given that some of the essential quality controls of the Ribo-seq data (see below) have not been provided, I am not at all confident in the validity of that data.

Assuming that the data are true and the RLMNL-mediated m2A modification in rRNAs and tRNAs affects global mRNA translation, the question arises as to what regulates m2A modifications? Are there specific environmental conditions or developmental processes that alter RLMNL levels or activity? Besides looking at the expression of the three RLMNL genes across plant organs, this question was not investigated. Just by looking at publicly available transcriptomic data, I see that some abiotic stress factors appear to down-regulate RLMNLs (e.g., salt and drought repress RLMNL1, and salt and mannitol decrease RLMNL2 levels), whereas heat and UVB induce RLMNL3 during the recovery period. The authors thus missed an obvious opportunity to connect these genes and the respective RNA modifications they confer with specific physiologically relevant biological pathways. It would have made sense to test the phenotypes of the mutants under the stresses that are found to regulate these three genes – again, this is another missed opportunity.

Furthermore, no overexpression studies are reported – do the levels of m2A modifications in rRNA and tRNA increase and does that alter global translational profiles? In the absence of these studies to suggest a clear physiological role for RLMNL-mediated m2A modifications in housekeeping RNAs, my enthusiasm for this work is moderate at best.

My detailed comments are provided below.

Fig 1D – "Homo sapiens" is misspelled

Line 120-126 and fig 1D-E, the math does not make sense. One out of seven copies is not 0.7%, but 14.2%. 0.7% means 7 out of a 1000 (or 1 out of 142). The ratios should not be measured in %. So, the graph labeling in Fig 1d and 1e are clearly misleading and the respective description in the text for the different kinds of tRNA makes no sense. Also, it would be more logical to plot $m2A/(m2A+A)$, i.e. the proportion of modified A's relative to the total (modified and unmodified) amount of A's. These comments also apply to fig 3A, C, D.

Line 132, please, specify what is meant by "high sequence similarity" (i.e., provide numerical values of sequence identities).

Fig 2A, in the legend, provide accession numbers of the proteins used in the phylogenetic analysis

Fig S4, indicate the program used to make protein alignments

Lines 146-148 and fig S5, provide references for the sequence similarity network algorithm used.

Fig 7, in the gene schematics in panel A, please, mark positions of genotyping primers employed in panel B and qRT-PCR primers used in panel C. The genotyping gels shown in panel B do not make sense, as for each sample amplification of the mutant and WT fragment needs to be performed and thus two lanes should be displayed for each DNA sample. I assume only WT fragment reactions are shown, but this is not obvious, as no description of what is being amplified is provided.

Lines 174-176, lack of morphological defects in the *rlmnl* mutants suggests that the m2A modification does not play a physiologically relevant role. This does not agree with the dramatic translational changes reported later in this study.

Lines 179-180, the way this sentence is phrased, it is unclear that the 5% modification in the mutant is not the absolute level but rather 5% of the WT level. Again, the ratios should not be expressed in %.

Fig 3A, given the exonic location of the *rlmnl2* and *rlmnl3* T-DNA mutants, how are residual levels of m2A modifications maintained in the chloroplast and cytoplasmic tRNA (as 0-15% of WT activity are seen)? Is it possible that a small fraction of the cytoplasmic enzyme finds its way into plastids and, vice versa, some of the plastidic enzyme is retained in the cytoplasm? This should be easy to test with the generation of double *rlmnl2 rlmnl3* and triple *rlmnl1 rlmnl2 rlmnl3* mutants.

Lines 189, what do the authors mean by "other related proteins"? Are there additional RLMNL-like genes in the Arabidopsis genome besides those three?

Fig 4B, S10A and lines 235-238, in the absence of a proper loading control, the data are difficult to interpret. Less puromycylation signal in the *rlmnl3* mutant may simply mean less sample was loaded, but the Coomassie Blue-stained loading control is clearly overloaded and impossible to accurately compare across the lanes.

Line 255 and 266, fig 4D should be referenced here, not fig 4C.

Fig 4B, the reduction in the PSBA protein levels in *rlmnl2* is very obvious, but for PSAF, the protein level seems to be elevated in *rlmnl2*, concomitant with an increase in its RNA. If the authors want to make a point that there is a defect in PSAF translation in the mutant, protein quantification

needs to be provided that is performed on multiple independent replicates of the Western blots.

Fig 5 and S11-12, The Ribo-seq data are not thoroughly described. How many million reads were obtained for each library, what percentage corresponded to mRNA, rRNA, and tRNA, etc. and what percentage mapped uniquely to the genome? These are essential parameters that need to be reported. It is also customary for Ribo-seq studies to show alignments of reads around the start and stop codons to demonstrate low read number in the 5'UTR and lack of reads in the 3'UTR and to show read periodicity in the genic ORF. Without those data, it is impossible to evaluate the quality of the obtained datasets.

English language needs to be edited throughout the entire manuscript by a native speaker. For example, in the abstract and introduction the following sentences need to be modified (but this list is not comprehensive):

Line 16, "the study of m2A in eukaryotes have remained unclear" sounds awkward.

Line 51, change "locates" to "is located"

Line 70-71, change "via a similar mechanism to RlmN catalyzing" to "via a mechanism similar to that catalyzed by RlmN"

Line 71, change "RlmN (or Cfr), which affecting the susceptibility to multi-kinds of antibiotics" to "RlmN (or Cfr), which affects the susceptibility to a variety of antibiotics"

Line 76, change "tRNA contain m2A modification" to "tRNA contain an m2A modification"

Line 77-78, what did the authors mean by "quantitative m2A levels", quantifiable or variable? All "levels" by definition are "quantitative".

Line 80, remove "in" from "in consistent"

Anna Stepanova

Reviewer #4 (Remarks to the Author):

In this study, authors investigate the function of C2 methylation of adenosine (m2A), an unusual and poorly understood RNA modification, in Arabidopsis RNA. While RlmN, the enzyme that catalyzes m2A formation, has been investigated in bacteria, and is known to be present in plants, prior to this work it was unknown if plant homologs of RlmN also catalyze formation of m2A modification, and if they do what RNA might be modified and what the function of the introduced modification is. In this work, authors combine plant genetics with quantitative analysis of RNA modification, dual luciferase reporters, and ribosome profiling to determine that chloroplast rRNA as well as a subset of chloroplast and cytoplasmic tRNAs contain m2A modification. They further show that three RlmN-like proteins catalyze these methylations, each with specialized set of substrates: chloroplast rRNA (RLMNL1) or tRNA (RLMNL2) and cytoplasmic tRNA (RLMNL3). This unprecedented separation of function provided by three RlmN variants allowed authors to determine that the modification's major function in Arabidopsis is to facilitate translation, as disruption of RLMNL2 gene decreases protein synthesis of chloroplast encoded proteins. Furthermore, using ribosome footprinting, they demonstrated that loss of function of RLMNL3 gene reduces global translational efficiency of a number of protein coding genes in cytoplasm. The study is overall very well executed and resulting findings are highly significant for several reasons. While RlmN of E. coli has been implicated in regulation of translation, this phenotype could not be unambiguously assigned to specific RNA substrates as E. coli RlmN is a multifunctional enzyme that modifies both 23S rRNA and a subset of tRNAs. Separation of function afforded by Arabidopsis enzymes allows separate interrogation of these methylation events. Moreover, combination of ribosome profiling with genetic manipulation is particularly powerful method to interrogate translation on a global scale, and has allowed the authors to conclude that it is tRNA m2A modification that is responsible for facilitating translational efficiency.

There are few important questions that need to be addressed:

1. Site of tRNA modification. While likely evolutionarily conserved, site of modification is not unambiguously assigned. The assumption made here is that A37, like in RNA, is the site of modification. Is there a direct evidence that authors can provide, perhaps through partial RNA digestion and fragment analysis? Additionally what are negative control tRNA that were used in Suppl Figure 3?

2. Dual Luciferase reporter system needs additional controls. Have non m2A modified tRNA, ideally encoding same amino acids, been used in this assay and are any changes in luc expression observed? This is essential to eliminate possible sequence dependent translational changes unrelated to tRNA modification (for example, as described here:

<https://www.biorxiv.org/content/10.1101/571059v2.full>).

3. Additional clarifications are needed regarding Fig 4 and Suppl Fig 10. First, the sample size of two randomly selected proteins is rather low. How many biological replicates were analyzed? PSAF transcript seems 3x increased, couldn't that cause increase in protein level? What do authors hypothesize is the reason that some genes, but not others are affected by the absence on m2A in tRNAs? Is there any information that can be derived from the nature of codons? Any ways to test, perhaps similar to luciferase assay they used?

4. Discussion, lines 310-312: Disruption of RLMNL2 significantly decreases the protein synthesis of PSBA and PSAF. This does not appear to be the case for PSAF.

5. Discussion, lines 321-322: this effect should not be attributed to 23S rRNA (event though that explanation has been suggested) as E. coli RlmN is a multifunctional enzyme.

6. Interesting that some genes have increased translational efficiency. Any discussion on why that might be the case?

Figures:

Figure 4 title: word protein is duplicated. The up panel/ the below panel replace with Upper graph, lower panel.

Figure 41: How was protein quantity normalized?

Suppl Figure 3: more descriptive legend is needed. What was the process? What were controls?

What was analyzed?

Suppl Fig 4: replace elections with electrons

Suppl Fig 10: Are antibodies corresponding to panels c and d mislabeled in the legend?

Minor revisions:

page 2, line 64: "which deprives ..." replace with "which abstracts"; hydrogen atom abstraction is the accepted term.

Page 5, line 138: replace "elections" with "electrons"

Discussion, line 335; replace "radical" with "intermediate"

Reviewer #1 (Remarks to the Author):

In the manuscript “C2-methyladenosine is a plant specific RNA modification in eukaryotes facilitating protein translation” the authors identify the presence of C2-methyladenosine (m2C) in the Arabidopsis transcriptome. Specifically, they find that this modification is added to chloroplast rRNA, 6 chloroplast tRNAs, and 4 cytoplasmic tRNAs. They then go on and identify the three RNA methyltransferases responsible for this methylation, which are the plant orthologs of the enzyme that adds this modification in bacterial transcriptomes (RlmN), so they name these three enzymes RLMN-LIKE1-3 (RLMNL1-3). The authors find that RLMNL1, RLMNL2, and RLMNL3 add m2C to chloroplast rRNA, chloroplast tRNAs, and cytoplasmic tRNAs, respectively. Finally, the group finds that knocking out the function of RLMN2 (*rlmnl2* mutant plants) and RLMNL3 (*rlmnl3* mutant plants) results in decreased translation output in the chloroplast and cytoplasm respectively. More specifically, the translation of transcripts encoding proteins involved in basic nucleic acid processes seem to be those whose translation is most affected in the absence of RLMNL3 function. However, the importance of this finding was not further explored or described. This is an interesting and timely study that is well done and mostly comprehensive. However, I have identified a few necessary experimental additions and manuscript edits that I have outlined below that need to be added to a revised version of the manuscript.

Necessary experimental revisions:

1) The protein blots done in Figure 4a with chloroplast protein lysates need to be redone and these samples from *rlmnl3* mutant plants also need to be added. We need to know that RLMNL2 is truly chloroplast specific and RLMNL3 is not functioning in the chloroplast.

Response #1: We thank this referee for the positive comments and thorough reviews. We have now performed extensive additional experiments to support our conclusions as suggested by this and other reviewers.

We have refined the puromycin labelling conditions for chloroplast nascent proteins (see methods). New experiments were conducted upon all three mutants and wild-type plants. Rubisco large subunit (RbcL) visualized on Ponceau S stained PVDF membrane was used as a loading control. The results confirmed that only *RLMNL2* deficiency reduces chloroplast protein translation (Please see the revised **Fig. 6a**). Please see line 250-261 for detail.

2) Relatedly, the cytoplasmic protein blots in Figure 4b should contain samples from *rlmnl1* and *rlmnl2* mutant plants to demonstrate that these two methyltransferases are chloroplast specific while RLMNL3 is the one that functions in the cytoplasm.

Response #2: We have re-performed puromycin labelling for cytosolic nascent proteins from all three mutants and WT plants. Total proteins visualized on Ponceau S stained membrane and actin were used as loading controls. The results confirmed that only *RLMNL3* deficiency reduces cytosolic protein translation (Please see the revised **Fig. 6b**). Please see line 250-261 for detail.

3) Additionally, the experiments presented in Figure 4e need to also have samples from *rlmnl3* added to once again compare and contrast the effects of the three different methyltransferases. Related to this figure better discussion should be added to the manuscript to describe why PSBA shows a protein

reduction in *rlmn12* mutants while PSAF does not. To me, it looks like chloroplasts in *rlmn12* mutant plants are upregulating the transcript of PSAF to increase the needed protein output for this locus. Maybe the authors should do a ratio of protein signal to RNA signal to demonstrate that this type of gene regulation is occurring due to the loss of m²C of tRNAs in the *rlmn12* mutant plants

Response #3: Inspired by the dual-luciferase assay, we speculated that the translation of chloroplast-encoded genes containing tandem m²A-tRNA-dependent codons might be affected by chloroplast tRNA m²A₃₇ modification. Although chloroplast-encoded genes do not contain five or more tandem m²A-tRNA-dependent codons, fourteen chloroplast-encoded genes contain four or three tandem codons (Supplementary Fig. 11a). We chose two genes—*NDHH* and *PSBA*—containing three tandem m²A-tRNA-dependent codons to examine whether their translation would be affected by m²A deficiency (Supplementary Fig. 11b). The protein gel blotting results showed the protein levels of NDHH and PSBA were decreased by 30% in *rlmn12*, but not in *rlmn11* and *rlmn13* plants (Fig. 6e and Supplementary Fig. 12a). Meanwhile, the mRNA levels of these two genes are not significantly affected in mutant plants compared with Col-0 (Supplementary Fig. 12b). Further calculation of the apparent translation efficiency (TE) by dividing the relative protein amount by the mRNA level of the corresponding gene showed that the TE of NDHH and PSBA was also significantly dropped in *rlmn12* (Supplementary Fig. 12c). Please see line 279-294 for detail.

In the previous manuscript, we mistook the nuclear-encoded gene *PSAF* as a chloroplast encoded gene. Therefore, we deleted the results of *PSAF* in the revision.

4) The experiments presented in Figures 4c and 4d that were performed for RLMN3, also need to be done for the codons of the tRNAs that are methylated by RLMN2. The readers need to see that m²C is necessary for proper translation efficiency for both chloroplast and nuclear-encoded transcripts.

Response #4: Dual-luciferase assays were performed to examine whether tRNA m²A₃₇ modification promotes translation through the decoding of corresponding codons directly. This assay is carried out in protoplast via transient transformation (see methods). But the plasmid transfection efficiency into chloroplast is very low, and it is hard to eliminate disturbance from cytosolic translation. Therefore, we turned to search naturally existing tandem m²A-tRNA-dependent codons in chloroplast-encoded genes. Please see **Response #3**.

Necessary text revisions:

1) I don't like the use of plant-specific in reference to this modification or the enzymes that add it to plant transcripts since it also occurs in bacteria and likely other organisms. I think plant-specific should be removed from this manuscript, especially from the title of the manuscript.

Response #5: In the revised manuscript, we have deleted “plant-specific” in the title and text. Although m²A was regarded as bacteria specific RNA modification previously, here we verified that m²A exists in multiple representative plant species across phyla of kingdom Plantae, and we claim that m²A is a ubiquitous RNA modification in plants.

2) There are numerous grammatical errors throughout the manuscript. Please clean these up in a revised manuscript version.

Response #6: We have edited the grammar carefully in the revision.

3) Please discuss the implications and importance of translation efficiency (TE) findings that are presented in Figure 5. Furthermore, the GO analysis that accompanies the TE analyses is very briefly described but some discussion of these findings needs to be added to both Results and Discussion to describe the importance and what the implications of these findings are for the m²C field.

Response #7: In the revision, we re-analyzed the ribosome footprinting sequencing results (please see **Fig. 7**). (1) We observed a significantly decreased overall TE in *rlmnl3* compared with Col-0 (**Fig. 7b** and **7c**), consistent with the results of polysome profiling and puromycin labelling assay (**Fig. 6b** and **7a**). (2) As the dual-luciferase assay indicated, tandem m²A-tRNA-dependent codons could lead to protein translation reduction in *rlmnl3*. We thus investigated and confirmed that the global TE decrease in *rlmnl3* is induced by the defects of tandem m²A-tRNA-dependent codons translation. (**Fig. 7d-f**, **Supplementary Fig. 17** and **18**). (3) Although we observed that disruption of *RLMNL3* led to significantly reduced TE in the genes with the high frequency of m²A-tRNA-dependent codon (top 10% in ranking; termed m²A top high frequency-decoding genes) compared with the genes with low m²A codon frequency (bottom 10% in ranking) (**Supplementary Fig. 17**), we found the result of this observation is due to the m²A top high frequency-decoding genes contains tandem m²A-tRNA-dependent codons (**Fig. 7g-i**). Thus, we demonstrated that disruption of cytosol tRNA m²A modification in *rlmnl3* mutant reduces global translation through decoding tandem m²A-tRNA-dependent codons. Rare or unfavorable codons in tandem have been proved to manipulate translation. **Here we showed the first case that tandem codons decoding by certain tRNA modification can affect protein translation.** Please see line 295-339 for detail.

Additionally, we investigated the molecular function of m²A in tRNA in the revision. We performed tRNA folding analysis using native PAGE to test the conformation of four m²A-modified chloroplast tRNAs in WT and *rlmnl2* mutant and of two m²A-modified cytosolic tRNAs from WT and *rlmnl3* mutant. We found that m²A-modified tRNAs from Col-0 had a slower migration in the native PAGE compared with m²A deficiency in the mutant plants (Fig. 5a, b), showing that m²A-modified tRNA exhibits a more loosely folded structure compared with m²A deficient tRNA. In the revision, we discussed the tRNA m²A37 function. We speculated that m²A strengthens the codon-anticodon interaction by maintaining the open-loop conformation of ASL, thereby accelerating translation elongation and preventing translation pauses on tandem codons (**Fig. 7j**).

Reviewer #2 (Remarks to the Author):

This manuscript describes the detection and superficial analysis of m²A modifications in eukaryotes, in particular in plants. LCMS was used to quantify the m²A content in isolated RNA species, but not further mapping or experimental assignment of positions of m²A modifications was performed.

Response #8: We thank this referee for the thorough reviews.

In the revision, we developed an MBN-assisted LC-MS/MS detection method (Malc) to determine the location and fraction of m²A modification in rRNA and tRNA at single-nucleotide resolution (**Fig. 2**). We performed Malc to test all identified m²A-modified tRNA and rRNA using LC-MS/MS (Table 2). We confirmed m²A locates at 3' adjacent to anticodon, typically at position 37, in m²A-modified tRNAs, and at position 2521 of Chloroplast 23S rRNA corresponding to m²A2503 in e.

Coli 23S rRNA. The Malc results showed m²A levels for each tRNA and rRNA (Fig. 2d). Please see line 106-131 for detail.

A translation phenotype in Ko lines was characterized on the molecular level. The overall results of the characterization of ko lines appear arbitrary and nondescript as might result from disruption generic gene peripherally involved in translation.

Response #9: In the revision, we performed more experiments to validate our conclusion that tRNA m²A³⁷ modification promotes protein translation through decoding tandem m²A-tRNA-dependent codons.

In the dual-luciferase assay, we added more negative controls and tested all codons decoded by m²A-modified tRNA^{Arg}_{A(D)CG} and tRNA^{Gln}_{UUG}. As the negative controls, translation levels of the empty control reporter and the negative control reporter with 6×CGG decoded by non-m²A-modified tRNA^{Arg}_{CCG} have no significant differences among different genotypic protoplasts (**Fig. 6d**). We found translation levels of 6×CAA and 6×CAG reporters decoded by the isoacceptor m²A-modified tRNA^{Gln}_{UUG} and 6×CGU and 6×CGC inserted reporters decoded by m²A-modified tRNA^{Arg}_{A(D)CG} were dramatically decreased in *rlmnl3* protoplasts compared with Col-0 (**Fig. 6d**). The reduced translation of these m²A-tRNA-dependent codons inserted reporters in *rlmnl3* was recovered by complementary expression of wild-type RLMNL3, but not catalytically dead protein RLMNL3m, suggesting the translation regulation is dependent on tRNA m²A modification. We did not observe significant differences in translation of 6×CGA reporter among different genotypic protoplasts (**Fig. 6d**), suggesting that the CGA codon might not be decoded by the m²A-modified tRNA^{Arg}_{A(D)CG}. The result was supported by the previous report showing that the anticodon stem-loop of *E. coli* tRNA^{Arg}_{A(D)CG} with m²A at position 37 negates the binding of CGA codon *in vitro* (reference 40 in manuscript).

We also performed protein immunoblotting assays for chloroplast-encoded genes *NDHH* and *PSBA* containing three tandem m²A-tRNA-dependent codons in WT and *rlmnl2*. The protein levels of NDHH and PSBA were decreased by 30% in *rlmnl2* (**Fig. 6e** and **Supplementary Fig. 14**). Polysomal profiling and ribosome footprinting sequencing results showed that disruption of cytosol tRNA m²A modification in *rlmnl3* reduced global translation through decoding tandem m²A-tRNA-dependent codons (**Fig. 7** and **Supplementary Fig. 17-19**).

The results of these experiments were consistent with each other, confirming that tRNA m²A modification facilitates protein translation through decoding tandem m²A-tRNA-dependent codons. The results in the revision should dispel the reviewer's concern about disrupting generic genes peripherally involved in translation.

The GO results do not promote a hypothesis for a molecular mechanism. The fact that a ko affects translation is trivial and very much expected when looking at these classes of RNA; again there are several components missing to construct even the beginnings of a molecular mechanism of some novelty.

Response #10: We agreed with the reviewer's comment that the GO results do not promote a hypothesis for a molecular mechanism. Thus, in the revision we deleted the GO analysis and took our efforts on the investigation of the molecular function (how and whether m²A modification modulates RNA structure, function, and fate) and the biological role of m²A (how m²A

modification takes part in the biological process of protein translation).

The investigation of m²A molecular function:

- 1) The quantification assays of m²A-modified rRNAs and tRNAs in wild-type and mutant plants showed that removal of m²A modification in mutant plants did not change the transcript levels of m²A-modified rRNAs and tRNAs compared with Col-0 (**Supplementary Fig. 9 and 10**), suggesting that lack of m²A modification has little effect on RNA steady-state. Please see line 204-215 for detail.
- 2) We performed new experiments to examine the aminoacylation levels of all m²A-modified tRNAs in wild-type and mutant plants. The results show that tRNA m²A37 modification does not affect aminoacyl-tRNA synthesis (**Supplementary Fig. 11**). Please see line 216-223 for detail.
- 3) We analyzed the conformation of m²A-modified tRNAs in wild-type and mutant plants by native polyacrylamide gel electrophoresis, and showed that m²A-modified tRNA exhibits a more loosely folded structure compared with m²A deficient tRNA (**Fig. 5**), demonstrating that tRNA m²A37 modification promotes a more flexible conformation of tRNA. Please see line 224-237 for detail.

The investigation of the biological role of m²A:

Due to the dual-specificity of bacteria RlmN, previous works could not elucidate the respective biological roles of rRNA and tRNA m²A through the *ΔrlmN* E. coli strain. Substrates of RlmN-like proteins in Arabidopsis are specialized, thus we could investigate the function of rRNA and tRNA m²A in protein translation separately.

- 1) The results of the puromycin labelling assay demonstrate that tRNA m²A37 modification, but not rRNA m²A modification, increases the efficiency of protein translation (Please see the revised **Fig. 6a and 6b**). Please see line 238-249 for detail.
- 2) The dual-luciferase assays with six repeated cytosolic m²A-tRNA-dependent codons decoded by m²A-modified tRNA and further protein quantification of individual chloroplast-encoded genes (*NDHH* and *PSBA*) containing three tandem chloroplast m²A-tRNA-dependent codons demonstrate that the tRNA m²A37 facilitates protein translation through decoding m²A-tRNA-dependent codons in chloroplast and cytosol (Please see the revised **Fig. 6c-e** and **Supplementary Fig. 12-14**). Please see line 250-294 for detail.
- 3) The polysome profiling results further demonstrate that loss-of-function of *RLMNL3* decreases cytosolic translation. The ribosome footprinting sequencing results showed that deficiency of tRNA m²A37 reduces global translation efficiency through decoding tandem m²A-tRNA-dependent codons (Please see the revised **Fig. 7** and **Supplementary Fig. 17-19**). Please see line 295-339 for detail.

Often, statements on results lack tangible content. For example, the statement that „the eukaryotic system is more complex than bacteria” (line 301) is naïve. First, this is true for essential all types of biomolecules, including DNA, Proteins, RNA and every single type of RNA modification. Second, chloroplast and mitos are of bacterial origin, and together with the cytosolic system it is perfectly normal

to have multiple homologues of an enzyme to address different cellular compartments. Indeed this is the observed rule, so RLMNL is just another example of a standard case, rather than anything extraordinary.

Response #11: In the revision, we have deleted the sentence “indicating the eukaryotic system is more complex than bacteria”. Although m²A was regarded as bacteria specific RNA modification previously, here we verified that m²A exists in multiple representative plant species across phyla of kingdom plantae, and it is a ubiquitous RNA modification in plants.

Furthermore, we identified six evolutionarily distinct clades of eukaryotes RlmN-like proteins, especially in Diaphoretickes. At least four of these clades have definite phylogenetic relationships with prokaryotic analogues from diverse taxa (**Fig. 3**). However, only the origination of clade I which has the highest similarity to blue-green algae RlmN could be rationalized through plastid endosymbiogenesis. The HGT origination of RlmN-like proteins in eukaryotes is likely to occur several times independently, indicating the universal benefits of RlmN and/or m²A gaining. Please see line 145-159 for detail.

In addition, the authors do not really have any novel sites to report; m²A appears in tRNA and rRNA, which was known before. The authors are not capable of precisely defining the position of m²A modification within tRNA or rRNA by themselves, instead they infer sites by homology from literature on other organisms.

Response #12: In the revision, we have performed additional experiments. We found that m²A-modified tRNA exhibits a more loosely folded structure compared with m²A deficient tRNA, which regulates protein translation efficiency through decoding tandem m²A-tRNA-tRNA-dependent codons. The novel sites in the revised manuscript are listed as follows.

- 1) We developed an MBN-assisted LC-MS/MS detection method (termed Malc) at single-nucleotide resolution to determine the precise location and fraction of m²A modification in rRNA and tRNA. The Malc method can also be applied to other modifications and RNA types. Please see new **Fig. 2** and Line 106-123.
- 2) In previous structural and biochemistry studies, the molecular function of 23S rRNA m²A2503 has been partially revealed. However, the functions of m²A on tRNA remain largely unknown. Our results showed that m²A37 promotes a more flexible conformation of tRNA, but does not affect tRNA steady-state and aminoacylation. Please see new **Fig. 5** and **Supplementary Fig. 9-11**.
- 3) Due to the dual-specificity of bacteria RlmN, previous works could not elucidate the exact biological role of rRNA and tRNA m²A through $\Delta rlmN$ *E. coli* strain. Substrate specialization of RlmN-like proteins in plants provides feasibilities to investigate the biological functions of m²A on tRNA separately. We demonstrate that tRNA m²A37 modification, but not rRNA m²A modification, increases the efficiency of protein translation through decoding tandem m²A-tRNA-dependent codons. Please see the revised **Fig. 6** and **Fig. 7**, and **Supplementary Fig. 12-19**.
- 4) The ribosome footprinting sequencing results showed that genes with tandem m²A-tRNA-dependent codons reduce protein translation in *rlmn13* mutant plants. Although rare or

unfavorable codons in tandem have been proved to manipulate translation, we for the first time demonstrated that tandem codons dependent on certain tRNA modification can affect protein translation.

- 5) Previous studies about m²A were limited in bacteria, and only several plant tRNAs were identified to contain m²A modification. We verified the ubiquity of m²A in plants. Moreover, we identified six evolutionarily separated clades of RlmN-like proteins in eukaryotes, especially in Diaphoretickes. The intricate phylogeny and widely spreading of eukaryote RlmN-like proteins indicate their complicated HGT origination and universal benefits of RlmN and/or m²A gaining.

Beyond this, the study raises, but then leaves unanswered, some questions that might eventually become interesting. Can they confirm by methods other than LCMS, that the tRNAs are only partially modified with m²A at A37? If this were accurate (which was not convincingly shown), then the question would be, if there is another modification at A37 instead, resulting in populations of mixed A37 modifications, which is doubtful. If the remainder of A37 were instead unmodified, should that not cause massive problems with frameshifting, as observed for various ko of A37 modifications? Again, a molecular mechanism is missing.

In summary, the study is largely descriptive, parts of it are immature, and it lacks the kind of conceptual novelty appropriate for this journal. I recommend rejection.

Response #13: In the revision, we have developed the Malc method to detect m²A fraction and position in tRNA and rRNA at single-nucleotide resolution (Please see new **Fig. 2**). We found the modification ratio of all detected m²A sites is around 80%. In addition, we separately overexpressed these three m²A methyltransferases in Col-0 and found the m²A levels in the overexpressed lines were not changed compared to Col-0 (**Supplementary Fig. 7b** and **Supplementary Fig. 8**), indicating that m²A is a constitutive modification. In the revision, we have performed new experiments to demonstrate that m²A-modified tRNA exhibits a more loosely folded structure compared with m²A deficient tRNA, which regulates protein translation efficiency through decoding tandem m²A-tRNA-dependent codons. Please see **Response #12** for detail.

Reviewer #3 (Remarks to the Author):

In this manuscript, a specific type of adenosine methylation (the m²A modification) in different classes of RNA was investigated in Arabidopsis, 10 tRNA genes that harbor these methyl marks in the anticodon loop were identified, the RLMNL enzymes responsible for depositing the m²A modifications were knocked out, and phenotypic consequences of the mutations were examined. The authors provide convincing evidence to suggest that the three RLMNL genes in Arabidopsis play specialized roles, with RLMNL1 modifying chloroplast rRNA, RLMNL2 methylating chloroplast tRNA, and RLMNL3 targeting cytoplasmic tRNA. Loss of function T-DNA mutants (one allele per gene) were isolated, but showed no morphological defects. In contrast, at the molecular level, all three mutants had abnormal m²A profiles, that were complementable by a WT copy of the respective RLMNL gene. Two of the

mutants, *rlmnl2* and *rlmnl3*, were concluded to have global defects in translation, but the data provided in support of that claim are not very robust and missing important quality control metrics (see below). With *rlmnl3* having over a thousand genes translationally downregulated and nearly 300 genes upregulated at the translational level, it does not make sense to me that the mutant displays normal plant morphology. Such a drastic and global effect on translation should lead to a sick or at least slow-growing plant. In light of that discrepancy and given that some of the essential quality controls of the Ribo-seq data (see below) have not been provided, I am not at all confident in the validity of that data.

Response #14: We thank this referee for the thorough reviews.

According to previous works, single modification loss from tRNA and rRNA usually do not lead to severe development and growth defect under normal conditions (Please see Vicente Ramírez et al. *MPMI* 31:12, 1323-1336 (2018) and Peng Chen et al. *BMC Plant Biol* 10, 201 (2010)). We observed all these three mutant lines are hypersensitive to aminoglycoside antibiotics, consistent with the functions of RlmN-like proteins in translation (**Supplementary Fig. 20**).

Assuming that the data are true and the RLMNL-mediated m2A modification in rRNAs and tRNAs affects global mRNA translation, the question arises as to what regulates m2A modifications? Are there specific environmental conditions or developmental processes that alter RLMNL levels or activity? Besides looking at the expression of the three RLMNL genes across plant organs, this question was not investigated. Just by looking at publicly available transcriptomic data, I see that some abiotic stress factors appear to down-regulate RLMNLs (e.g., salt and drought repress RLMNL1, and salt and mannitol decrease RLMNL2 levels), whereas heat and UVB induce RLMNL3 during the recovery period. The authors thus missed an obvious opportunity to connect these genes and the respective RNA modifications they confer with specific physiologically relevant biological pathways. It would have made sense to test the phenotypes of the mutants under the stresses that are found to regulate these three genes

Response #15: We carried out a series of phenotypic analyses on wild-type and mutant plants. Regrettably, these homozygous mutant plants exhibit neither distinct phenotypes under normal conditions nor changed sensitivity to heat, cold, and dark stress. We found that defects of *RLMNL1-3* lead to hypersensitivity to aminoglycoside antibiotics and marginally increased resistance to osmotic stress (**Supplementary Fig. 20**). Aminoglycoside antibiotics (such as paromomycin and G418) disturb protein translation elongation. Consistent with the functions of RLMNL1/2/3 in translation, all three mutant lines showed delayed germination and suppressed early-stage vegetative growth (**Supplementary Fig. 20a and 20b**).

We also adopted expression data of RLMNL1/2/3 from eFP browser. However, we found that abiotic stresses only lead to limited expression variation of these genes compared with stress response genes. Again, this is usually the case for a single modification loss from tRNA or rRNA. We expect the loss of multiple modifications would lead to more severe defects in plants, which could be demonstrated in further studies. Thus, we focus our studies on the molecular functions of tRNA m²A³⁷ in the revised manuscript.

– again, this is another missed opportunity. Furthermore, no overexpression studies are reported – do the levels of m2A modifications in rRNA and tRNA increase and does that alter global translational profiles? In the absence of these studies to suggest a clear physiological role for RLMNL-mediated

m2A modifications in housekeeping RNAs, my enthusiasm for this work is moderate at best.

Response #16: We conducted overexpression of all three RLMNL1/2/3 in Col-0 and found the m²A levels in the overexpressed lines were not changed compared to Col-0 (**Supplementary Fig. 7b** and **Supplementary Fig. 8**), indicating that m²A is a constitutive modification. Please see line 199-203 for detail.

My detailed comments are provided below.

Fig 1D – "Homo sapiens" is misspelled

Response #17: This figure has been replaced in the revision.

Line 120-126 and fig 1D-E, the math does not make sense. One out of seven copies is not 0.7%, but 14.2%. 0.7% means 7 out of a 1000 (or 1 out of 142). The ratios should not be measured in %. So, the graph labeling in Fig 1d and 1e are clearly misleading and the respective description in the text for the different kinds of tRNA makes no sense. Also, it would be more logical to plot m²A/(m²A+A), i.e. the proportion of modified A's relative to the total (modified and unmodified) amount of A's. These comments also apply to fig 3A, C, D.

Response #18: The quantification of m²A has been re-performed in the revision. Specifically, it is routine to calculate the RNA modification ratio (modified versus unmodified base) in the field of RNA modifications. Please see these literatures (Shay Geula et al. *Science* 347:6225, 1002-1006 (2015); Xuechai Chen et al. *ACS Chemical Neuroscience* 10:5, 2355-2363 (2019); and Wei Li et al. *BMC Genomics* 23, 105 (2022)).

Line 132, please, specify what is meant by "high sequence similarity" (i.e., provide numerical values of sequence identities).

Response #19: In the revision, we provided BLAST E-value for RLMNL1/2/3 against *E. coli* RlmN. Please see line 137 for detail.

Fig 2A, in the legend, provide accession numbers of the proteins used in the phylogenetic analysis

Response #20: This figure has been replaced, and accession numbers are provided in the new phylogenetic figure (**Fig. 3**).

Fig S4, indicate the program used to make protein alignments

Response #21: We used NCBI-Blast to make protein alignments. We have mentioned it in the figure legend.

Lines 146-148 and fig S5, provide references for the sequence similarity network algorithm used.

Response #22: This part was deleted in the revision

Fig 7, in the gene schematics in panel A, please, mark positions of genotyping primers employed in panel B and qRT-PCR primers used in panel C. The genotyping gels shown in panel B do not make sense, as for each sample amplification of the mutant and WT fragment needs to be performed and thus two lanes should be displayed for each DNA sample. I assume only WT fragment reactions are shown,

but this is not obvious, as no description of what is being amplified is provided.

Response #23: The primers used for genotyping were illustrated in the revision. The gel band amplified by LP+RP (for wild-type genes) and LB+RP (for T-DNA insertions) was indicated in the revised **Supplementary Fig. 7a**. For the detail of three-primer genotyping, please see the website <http://signal.salk.edu/tdnaprimers.2.html>.

Lines 174-176, lack of morphological defects in the *rlmnl* mutants suggests that the m²A modification does not play a physiologically relevant role. This does not agree with the dramatic translational changes reported later in this study.

Response #24: Similar questions have been answered in **Response #14** and **Response #15**.

Lines 179-180, the way this sentence is phrased, it is unclear that the 5% modification in the mutant is not the absolute level but rather 5% of the WT level. Again, the ratios should not be expressed in %.

Response #25: The quantification of m²A has been re-performed in the revision, and these descriptions have been deleted from the manuscript.

Fig 3A, given the exonic location of the *rlmnl2* and *rlmnl3* T-DNA mutants, how are residual levels of m²A modifications maintained in the chloroplast and cytoplasmic tRNA (as 0-15% of WT activity are seen)? Is it possible that a small fraction of the cytoplasmic enzyme finds its way into plastids and, vice versa, some of the plastidic enzyme is retained in the cytoplasm? This should be easy to test with the generation of double *rlmnl2 rlmnl3* and triple *rlmnl1 rlmnl2 rlmnl3* mutants.

Response #26: We have re-performed the experiments and solved the problem. Please see the revised Fig. 4. We found that m²A modifications in chloroplast rRNA, four chloroplast tRNA species (tRNA^{Arg}_{ACG}, tRNA^{His}_{GUG}, tRNA^{Ser}_{GGA}, and tRNA^{eMet}_{CAU}), and two cytosolic tRNAs (tRNA^{Arg}_{ACG} and tRNA^{Gln}_{UUG}) were completely removed in *rlmnl1*, *rlmnl2*, and *rlmnl3* mutant, respectively. The m²A modification in these types of RNAs was dependent on the activity of RLMNL1/2/3. The results demonstrated that RLMNL1/2/3 are m²A methyltransferase for chloroplast rRNA and tRNA, and cytosolic tRNA, respectively. Please see line 169-184 for detail.

Lines 189, what do the authors mean by “other related proteins”? Are there additional RLMNL-like genes in the Arabidopsis genome besides those three?

Response #27: There is no additional RlmN-like protein in Arabidopsis. We have deleted the sentence in the revised manuscript.

Fig 4B, S10A and lines 235-238, in the absence of a proper loading control, the data are difficult to interpret. Less puromycylation signal in the *rlmnl3* mutant may simply mean less sample was loaded, but the Coomassie Blue-stained loading control is clearly overloaded and impossible to accurately compare across the lanes.

Response #28: In the revision, Rubisco large subunit (RbcL) visualized on Ponceau S stained PVDF membrane was used as a loading control for puromycin-labelled chloroplast nascent proteins. Total proteins visualized on Ponceau S stained membrane and Actin were used as loading controls for puromycin-labelled cytosolic nascent proteins.

Line 255 and 266, fig 4D should be referenced here, not fig 4C.

Response #29: We have corrected the mistake.

Fig 4B, the reduction in the PSBA protein levels in *rlmn2* is very obvious, but for PSAF, the protein level seems to be elevated in *rlmn2*, concomitant with an increase in its RNA. If the authors want to make a point that there is a defect in PSAF translation in the mutant, protein quantification needs to be provided that is performed on multiple independent replicates of the Western blots.

Response #30: In the previous manuscript, we mistook the nuclear-encoded gene PSAF as a chloroplast encoded gene. Therefore, we have deleted the results of PSAF in the revision. As mentioned in **Response #3**, we chose *NDHH* to examine the protein expression together with *PSBA*. Both *NDHH* and *PSBA* are chloroplast encoded genes containing three tandem chloroplast m²A-tRNA-dependent codons. The replicated protein gel blotting results showed that both *NDHH* and *PSBA* protein levels were decreased in *rlmn2*, but not affected in *rlmn1* and *rlmn3* plants (**Fig. 6e and Supplementary Fig. 14a**). Meanwhile, the mRNA levels of these two genes are not significantly affected in mutant plants compared with Col-0 (**Supplementary Fig. 14b**). We calculated the ratio of protein signal to RNA signal as Reviewer #1 suggested and found that the translation efficiency of *NDHH* and *PSBA* was reduced in *rlmn2* (**Supplementary Fig. 14c**). Please see line 279-294 for detail.

Fig 5 and S11-12, The Ribo-seq data are not thoroughly described. How many million reads were obtained for each library, what percentage corresponded to mRNA, rRNA, and tRNA, etc. and what percentage mapped uniquely to the genome? These are essential parameters that need to be reported. It is also customary for Ribo-seq studies to show alignments of reads around the start and stop codons to demonstrate low read number in the 5'UTR and lack of reads in the 3'UTR and to show read periodicity in the genic ORF. Without those data, it is impossible to evaluate the quality of the obtained datasets.

Response #31: The quality summary of ribosome footprinting sequencing is presented in the revised Supplementary Dataset 1.

English language needs to be edited throughout the entire manuscript by a native speaker. For example, in the abstract and introduction the following sentences need to be modified (but this list is not comprehensive):

Line 16, “the study of m2A in eukaryotes have remained unclear” sounds awkward.

Line 51, change “locates” to “is located”

Line 70-71, change “via a similar mechanism to RlmN catalyzing” to “via a mechanism similar to that catalyzed by RlmN”

Line 71, change “RlmN (or Cfr), which affecting the susceptibility to multi-kinds of antibiotics” to “RlmN (or Cfr), which affects the susceptibility to a variety of of antibiotics”

Line 76, change “tRNA contain m2A modification” to “tRNA contain an m2A modification”

Line 77-78, what did the authors mean by “quantitative m2A levels”, quantifiable or variable? All “levels” by definition are “quantitative”.

Line 80, remove “in” from “in consistent”

Response #32: Thanks. We have changed them and edited the grammar carefully in the revision.

Reviewer #4 (Remarks to the Author):

There are few important questions that need to be addressed:

1. Site of tRNA modification. While likely evolutionarily conserved, site of modification is not unambiguously assigned. The assumption made here is that A37, like in RNA, is the site of modification. Is there a direct evidence that authors can provide, perhaps through partial RNA digestion and fragment analysis? Additionally what are negative control tRNA that were used in Suppl Figure 3?

Response #33: We thank this referee for the positive comments and thorough reviews.

In the revision, we developed an MBN-assisted LC-MS/MS detection method (“Malc”) at single-nucleotide resolution to determine the location and fraction of m²A modification in rRNA and tRNA (Please see the new **Fig. 2**). In all detected tRNAs, m²A is located 3’ adjacent to anticodon, typically at position 37. The m²A modification is located at position 2521 of Chloroplast 23S rRNA, which is the corresponding position of *E. coli* 23S rRNA m²A2503. Please see line 106-131 for detail.

As we understand it, “negative control tRNA” mentioned by this referee means the tRNA has G37 rather than A37 or the tRNA has A37 but proved to be non-m²A-modified. We examined cytosolic tRNA^{His}_{GUG} in Arabidopsis and other representative plants, which have G37 rather than A37 and do not contain m²A. Cytosolic tRNA^{Gln}_{CUG} has A37 but it does not contain m²A in tobacco as previously reported. Please see Table 2 in the revised manuscript.

2. Dual Luciferase reporter system needs additional controls. Have non m²A modified tRNA, ideally encoding same amino acids, been used in this assay and are any changes in luc expression observed? This is essential to eliminate possible sequence dependent translational changes unrelated to tRNA modification (for example, as described here: <https://www.biorxiv.org/content/10.1101/571059v2.full>).

Response #34: In the revision, we adopt reporters with 6×CGG for non-m²A modified tRNA^{Arg}_{CCG} as a negative control since it encodes arginine but does not contain m²A. The results showed that translation levels of the negative control reporter with 6×CGG have no significant differences among the four different genotypic protoplasts (**Fig. 6d**). However, the translation levels of the reporters with 6×CGU or CGC for m²A-modified tRNA^{Arg}_{A(D)CG} were significantly decreased in *rlmnl3*, which is dependent on the activity of RLMNL3 (**Fig. 6d**). Additionally, we found no significant differences in translation of 6×CGA reporter for m²A-modified tRNA^{Arg}_{A(D)CG} among different genotypic protoplasts (**Fig. 6d**), suggesting that the CGA codon might not be decoded by the m²A-modified tRNA^{Arg}_{A(D)CG}. The result was supported by the previous report showing that the anticodon stem-loop of *E. coli* tRNA^{Arg}_{A(D)CG} with m²A at position 37 negates the binding of CGA codon *in vitro* (reference 40 in the manuscript). Please see line 250-278 for detail.

3. Additional clarifications are needed regarding Fig 4 and Suppl Fig 10. First, the sample size of two randomly selected proteins is rather low. How many biological replicates were analyzed? PSAF transcript seems 3x increased, couldn’t that cause increase in protein level? What do authors hypothesize is the reason that some genes, but not others are affected by the absence on m²A in tRNAs? Is there any information that can be derived from the nature of codons? Any ways to test, perhaps similar to luciferase assay they used?

Response #35: Reviewer #1 has a similar question. Dual-luciferase assays are carried out in protoplast via transient transformation (see methods). But the transferring efficiency of plasmids into chloroplast is very low, and it is hard to eliminate disturbance from cytosolic translation. Therefore, we searched chloroplast-encoded genes containing tandem codons decoded by m²A-modified tRNA, whose translation would be decreased in *rlmnl2* mutant plants. The pattern of five or more tandem codons decoded by m²A-modified tRNA is absent. Nevertheless, the pattern of four or three tandem chloroplast m²A-tRNA dependent codons was found in several genes, and the details were illustrated in **Supplementary Figure 13**. Intriguingly, *PSBA* is one of them. It may explain why protein levels of *PSBA* were reduced in the *rlmnl2* mutant line. Besides, we also chose another gene containing the tandem codons, *NDHH*, to examine the protein amount together with *PSBA*. The protein gel blotting results show that both *NDHH* and *PSBA* are decreased in *rlmnl2*, but not affected in *rlmnl1* and *rlmnl3* plants (**Fig. 6e and Supplementary Fig. 14a**). Meanwhile, the mRNA levels of these two genes are not significantly affected in mutant plants compared with Col-0 (**Supplementary Fig. 14b**) We also calculated the ratio of protein signal to RNA signal as Reviewer #1 suggested, demonstrating the translation efficiency of *NDHH* and *PSBA* only affected in *rlmnl2* (**Supplementary Fig. 14c**). Please see line 279-294 for detail.

In the previous manuscript, we mistook nuclear-encoded *PSAF* as a chloroplast encoded gene. Therefore, we deleted the results of *PSAF* in the revision. Two biological replicated were performed for *NDHH* and *PSBA* (Please see the revised **Fig. 6e and Supplementary Fig. 14**).

4. Discussion, lines 310-312: Disruption of *RLMNL2* significantly decreases the protein synthesis of *PSBA* and *PSAF*. This does not appear to be the case for *PSAF*.

Response #36: In the previous manuscript, we mistook nuclear-encoded *PSAF* as a chloroplast encoded gene. In the revision, we have deleted the results of *PSAF*.

5. Discussion, lines 321-322: this effect should not be attributed to 23S rRNA (event though that explanation has been suggested) as *E. coli* RlmN is a multifunctional enzyme.

Response #37: The discussion part has been rewritten in the revised manuscript.

6. Interesting that some genes have increased translational efficiency. Any discussion on why that might be the case?

Response #38: We suppose it might result from the ribosome footprinting sequencing method itself. The polysome profiling showed that global protein translation was decreased in *rlmnl3* mutant compared with Col-0 (Please see the revised Fig. 7a). In the revision we did not differentiate “TE up” or “TE down” genes. We carried out analyses based on the overall TE fold change of genes grouped by different criteria. We found disruption of cytosol tRNA m²A37 modification in *rlmnl3* mutant reduces global translation through decoding tandem m²A-tRNA-dependent codons.

Specifically, Nuclear-encoded genes with no less than three tandem cytosolic m²A-tRNA-dependent codons (CAA, CAG, CGU, and CGC) were grouped by their maximum number of the tandem codons (**Fig. 7d and Supplementary Fig. 17**), and other nuclear-encoded genes were assigned to a group termed “others” (**Fig. 7d**). Overall, the translation efficiency of the genes with no less than three tandem m²A-tRNA-dependent codons was significantly decreased in *rlmnl3*

compared with that of others (Fig. 7e). The extent of TE reduction in *rlmnl3* further showed an ascending trend over the extended number of the tandem codons in genes (Fig. 7f). As negative controls, we also chose six sets of non-m²A-dependent codons (four codons for each set) and grouped all nuclear-encoded genes by their maximum number of these codons in tandem. Analysis of their translation efficiency variation showed no obvious difference between the genes with three or more tandem non-m²A-dependent codons and other genes upon the disruption of *RLMNL3* (Supplementary Fig. 18). These results suggest that m²A promotes tandem m²A-tRNA-dependent codons translation.

Rare or unfavorable codons in tandem have been proved to manipulate translation. **We for the first time demonstrated that tandem codons dependent on certain tRNA modification can affect protein translation.** Please see line 295-339 for detail.

Figures:

Figure 4 title: word protein is duplicated. The up panel/ the below panel replace with Upper graph, lower panel.

Response #39: This figure has been replaced with a new version.

Figure 41: How was protein quantity normalized?

Response #40: In the revision, Rubisco large subunit (RbcL) visualized on Ponceau S stained PVDF membrane was used as a loading control for puromycin-labelled chloroplast nascent proteins. Total proteins visualized on Ponceau S stained membrane and β -actin were used as loading controls for puromycin-labelled cytosolic nascent proteins.

Suppl Figure 3: more descriptive legend is needed. What was the process? What were controls? What was analyzed?

Response #41: This figure has been replaced with a new version. All nuclear and chloroplast encoded tRNA isoacceptors were grouped according to the anticodon. The tRNAs with adenosine at position 37 were illustrated in red color, and m²A-modified tRNAs were highlighted with a black box. Please see the revised Supplementary Fig. 3.

Suppl Fig 4: replace elections with electrons

Response #42: We have revised it.

Suppl Fig 10: Are antibodies corresponding to panels c and d mislabeled in the legend?

Response #43: The figure has been deleted in the revision.

Minor revisions:

page 2, line 64: “which deprives ...” replace with “which abstracts”; hydrogen atom abstraction is the accepted term.

Page 5, line 138: replace “elections” with “electrons”

Discussion, line 335; replace “radical’ with “intermediate”

Response #44: We have revised them and edited the grammar carefully in the revision.

Reviewer #1 (Remarks to the Author):

The revised version of the manuscript is significantly improved overall. However, in my first round of review I missed that there are only 2 replicates of all sequencing experiments performed for this study. This is not sufficient for a high level publication and at least 1 additional biological replicate should be performed and added to the analysis to publish a study based on genomics experiments in a Nature group journal. These additional data should be added to a revised manuscript before publication of this study.

Reviewer #3 (Remarks to the Author):

This is a resubmission and was one of the original reviewers for this manuscript. Some of my previous comments have been addressed, but some concerns remain. As three years have passed since the first submission and many major changes have been made to the manuscript, I chose to evaluate this work as a brand-new submission. Based on the data presented, I have no doubt that RLNML1,2,3 in Arabidopsis catalyze the m2A modifications in chloroplast rRNA, chloroplast tRNA, and cytosolic tRNA, respectively. I was also impressed by the new Malc method to confirm the specific locations of m2A modifications in the RNA. However, the data on the mechanistic consequences of m2A modifications on the structure and function of target RNAs are not very robust or internally consistent (see below). Likewise, the biological outcomes of rlnml mutations at the phenotypic level are not well characterized (only one allele per gene is examined, and the morphological defects are super-mild, which is inconsistent with the strong molecular phenotypes presented). Finally, the question of how specific plastidic and cytoplasmic target RNAs (and specific adenosines within them) are chosen by RLNML1/2 and 3, respectively, is not addressed.

The data to suggest a more compact structure of plastidic and cytosolic tRNAs in the absence of m2A modifications at A37 in rlnml2 and rlnml3 mutants, respectively, are not very robust (mobility shift results are questionable, see below). The data on puromycin labeling of nascent peptides in rlnml2 and rlnml3 are inconsistent with the overall plastidic and cytoplasmic protein levels in the mutants (should not nascent peptide synthesis be proportional to total protein levels in these lines?). Some Ribo-seq data quality controls are missing (even though I specifically asked for them in the original review 3 years ago). The observation on the need for the m2A-modified-tRNA-dependent codons to be found in tandem in transcripts for the rlnml3 mutant to have an effect on mRNA translational efficiency was not re-retested in dual luciferase assays (only the effect of tandem codons was analyzed). The lack of major phenotypes in the rlnml mutants is surprising given the prominent molecular defects of these lines and examining only one allele of each gene is not sufficient (in the absence of additional alleles, complementation lines should have been included in the phenotypic assays).

My specific comments are listed below:

Lines 65 and 77, specify: digested with what?

Figure 1b, c – define TIC and m/z in the legend (the same applies to Supplemental figure 1). Specify from what samples (tissue, growth conditions, age) total RNA and purified RNAs were extracted.

Lines 72-73, I do not think this conclusion can be made from looking at RNA from one tissue sample. Nowhere do the authors specify what tissues were used for RNA extraction. Plant growth conditions and sample types can influence the relative ratios of different types of RNA and their nucleotide modification profiles.

Lines 74-82 and Table 1, again, I could not find the information on the growth conditions and tissue types for any of the plant species. I do not think any of these values are reproducible or comparable given that the plant tissues were sourced from different, non-standardized places (colleagues, supermarkets and other merchants, etc.). The source of HeLa cells is not listed anywhere. The text refers to "human total RNA as well as separated RNA types", but Table 1 shows total and poly(A) RNA only. Reconcile!

Supplemental Figure 3, the table is very hard to read. Why does position 35 list the entire anticodon rather than a single nucleotide? Why is the position 37 (3' adjacent to the codon) that is key to this work not shown? What is iM and eM (it is not obvious that the initiator and elongator Met are meant)? Also, the reference to nuclear-encoded (table) versus cytoplasmic (narrative) tRNA is confusing. I think both terms are supposed to mean the same thing.

Lines 86-87, state the total number of tRNAs with m2A modifications for which the biotinylated probes needed to be designed.

Figure 2a, the Malc method is ingenious (kudos for that!), but it does require an a priori knowledge of the modified nucleotide.

Figure 2b,c, mark Fx-1, Fx and Fx+1 next to the probe names (the same is true of supplemental figure 4).

Figure 2 legend, change the description of the procedure to present tense.

Lines 119-123, while I understand the logic of why FX-1 (rather than Fx) was used to calculate the m2A modification frequency, I do not understand HOW that frequency was calculated and the relation between the m2A/A value plotted in figure 2C (~0.14) and the modification ratio stated in figure 2D (85%). Please, explain.

Line 130, what is the reason to think that the modification is constitutive if only one tissue was examined?

Line 138, no data are presented in support of RMNLs being divergent in sequence from Cfr. Is there a Cfr like protein in Arabidopsis? If not, to what Cfr are RMNLs being compared and what are the E-values?

Lines 166-168 and Supplemental figure 6b, it is customary to include marker proteins for comparison. While chlorophyll autofluorescence is a good marker substitute for chloroplasts, cytoplasmic protein localization may be confused with that in the endomembrane system, for example.

Supplemental figure 7, with three primer genotyping of insertional mutant lines, it is helpful to show that in a heterozygous sample, both WT and mutant bands are detectable, as it is not uncommon for one of the bands (usually, the smaller-sized one) to amplify preferentially. This is not essential to include as long as the RT-qPCR data are "clean", meaning that there is no amplification product detected in the homozygous mutant with the primers spanning the insertion site. Herein, where does the residual RT-qPCR signal in the homozygous T-DNA lines come from? Do the qPCR primers used indeed span the insertion site? Does at least one of the RT-qPCR primers anneal to an exon-exon junction to avoid the amplification from contaminating genomic DNA? Please, mark the RT-qPCR primers used in the gene models shown in figure 7a (but distinguish them from the T-DNA genotyping primers, e.g., by color and figure legend clarification). This is something I requested in my original review, but the authors only marked the genotyping primers.

Line 211, rRNA transcription and stability.

Supplemental figure 11, what does OH- mean? Please, specify in the figure legend.

Figure 5 (and supplemental figure 21), the mobility shift does not look very convincing for some tRNAs. For example, the tRNASer GGA gel seems to be slightly slanted, and other shifted samples (e.g., tRNAeMet CAU) may show migration differences simply due to uneven RNA quantity (with overloaded samples traveling faster) or different concentration of impurities (e.g., salt levels). I am not sure what the authors mean by "more loosely-folded conformation of tRNA" – some of the base pairing not happening? If so, which bases (I assume that the modified base itself is in the anticodon loop and is not paired)? If the hypothesis is that m2A affects base pairing and, hence,

RNA secondary structure, then single- and double-stranded nucleases should produce different tRNA cleavage patterns in WT versus the mutant on a polyacrylamide gel. Alternatively, as the authors themselves suggest in the discussion, NMR experiments and molecular dynamics simulations should have been performed.

Figure 6a,b, I am confused by the experimental design and data shown in panels A and B. I can imagine the data being normalized per fresh weight of each sample, but not per total protein. Would not the total and nascent protein levels be proportional (meaning reduced nascent protein synthesis would ultimately lead to less protein accumulating)? If the gel loading is controlled for to keep the total amount of protein in each lane comparable between samples (and for a translation-compromised mutant, one would need to load more sample to get comparable levels of the protein of interest, RbcL or Actin/total protein), would not that also equalize nascent protein amount? To me, the data make no sense. How can one have little nascent protein made but have comparable RbcL or Actin (or total) levels overall? Do RbcL and Actin not have m2A-modified-tRNA-dependent codons in their sequences? Even if that were the case, this would not explain the fairly even Ponceau S staining relative to the very skewed puromycin-labeled proteins in panel B. What percentages of genes (plastidic and nuclear, respectively) harbor m2A-modified-tRNA-dependent codons? Also, I keep thinking that the drop in puromycin-labelled nascent proteins in rlmn12 and 3 mutants is very profound to not have a major phenotypic consequence for plants.

Figure 6d, do F-luc and R-luc have any of the Rlmn13-dependent (i.e., m2A-modified-tRNA-dependent) codons in their native sequences?

Line 280 and Supplemental Figure 12, what does "two out of three" hypothesis refer to? Please, explain.

Line 282, why do the Rlmn13-dependent codons need to be found in tandem for the rlmn13 mutant to have an effect? Why was not this hypothesis tested in dual luciferase assays?

Supplemental figure 14a, are the protein expression differences statistically different? Also, the very mild reduction in the levels of two candidate proteins in rlmn12 is not consistent with the major (several fold) nascent protein reduction seen in figure 6a.

Figure 7a, I do not think such a mild reduction in the polysomal fraction is real, especially that it is not accompanied by an increase in monosomal peaks. Different biological replicates of WT samples often show more variation than that. I suggest that the authors overlay three biological replicates each of WT and mutant to demonstrate that the mutant is, in fact, different from WT. State in the narrative how translational efficiency was calculated (currently the definition is buried in the methods).

Lines 300-302, as written, it is unclear that the authors are talking about a ribosome footprinting experiment (aka Ribo-seq) when referring to sequencing data, since polysome profiling is mentioned in a prior sentence. Rephrase. Define FPKM.

Supplemental dataset 1 (and corresponding method description), in Ribo-seq quality metrics, the low number of reads in the UTRs looks good, but the 3nt periodicity of the dataset is not shown, so I am unable to reliably judge the validity of Figure 7 data and, accordingly, of the conclusions made. In the dataset table, I found the reference to "input" confusing. Please, rename "input" in the table as "RNA-seq" and change "Ribo" to "Ribo-seq". These terms would likely be more familiar to the reader and should probably be used in the narrative as well.

Supplemental figure 16, what is the cutoff for up- and down-regulated genes in RNA-seq data? The method section only talks about differentially expressed genes in Ribo-seq data. Would not one expect that a defect in translational efficiency (and hence protein levels) lead to secondary effects on transcription?

Line 338-341, these conclusions from Ribo-seq data need to be corroborated on a handful of genes in dual-luciferase assays.

Lines 345 and 350, please, include the actual data in supplemental figures, even if the morphometric traits analyzed show no difference between WT and the three mutant genotypes. Stating that something does not have a phenotype and showing no data to corroborate that statement is not acceptable, in my opinion.

Supplemental Figure 20, why is the germination (and root length) in control media (no antibiotic) not shown? I worry that with only one allele of each mutant tested, it is unknown if the mild defect in germination seen in the presence of drugs is a consequence of the *rlmN* gene disruptions or unrelated mutations in these genetic backgrounds (or even a seed batch variation). In the absence of additional *rlmN* alleles on hand, complementation lines need to be examined side by side with the mutants.

Line 354, in situations like that I wonder whether WT is coming from an age-matched seed batch (ideally propagated side-by-side with the mutants).

Line 379, what is ASL? Spell out.

Line 457, provide AtG gene numbers for ACT2 and TUB4 reference genes.

Lines 572 and 582, the reference to puromycin-labeled samples being dried is misleading. I think the authors mean pat-dried, not dried (e.g., lyophilized).

Lines 584 and on, the description of Ribo-seq does not make it clear that RNA-seq is performed in parallel on the same lysates to enable quantification of translational efficiency. Please, amend.

Grammar edits (this is not a comprehensive list and professional editing is recommended):

Lines 25 and 75, "Multi-types of RNA" sounds awkward. Rephrase to "different types of RNA".

Line 31, change "location and transportation" to "localization and transport", and "regulate" to "can regulate".

Line 32, change "function" to "may function".

Line 48, it is unclear what "it" refers to. Change "it" to "this modification".

Line 52, change "chloroplasts" to "chloroplast".

Line 53, change "fraction" to "frequency".

Line 59, change "cytosol tRNA" to "cytosolic tRNA".

Line 59-60, rephrase "tandem m2A-tRNA dependent codon dependent manner".

Line 63, change "multi-species" to "many species".

Line 64, change "isolated" to "purified".

Line 88, add "that" before "two".

Line 112, change "(t)RNA" to "tRNA".

Line 125, change "23S RNA" to "23S rRNA".

Line 139, "mechanically" or "mechanistically"? Mechanically does not make sense.

Line 151, "with deltaproteobacteria RlmN"?

Line 157, "relate to that of alphaproteobacterial".

Line 158, change "widely spreading" to "wide distribution".

Line 159, change "origination" to "origin".

Line 161, change "separated" to "different".

Line 167, change "indicating" to "suggesting".

Line 227, change "parallely" to "in parallel".

Line 240, change "bacteria" to "bacterial".

Line 246, change "instantaneous translation velocity" to "new protein synthesis".

Line 262, 265 and 273, change "different genotypic protoplasts" to "protoplasts of different genotypes".

Line 269, change "are reduced" to "is reduced".

Line 299, rephrase this sentence.

Line 614, change "alternative expressed" to "differentially expressed".

Reviewer #4 (Remarks to the Author):

In this thoroughly revised manuscript, authors have addressed the requested revisions by adding a number of new experiments. The most notable are:

- Experiments to show that tandem codons decoding by m² modified tRNA can affect protein translation
- Gel migration analysis that suggests a more open tRNA induced by modification, which could accelerate translation elongation and prevent translation pauses on tandem codons.
- Development of MBN-assisted LC-MS/MS detection method (Malc) to determine the location and fraction of m² A modification in rRNA and tRNA at single-nucleotide resolution

Overall, these revisions significantly strengthen the manuscript and its quality and impact.

I would like to suggest the following revisions:

A. Page 12 of the combined pdf:

1. We hypothesized that m² A37 could play a role in tRNA conformation maintenance.

Change maintenance to regulation or selection.

2. These results indicated a more compactly folded architecture of m² A deficient tRNAs, suggesting that m² A37 helps maintain the proper conformation of tRNA

Change proper to more relaxed as that is all one can conclude from the gel migration

B. Page 23: The mRNA was qualified using a 2100 Bioanalyzer (Agilent). Should read quantified.

C. Page 38, Figure 4: Define eMet in figure legend.

D. Page 38, Figure 5 legend: RNA m² A modification helps maintain the proper conformation.

Better to rephrase to: "m² A modification helps regulate RNA conformation"

Regarding gel migration data: Small migratory differences are indeed to be expected, given that this is a single carbon modification in otherwise very large tRNA molecule. However, given small migration differences, authors should: 1. provide at least two independent replicates of this experiment as a supplementary file to test if the finding is reproducible; 2. submit full gels (not only region of interest). This is important as this figure support main conclusions of the manuscript regarding the role of m²A in tRNA. Authors should also be careful not to overstate what they can reliably conclude from a gel-based migration assay.

Dear Reviewers,

We would very much like to thank all reviewers for their valuable comments about our work. In the revision, we performed new experiments to strengthen our conclusion that **tRNA m²A37 modification helps regulate tRNA conformation and promotes translation efficiency through decoding tandem m²A-tRNA-dependent codons**. In the revised Manuscript and Supplementary Information, changes were all highlighted in the blue-colored text. Comments of reviewers were responded to below in “Point-by-point responses”.

Reviewer #1 (Remarks to the Author):

The revised version of the manuscript is significantly improved overall. However, in my first round of review I missed that there are only 2 replicates of all sequencing experiments performed for this study. This is not sufficient for a high level publication and at least 1 additional biological replicate should be performed and added to the analysis to publish a study based on genomics experiments in a Nature group journal. These additional data should be added to a revised manuscript before publication of this study.

Response #1: We thank this referee for the suggestion. In our opinion, the number of biological replicates is dependent on the repeatability and replicability of experiments. Ribosome footprinting sequencing, or Ribo-seq, is not an experiment with high variance (e.g., CLIP-seq). Our two replicates of sequencing are with high correlation (Supplementary Fig. 18). Besides, we searched publications of 2021 and 2022 conducting Ribo-seq in Nature Press Group journals with high impact factors and found out at least eight papers performed two (not three) biological replicates (about half of all NPG papers which conducted Ribo-seq these two years):

Xin Erica Shu et al. Dynamic eIF3a O-GlcNAcylation controls translation reinitiation during nutrient stress. *Nature Chemical Biology* **18**, 134-141 (2022). See GSE181040.

GSM5482341 Ribo-seq_Control_rep1
 GSM5482342 Ribo-seq_Control_rep2
 GSM5482343 Ribo-seq_Starvation_rep1
 GSM5482344 Ribo-seq_Starvation_rep2
 GSM5482345 Ribo-seq_OGT_KO_rep1
 GSM5482346 Ribo-seq_OGT_KO_rep2
 GSM5482347 Ribo-seq_OGT_KO_Starvation_rep1
 GSM5482348 Ribo-seq_OGT_KO_Starvation_rep2
 GSM5482349 Ribo-seq_ThiametG_rep1
 GSM5482350 Ribo-seq_ThiametG_rep2
 GSM5482351 Ribo-seq_ThiametG_Starvation_rep1
 GSM5482352 Ribo-seq_ThiametG_Starvation_rep2

Kevin C. Stein et al. Ageing exacerbates ribosome pausing to disrupt cotranslational proteostasis. *Nature* **601**, 637-642 (2022). See Extended Data Fig. 1e.

Palaniraja Thandapani et al. Valine tRNA levels and availability regulate complex I assembly in leukaemia. *Nature* **601**, 428-433 (2022). See the figure legend of Extended Data Fig. 8 and GSE167535.

GSM5106818 TALL2_8g_valine_total_RNA
 GSM5106819 TALL2_8g_valine_RPF
 GSM5106820 TALL2_0-8g_valine_total_RNA
 GSM5106821 TALL2_0-8g_valine_RPF
 GSM5106822 TALL2_0-4g_valine_total_RNA
 GSM5106823 TALL2_0-4g_valine_RPF
 GSM5106824 TALL2_0g_valine_total_RNA
 GSM5106825 TALL2_0g_valine_RPF
 GSM5106826 TALL3_8g_valine_total_RNA
 GSM5106827 TALL3_8g_valine_RPF
 GSM5106828 TALL3_0-8g_valine_total_RNA
 GSM5106829 TALL3_0-8g_valine_RPF
 GSM5106830 TALL3_0-4g_valine_total_RNA
 GSM5106831 TALL3_0-4g_valine_RPF
 GSM5106832 TALL3_0g_valine_total_RNA
 GSM5106833 TALL3_0g_valine_RPF

Nicola Guzzi et al. Pseudouridine-modified tRNA fragments repress aberrant protein synthesis and predict leukaemic progression in myelodysplastic syndrome. *Nature Cell Biology* **24**, 299-306 (2022).

See Supplementary Table 2 and GSE162050.

GSM4932271 H9 WT Total replicate 1
GSM4932272 H9 WT Total replicate 2
GSM4932273 H9 PUS7KO Total replicate 1
GSM4932274 H9 PUS7KO Total replicate 2
GSM4932275 H9 WT RPF replicate 1
GSM4932276 H9 WT RPF replicate 2
GSM4932277 H9 PUS7KO RPF replicate 1
GSM4932278 H9 PUS7KO RPF replicate 2

Maxim S. Svetlov et al. Context-specific action of macrolide antibiotics on the eukaryotic ribosome. *Nature Communications* **12**, 2803 (2021). See Supplementary Fig. 8 and GSE164275

GSM5006256 non-treated yeast, biol rep1
GSM5006257 non-treated yeast, biol rep2
GSM5006258 telithromycin-treated yeast, biol rep1
GSM5006259 telithromycin-treated yeast, biol rep2

Johanna Schott et al. Nascent Ribo-Seq measures ribosomal loading time and reveals kinetic impact on ribosome density. *Nature Methods* **18**, 1068-1074 (2021). See the figure legend of Fig. 3 and GSE155236.

GSM5233895 mESCs_IN_4sU60_rep1
GSM5233896 mESCs_IN_4sU90_rep1
GSM5233897 mESCs_IN_4sU120_rep1
GSM5233898 mESCs_RPF_4sU60_rep1
GSM5233899 mESCs_RPF_4sU90_rep1
GSM5233900 mESCs_RPF_4sU120_rep1
GSM5233901 mESCs_IN_4sU60_rep2
GSM5233902 mESCs_IN_4sU90_rep2
GSM5233903 mESCs_IN_4sU120_rep2
GSM5233904 mESCs_RPF_4sU60_rep2
GSM5233905 mESCs_RPF_4sU90_rep2
GSM5233906 mESCs_RPF_4sU120_rep2

Eun Yu Kim et al. Ribosome stalling and SGS3 phase separation prime the epigenetic silencing of transposons. *Nature Plants* **7**, 303-309 (2021). See Supplementary Table 2 and PRJNA598331.

Samples	Raw Reads	Clean Reads	Total mapped	Uniquely mapped
RNA-seq_Col-0_rep1 (cont. for ribo-seq)	46,047,126	45,139,810	41,008,120	40,258,122
RNA-seq_Col-0_rep2 (cont. for ribo-seq)	52,046,884	51,109,468	49,844,718	48,878,726
RNA-seq_ddd1_rep1 (cont. for ribo-seq)	55,317,172	52,539,734	50,841,654	49,729,545
RNA-seq_ddd1_rep2 (cont. for ribo-seq)	46,567,406	45,892,166	44,687,347	43,731,004
Ribo-seq_Col-0_rep1	52,578,246	44,527,711	37,801,610	24,303,506
Ribo-seq_Col-0_rep2	69,952,186	56,604,548	47,020,134	32,101,023
Ribo-seq_ddd1_rep1	54,417,340	44,004,965	40,747,657	30,115,491
Ribo-seq_ddd1_rep2	53,154,939	42,013,949	34,065,658	24,189,782
RNA-seq_ddd1_rep1 (cont. for RG-RNA-seq)	45,631,136	44,664,672	42,910,976	41,950,104
RNA-seq_ddd1_rep2 (cont. for RG-RNA-seq)	46,119,730	45,224,068	43,911,352	42,952,790
RG-RNA-seq_ddd1_rep1	48,818,762	47,898,388	45,802,796	44,772,760
RG-RNA-seq_ddd1_rep2	43,518,858	42,765,376	41,143,912	40,206,878
RNA-seq_ddd1_rdr6 (cont. for ribo-seq)	48,973,288	48,355,352	47,545,301	46,384,306
RNA-seq_ddd1 (cont. for ribo-seq)	61,952,662	61,003,038	60,094,792	58,439,072
Ribo-seq_ddd1_rdr6	61,093,198	45,421,998	41,647,461	21,400,901
Ribo-seq_ddd1	62,576,020	50,498,998	34,532,242	10,298,192

J. Blaze et al. Neuronal Nsun2 deficiency produces tRNA epitranscriptomic alterations and proteomic shifts impacting synaptic signaling and behavior. *Nature Communications* **12**, 4913 (2021). See GSE165202.

GSM5664680 KO1_Riboseq
GSM5664681 WT1_Riboseq
GSM5664682 KO2_Riboseq
GSM5664683 WT2_Riboseq

Reviewer #3 (Remarks to the Author):

This is a resubmission and was one of the original reviewers for this manuscript. Some of my previous comments have been addressed, but some concerns remain. As three years have passed since the first submission and many major changes have been made to the manuscript, I chose to evaluate this work as a brand-new submission. Based on the data presented, I have no doubt that RLNML1,2,3 in *Arabidopsis* catalyze the m²A modifications in chloroplast rRNA, chloroplast tRNA, and cytosolic tRNA, respectively. I was also impressed by the new Malc method to confirm the specific locations of m²A modifications in the RNA. However, the data on the mechanistic consequences of m²A modifications on the structure and function of target RNAs are not very robust or internally consistent (see below). Likewise, the biological outcomes of *rlnml* mutations at the phenotypic level are not well characterized (only one allele per gene is examined, and the morphological defects are super-mild, which is inconsistent with the strong molecular phenotypes presented). Finally, the question of how specific plastidic and cytoplasmic target RNAs (and specific adenosines within them) are chosen by RLNML1/2 and 3, respectively, is not addressed.

Response #2: We thank this referee for the thorough reviews. Most comments in this paragraph would be addressed below in the “specific comments” part.

For the concern of “the morphological defects are super-mild, which is inconsistent with the strong molecular phenotypes presented”, we would take Up47, a recent discovered reversible tRNA modification as an example (Please see Ohira et al. *Nature* 605, 372-379 (2022)). Defects of Up47 dramatically reduce the thermostability of the corresponding tRNA, and the structural function of Up47 has been verified by crystallography data. However, lack of Up47 from archaea *T. kodakarensis* does not lead to significant growth inhibition at 87 and 91°C. The archaea only exhibit growth inhibition at 87°C when both Up47 and G⁺ (another tRNA modification associated with Up47 spatially) are deficient. As we have explained in the response of the last revision, single modification loss from tRNA and rRNA usually do not lead to severe development and growth defect under normal conditions (Please see Vicente Ramírez et al. *MPMI* 31:12, 1323-1336 (2018) and Peng Chen et al. *BMC Plant Biol* 10, 201 (2010)).

For the question of “how specific plastidic and cytoplasmic target RNAs (and specific adenosines within them) are chosen by RLNML1/2 and 3”, we consider it as a mechanistic question of RlmN protein. However, the main content of our manuscript is the molecular functions and the biological roles of m²A modification. The enzyme selectivity could be investigated in further research via biochemistry and structural biology.

The data to suggest a more compact structure of plastidic and cytosolic tRNAs in the absence of m2A modifications at A37 in *rlnml2* and *rlnml3* mutants, respectively, are not very robust (mobility shift results are questionable, see below). The data on puromycin labeling of nascent peptides in *rlnml2* and *rlnml3* are inconsistent with the overall plastidic and cytoplasmic protein levels in the mutants (should not nascent peptide synthesis be proportional to total protein levels in these lines?). Some Ribo-seq data quality controls are missing (even though I specifically asked for them in the original review 3 years ago). The observation on the need for the m2A-modified-tRNA-dependent codons to be found in tandem in transcripts for the *rlnml3* mutant to have an effect on mRNA translational efficiency was not re-tested in dual luciferase assays (only the effect of tandem codons was analyzed). The lack of major phenotypes in the *rlnml* mutants is surprising given the prominent molecular defects of these lines and examining only one allele of each gene is not sufficient (in the absence of additional alleles, complementation lines should have been included in the phenotypic assays).

Response #3: Comments in this paragraph would be addressed below in the “specific comments” part.

My specific comments are listed below:

Lines 65 and 77, specify: digested with what?

Response #4: We have rephrased it as “enzymatically digested” (Lines 66 and 78). Detail can be found in the Methods section.

Figure 1b, c – define TIC and m/z in the legend (the same applies to Supplemental figure 1). Specify from what samples (tissue, growth conditions, age) total RNA and purified RNAs were extracted.

Response #5: In the revision, we have defined TIC and m/z in the figure legend. Additionally, we have mentioned plant samples in the Methods section that except where otherwise indicated, 14-day-old seedlings were used in all experiments.

Lines 72-73, I do not think this conclusion can be made from looking at RNA from one tissue sample. Nowhere do the authors specify what tissues were used for RNA extraction. Plant growth conditions and sample types can influence the relative ratios of different types of RNA and their nucleotide modification profiles.

Response #6: We agree with the reviewer that plant growth conditions and sample types can influence the relative ratios of different types of RNA and their nucleotide modification profiles. However,

tissue specificity and growth conditions would not affect the existence of one RNA modification. To be more precise, we have changed the description to “we did not identify m⁸A in examined *Arabidopsis* RNA samples”. (Line 74)

Lines 74-82 and Table 1, again, I could not find the information on the growth conditions and tissue types for any of the plant species. I do not think any of these values are reproducible or comparable given that the plant tissues were sourced from different, non-standardized places (colleagues, supermarkets and other merchants, etc.). The source of HeLa cells is not listed anywhere. The text refers to “human total RNA as well as separated RNA types”, but Table 1 shows total and poly(A) RNA only. Reconcile!

Response #7: In the revision, more details about plant material sources, plant growth conditions, tissue types, and cell line information were added in the Methods section (Lines 475-489, Lines 507-508). Results presented in Table 1 of the old version are now presented in Supplementary Fig. 3 of the revision.

Oryza sativa (japonica group) cv. Nipponbare was cultivated in Hoagland solution at 30°C under a 12 h light/12 h dark cycle. Shoots of 15-day-old Nipponbare hydroponic seedlings were used for RNA extraction (As in our previous publication: Yu et al. Nat. Biotech. 39:1581-1588).

Spinach was purchased from the local market and sterilized following the steps: the material was first dipped in 70% alcohol for 30 seconds, then in 0.5% SDS for 5 minutes. After that, the material was washed 5 times with sterilized distilled water and wiped dry. Leaf edge was used for DNA and RNA extraction. Two pairs of species-specific primers on mitochondrial gene CytB were used to verify spinach material (Supplementary Fig. 26).

Physcomitrella patens Gransden (gifts from Prof. Haodong Chen’s lab at Tsinghua University. Please see their previous publication: Li et al. Nat. Comm. 12:4470) was cultivated on BCD medium with 1% sucrose at 22°C under long-day conditions (16 h light/8 h dark, white fluorescent tubes, 100 μmol m⁻² s⁻¹), and whole plants were used for RNA extraction.

Chlamydomonas reinhardtii (FACHB-265) was obtained from Freshwater Algae Culture Collection at the Institute of Hydrobiology (FACHB. It is an authoritative source of algae in China.) and cultivated in SE medium (Bristol’s solution with A5 trace metal and soil extract) at 25°C under a 12 h light/12 h dark cycle. Algal cells were harvested at the exponential phase for RNA extraction.

HeLa cells (ATCC CCL-2) were obtained from ATCC and cultivated at 37°C in DMEM with 10%

fetal bovine serum and 1% penicillin/streptomycin in 5% CO₂ (As in our previous publication: Song et al. *Cell Chemical Biology* 27:283-291)

Additionally, we learned that *Azolla filiculoides* always have symbiotic blue bacteria in the natural condition which could bias our results. Thus, we deleted the contents about *Azolla filiculoides* in the revision.

Supplemental Figure 3, the table is very hard to read. Why does position 35 list the entire anticodon rather than a single nucleotide? Why is the position 37 (3' adjacent to the codon) that is key to this work not shown? What is iM and eM (it is not obvious that the initiator and elongator Met are meant)? Also, the reference to nuclear-encoded (table) versus cytoplasmic (narrative) tRNA is confusing. I think both terms are supposed to mean the same thing.

Response #8: This figure is now Supplementary Fig. 4 in the revision. The tables are in a common form illustrating integrated information on tRNAs. Here is an example from Yacoubi et al. *Annu. Rev. Genet.* 46, 69-95 (2012).

Anticodon 35

		A		G		U		C						
A	Phe			[E] [A,B]G		Tyr	Q and Q [#]		Cys	G**				
G		Gm		[B]xo ⁵ U [E]ncm ⁵ U			[B]U*			Stp				
U		[B]xmn ⁵ U [E]xcm ⁵ U [A]?U		C							Trp	Cm		
C		C												
A	Leu	[E]i [A,B]G		[E]i [A,B]G		His		Q				Arg	[E,B]i [A]G	
G		[B]xo ⁵ U [E]U		[B]xo ⁵ U [E]ncm ⁵ U			[B]U*	[B]xmn ⁵ s ² U [E]xcm ⁵ s ² U [A]?U		[B]mnm ⁵ U [E]mcm ⁵ U				
U		C		C				C		C				
C														
A	Ile	[E]i [A,B]G		[E]i [A,B]G		Asn		Q		Ser	G***			
G		[E]Ψ [B]k ² C*** [A]agm ² C***		[B]xo ⁵ U [E]ncm ⁵ U			[B]U*	[B]xmn ⁵ s ² U [E]xcm ⁵ s ² U [A]?U			[B]xmn ⁵ U [E]xcm ⁵ U [A]?U			
U		C		C				C			C			
C														
A	Val	[E]i [A,B]G		[E]i [A,B]G		Asp		Q		Gly	G			
G		[B]xo ⁵ U [E]ncm ⁵ U		[B]xo ⁵ U [E]ncm ⁵ U			[B]U*	[B]xmn ⁵ s ² U [E]xcm ⁵ s ² U [A]?U			[B]xmn ⁵ Um [E]xcm ⁵ Um [A]?U			
U		C		C				C			C			
C														

34 C or^e Met

And an example from Hermand, *Epigenomes* 4, 7 (2020):

C

First codon base (1)	Middle codon base (2)								Third (wobble) codon base (3)
	U		C		A		G		
	Anticodon Codon:aa +modification	Codon:aa +modification	Anticodon Codon:aa +modification	Codon:aa +modification	Anticodon Codon:aa +modification	Codon:aa +modification	Anticodon Codon:aa +modification	Codon:aa +modification	
U	UUU:Phe		UCU:Ser	IGA	UAU:Tyr		UGU:Cys		U
	UUC:Phe	GAA	UCC:Ser		UAC:Tyr	G*A	UGC:Cys	GCA	C
	UUA:Leu	ncm5UmAA	UCA:Ser	ncm5UGA	UAA STOP		UGA STOP		A
C	UUG:Leu	m5CAA	UCG:Ser	CGA	UAG STOP		UGG:Trp	CmCA	G
	CUU:Leu	AAG	CCU:Pro	IGG	CAU:His		CGU:Arg	ICG	U
	CUC:Leu		CCC:Pro		CAC:His	GUG	CGC:Arg		C
	CUA:Leu	UAG	CCA:Pro	ncm5UGG	72%CAA:Gln	mcm's ² UUG	CGA:Arg	UCG	A
A	CUG:Leu		CCG:Pro	CGG	28%CAG:Gln	CUG	CGG:Arg		G
	AUU:Ile	IAU	ACU:Thr	IGU	AAU:Asn		AGU:Ser		U
	AUC:Ile		ACC:Thr		AAC:Asn	GUU	AGC:Ser	GCU	C
	AUA:Ile	*A*	ACA:Thr	ncm5UGU	62%AAA:Lys	mcm's ² UUU	AGA:Arg	mcm5UCU	A
G	AUG:Met	CAU	ACG:Thr	CGU	38%AAG:Lys	CUU	AGG:Arg	CCU	G
	GUU:Val	AAC	GCU:Ala	IGC	GAU:Asp		GGU:Gly		U
	GUC:Val		GCC:Ala		GAC:Asp	GUC	GGC:Gly	GCC	C
	GUA:Val	ncm5UAC	GCA:Ala	ncm5UGC	68%GAA:Glu	mcm's ² UUC	GGA:Gly	mcm5UCC	A
		GCG:Ala	CGC	32%GAG:Glu	CUC	GGG:Gly	CCC	G	

Position 34, 35, and 36 are anticodons of tRNA and they decide what species the tRNA is. It is similar to the common codons table of protein translation, which organizes codons according to the base of positions 1, 2, and 3. Unlisted anticodon means tRNA with this anticodon is nonexistent.

		Position 35					
Position 36	A	G	C	U	Position 34	A	G
A	GAA (F)	AGA (S)	GCA (C)	GUA (Y)		A	G
	CAA (L)	CGA (S)	CCA (W)			C	C
	UAA (L)	UGA (S)				U	U
G	AAG (L)	AGG (P)	ACG (R)			A	G
	CAG (L)	CGG (P)	CCG (R)	GUG (H)		C	C
	UAG (L)	UGG (P)	UCG (R)	CUG (Q) UUG (Q)		U	U
C	AAC (V)	AGC (A)	GCC (G)	GUC (D)		A	G
	CAC (V)	CGC (A)	CCC (G)	CUC (E)		C	C
	UAC (V)	UGC (A)	UCC (G)	UUC (E)		U	U
U	AAU (I)	AGU (T)	GCU (S)	GUU (N)		A	G
	CAU (iM)	CGU (T)	CCU (R)	CUU (K)		C	C
	CAU (eM)						
	UAU (I)	UGU (T)	UCU (R)	UUU (K)		U	U

anticodon table

		Second base					
		U	C	A	G		
U	UUU } Phenylalanine F	UCU } Serine S	UAU } Tyrosine Y	UGU } Cysteine C			
	UUC } Serine S	UCC } Serine S	UAC } Tyrosine Y	UGC } Cysteine C			
	UUA } Leucine L	UCA } Serine S	UAA } Stop codon	UGA } Stop codon			
	UUG } Leucine L	UCG } Serine S	UAG } Stop codon	UGG } Tryptophan W			
C	CUU } Leucine L	CCU } Proline P	CAU } Histidine H	CGU } Arginine R			
	CUC } Leucine L	CCC } Proline P	CAC } Histidine H	CGC } Arginine R			
	CUA } Leucine L	CCA } Proline P	CAA } Glutamine Q	CGA } Arginine R			
	CUG } Leucine L	CCG } Proline P	CAG } Glutamine Q	CGG } Arginine R			
A	AUU } Isoleucine I	ACU } Threonine T	AAU } Asparagine N	AGU } Serine S			
	AUC } Isoleucine I	ACC } Threonine T	AAC } Asparagine N	AGC } Serine S			
	AUA } Methionine start codon M	ACA } Threonine T	AAA } Lysine K	AGA } Arginine R			
	AUG } Methionine start codon M	ACG } Threonine T	AAG } Lysine K	AGG } Arginine R			
G	GUU } Valine V	GCU } Alanine A	GAU } Aspartic acid D	GGU } Glycine G			
	GUC } Valine V	GCC } Alanine A	GAC } Aspartic acid D	GGC } Glycine G			
	GUA } Valine V	GCA } Alanine A	GAA } Glutamic acid E	GGA } Glycine G			
	GUG } Valine V	GCG } Alanine A	GAG } Glutamic acid E	GGG } Glycine G			

codon table

Position 37 (3'-adjacent to anticodon) of tRNA is always A or G in almost all species. Therefore, we illustrated the tRNAs have A at position 37 in red, while the tRNAs have G at position 37 in grey.

“iM” and “eM” stand for the initiator and elongator methionine tRNA, respectively. In the revision, they have been defined in the figure legend.

The legend for Supplementary Fig. 4a has been changed to “cytosolic” as the reviewer suggested.

Lines 86-87, state the total number of tRNAs with m2A modifications for which the biotinylated probes needed to be designed.

Response #9: It has been rephrased as “Given that m²A is located at position 37 of tRNA in *E. coli*, we first identified total 57 cytosolic or chloroplast tRNAs which have an adenosine 3'-adjacent to the anticodon” (Line 86).

Figure 2a, the Malc method is ingenious (kudos for that!), but it does require an a priori knowledge of the modified nucleotide.

Response #10: Yes, we have put the sentence in the Discussion section: “The Malc method can be applied to other modifications and RNA types, while it requires a priori knowledge of the modified nucleotide” (Lines 432-433).

Figure 2b,c, mark Fx-1, Fx and Fx+1 next to the probe names (the same is true of supplemental figure 4).

Response #11: We have redrawn Fig. 2b, c and Supplementary Fig. 5 (Supplementary Fig. 4 of the old version) in the revision as suggested by the reviewer.

Figure 2 legend, change the description of the procedure to present tense.

Response #12: We have corrected it.

Lines 119-123, while I understand the logic of why FX-1 (rather than Fx) was used to calculate the m²A modification frequency, I do not understand HOW that frequency was calculated and the relation between the m²A/A value plotted in figure 2C (~0.14) and the modification ratio stated in figure 2D (85%). Please, explain.

Response #13: Sorry for the confusion. Data presented in Fig. 2c is obtained from LC-MS/MS. It stands for the ratio of m²A versus total adenosine in each extracted fragment. More detailed values of all biological replicates for each tRNA species are presented in the Source Data File. Data presented in Fig. 2d stand for the modification frequency on position 37. The calculation procedure has been provided in the Methods section (Lines 584-589): Specifically, the exact modification fraction of m²A (R_m) was calculated through the equation $R_m = AR / (1 + R)$. Here A stands for the number of adenosine in Fx-1, and R stands for the m²A/A ratio measured by LC-MS/MS.

Line 130, what is the reason to think that the modification is constitutive if only one tissue was examined?

Response #13: We agree with the concern of the reviewer, and we have rephrased it to a “high stoichiometry” modification. (Line 132)

Line 138, no data are presented in support of RLMNLs being divergent in sequence from Cfr. Is there a Cfr like protein in Arabidopsis? If not, to what Cfr are RLMNLs being compared and what are the E-values?

Response #14: These three proteins have high sequence similarity with *E. coli* RlmN (BLAST E-value was $8e-65$, $1e-76$, and $3e-61$ for RLMNL1, RLMNL2, and RLMNL3, respectively) and relative lower similarity with *S. aureus* Cfr (BLAST E-value was $6e-55$, $3e-47$, and $2e-52$ for RLMNL1, RLMNL2, and RLMNL3, respectively). Previous consensus sequence analysis of RlmN and Cfr revealed several positions that are selectively conserved in these two families of proteins (Atkinson et al. Antimicrob. Agents. Chemother. 57, 4019-4026). RLMNL1, RLMNL2, and RLMNL3 contain conserved residues in the same as RlmN but distinguished from Cfr (Supplementary Fig. 6b), coinciding with the results that m⁸A is not detected in Arabidopsis RNA samples. (Lines 138-145)

Lines 166-168 and Supplemental figure 6b, it is customary to include marker proteins for comparison. While chlorophyll autofluorescence is a good marker substitute for chloroplasts, cytoplasmic protein localization may be confused with that in the endomembrane system, for example.

Response #15: We changed the statement to “RLMNL3 was excluded from chloroplast” which is supported by the localization results (Lines 173-174). Supplementary Fig. 6 is now Supplementary Fig. 7.

Supplemental figure 7, with three primer genotyping of insertional mutant lines, it is helpful to show that in a heterozygous sample, both WT and mutant bands are detectable, as it is not uncommon for one of the bands (usually, the smaller-sized one) to amplify preferentially. This is not essential to include as long as the RT-qPCR data are “clean”, meaning that there is no amplification product detected in the homozygous mutant with the primers spanning the insertion site. Herein, where does the residual RT-qPCR signal in the homozygous T-DNA lines come from? Do the qPCR primers used indeed span the insertion site? Does at least one of the RT-qPCR primers anneal to an exon-exon junction to avoid the amplification from contaminating genomic DNA? Please, mark the RT-qPCR primers used in the gene models shown in figure 7a (but distinguish them from the T-DNA genotyping primers, e.g., by color and figure legend clarification). This is something I requested in my original review, but the authors only marked the genotyping primers.

Response #16: Thanks for the suggestion. In the revision, we re-designed the qPCR primers to ensure

that the amplicons span the T-DNA insertion site (previous primers are all downstream of the insertion sites). The positions of new primers are illustrated in Supplementary Fig. 8a and the new results of qPCR are presented in Supplementary Fig. 8b.

Line 211, rRNA transcription and stability.

Response #17: We have rephrased this.

Supplemental figure 11, what does OH- mean? Please, specify in the figure legend.

Response #18: Sorry for the confusion. OH- means alkaline hydrolysis treatment. In the revision, we have redrawn the figure to change OH- into “alkaline treatment”. Deacylated tRNAs obtained from the total RNA treated with mild alkaline hydrolysis were run in parallel as the indicator of uncharged tRNA. Supplementary Fig. 11 is now Supplementary Fig. 12 in the revision.

Figure 5 (and supplemental figure 21), the mobility shift does not look very convincing for some tRNAs. For example, the tRNA^{Ser} GGA gel seems to be slightly slanted, and other shifted samples (e.g., tRNA^{Met} CAU) may show migration differences simply due to uneven RNA quantity (with overloaded samples traveling faster) or different concentration of impurities (e.g., salt levels). I am not sure what the authors mean by “more loosely-folded conformation of tRNA” – some of the base pairing not happening? If so, which bases (I assume that the modified base itself is in the anticodon loop and is not paired)? If the hypothesis is that m2A affects base pairing and, hence, RNA secondary structure, then single- and double-stranded nucleases should produce different tRNA cleavage patterns in WT versus the mutant on a polyacrylamide gel. Alternatively, as the authors themselves suggest in the discussion, NMR experiments and molecular dynamics simulations should have been performed.

Response #19: We have re-performed the gel migration assay. In both the old version and the revision, three replicates of WT or mutant samples run in parallel to avoid the misleading of occasional and unexpected band shifts. This strategy was also used in previous publications. Here is an example from Zhou et al. J Biol Chem 293:1425-1438.

In the revision, we have rephrased “more loosely folded conformation of tRNA” to “more relaxed conformation of tRNA” as Reviewer #4 suggested. As the output of gel migration assay, compactly folded RNA or DNA migrates faster than relaxedly folded one with the same sequence in native gels. The compactness of RNA folding would be decided by not only base-pairing (secondary structure) but also tertiary structure. It is just like when we electrophorese purified plasmids there are usually two bands, the faster-migrated band for the supercoiled plasmids and the slower-migrated one for the open circular plasmids. The technical details of RNA folding analyses by native PAGE can be found in Woodson & Koculi *Methods Enzymol* 469:189-208, cited in both the revision and the old version.

To provide more details on how m²A affects tRNA structure, we measure the melting curve of m²A-modified tRNA from Col-0 and m²A-deficient counterparts from mutants. The melting curves of tRNA usually show two transition points (Tanner & Cech, *RNA* 2:74-83 (1996)). The lower one is regarded as tertiary structure dissolving temperature (T_{m1}), and the higher one is regarded as secondary structure dissolving temperature (T_{m2}). The results showed that deficiency of m²A elevated T_{m1} in chloroplast tRNA^{Arg}_{ACG} and cytosolic tRNA^{Gln}_{UUG} significantly, while it increased T_{m2} in chloroplast tRNA^{eMet}_{CAU} and cytosolic tRNA^{Arg}_{ACG} (Fig. 5c). As negative controls, the non-m²A-modified chloroplast tRNA^{Gln}_{UUG} and cytosolic tRNA^{His}_{GUG} from Col-0 and *rlmnl2* or *rlmnl3* mutant have the same melting temperature (Fig. 5c). Since modifications in anticodon loop would have little effect on the L-shape tertiary structure of tRNA, a rational explanation is that m²A affects intra-loop base stacking (contribute to T_{m1}) or intra-loop base pairing (contribute to T_{m2}) in different tRNAs. It also coincided with the results of gel migration experiments as changes in intra-loop base interaction influence RNA folding. (Lines 245-261)

Figure 6a,b, I am confused by the experimental design and data shown in panels A and B. I can imagine the data being normalized per fresh weight of each sample, but not per total protein. Would not the total and nascent protein levels be proportional (meaning reduced nascent protein synthesis would ultimately lead to less protein accumulating)? If the gel loading is controlled for to keep the total amount of protein in each lane comparable between samples (and for a translation-compromised mutant, one would need to load more sample to get comparable levels of the protein of interest, RbcL or Actin/total protein), would not that also equalize nascent protein amount? To me, the data make no sense. How can one have little nascent protein made but have comparable RbcL or Actin (or total) levels overall? Do RbcL and Actin not have m²A-modified-tRNA-dependent codons in their sequences? Even if that were the case, this would not explain the fairly even Ponceau S staining relative to the very skewed puromycin-labeled proteins in panel B. What percentages of genes (plastidic and nuclear, respectively) harbor m²A-modified-tRNA-dependent codons? Also, I keep thinking that the drop in puromycin-labelled nascent proteins in *rlmnl2* and 3 mutants is very

profound to not have a major phenotypic consequence for plants.

Response #20: Sorry for the confusion. Puromycin-labeling is widely used to evaluate protein synthesis rates in human cells and plants. Specifically, puromycin is incorporated into nascent peptides and the new protein synthesis rate can be approximated via the immunoblotting of total protein lysates with an anti-puromycin antibody. To evaluate the puromycin intensities in different samples, the equal amounts of extracted total protein or chloroplast protein for each sample were resolved by SDS-PAGE, electrotransferred to PVDF membrane, and immunoblotted by the anti-puromycin antibody. Ponceau S staining was employed to ensure the equal loading of proteins. Using the same amount of total protein as loading control is a common procedure for puromycin-labelling assay. Please see References: Karunadasa et al. *New Phytologist* **227**: 50-64 (2020); Van Hoewyk, *Plant Methods* **12**:20 (2016); and Chunduri et al. *Nature Communications* **12**:5576 (2021) for detail.

RbcL and actin have m²A-modified-tRNA-dependent codons but not in tandem. Most labelled nascent peptides would be terminated midway and they do not appear as clear bands. Thus, we cannot identify nascent RbcL and actin on the anti-puromycin immunoblots exactly. As we illustrated in Fig. 7d, about 6% of all translated nuclear-encoded genes have no less than three tandem m²A-tRNA-dependent codons. Besides, about 17% of chloroplast-encoded genes (23% of non-ribosomal-protein genes) have three or four tandem m²A-tRNA-dependent codons. Apart from puromycin labelling, we also conducted dual-luciferase assay, polysome profiling, and ribosome footprinting sequencing to confirm the effect of m²A deficiency on protein translation. Our results suggest that tRNA m²A37 promotes protein translation through decoding tandem m²A-tRNA-dependent codons.

For the reviewer's concern "The drop in puromycin-labelled nascent proteins in *rlmnl2* and *3* mutants is very profound to not have a major phenotypic consequence for plants.", we would take Up47, a recent discovered reversible tRNA modification as an example (Please see Ohira et al. *Nature* **605**, 372-379 (2022)). Defects of Up47 dramatically reduce the thermostability of the corresponding tRNA, and the structural function of Up47 has been verified by crystallography data. However, lack of Up47 from archaea *T. kodakarensis* does not lead to significant growth inhibition at 87 and 91°C. The archaea only exhibit growth inhibition at 87°C when both Up47 and G+ (another tRNA modification associated with Up47 spatially) are deficient. As we have explained in the response of the last revision, single modification loss from tRNA and rRNA usually do not lead to severe development and growth defect under normal conditions (Please see Vicente Ramírez et al. *MPMI* **31**:12, 1323-1336 (2018) and Peng Chen et al. *BMC Plant Biol* **10**, 201 (2010)).

Figure 6d, do F-luc and R-luc have any of the *Rlmnl3*-dependent (i.e., m²A-modified-tRNA-

dependent) codons in their native sequences?

Response #21: They have m²A-modified-tRNA-dependent codons, but they do not have tandem codons depending on m²A-modified tRNAs.

Line 280 and Supplemental Figure 12, what does “two out of three” hypothesis refer to? Please, explain.

Response #22: We have provided this information. The “Two out of three” hypothesis means that a tRNA pairing with only the first two codon bases can be sufficient for translation. (Lines 303-304)

Line 282, why do the Rlmnl3-dependent codons need to be found in tandem for the rlmnl3 mutant to have an effect? Why was not this hypothesis tested in dual luciferase assays?

Response #23: The results of dual luciferase assays with six tandem m²A-tRNA-dependent codons in *rlmnl3* (Fig. 6c, d) and protein blotting of NDHH and PSBA with three tandem m²A-tRNA-dependent codons in *rlmnl2* (Fig. 6e) suggested a hypothesis that m²A³⁷ would promote translation efficiency through decoding tandem m²A-tRNA-dependent codons. To validate it, we performed ribosome footprinting sequencing to confirm the hypothesis in the *rlmnl3* mutant.

As suggested, we employed dual-luciferase assays in protoplasts isolated from 4-week-old leaves of wild-type Col-0, *rlmnl3*, *RLMNL3:RLMNL3/rlmnl3*, and *RLMNL3:RLMNL3m/rlmnl3* to further verify that tandem m²A-tRNA-dependent codons decoding was affected upon the absence of m²A. Three naturally existed tandem m²A-tRNA-dependent codons sequences in protein-coding genes (CAA-CGU-CGC-CGU from *AT2G18500*, CAG-CGU-CAA-CAA from *AT2G33640*, and CAA-CAG-CGU-CAA-CAA-CAG from *AT3G03460*) were inserted into dual-luciferase reporter as shown in Fig. 6c (termed as S1, S2, and S3, respectively). A sequence without m²A-tRNA-dependent codons (GUG-GUG-ACA-ACA, termed as S4) was chosen as a negative control. The results showed that translation levels of all three reporters with S1, S2, and S3 are significantly decreased in *rlmnl3* protoplasts compared with Col-0, which can be recovered in *RLMNL3:RLMNL3/rlmnl3*, but not in *RLMNL3:RLMNL3m/rlmnl3* (Supplementary Fig. 22). We did not observe significant differences in the translation of the empty control reporter and the negative control reporter among different genotypic protoplasts. These results demonstrated that translation of tandem m²A-tRNA-dependent codons is reduced by m²A³⁷ deficiency. See lines 349-362 for detail.

It should be noted that dual-luciferase assays are carried out in protoplast via transient transformation (see methods). But the transferring efficiency of plasmids into chloroplast is very low, and it is hard to

eliminate disturbance from cytosolic translation. Thus, the dual-luciferase assay could not be used to evaluate the translation efficiency changes in chloroplasts.

Supplemental figure 14a, are the protein expression differences statistically different? Also, the very mild reduction in the levels of two candidate proteins in *rlmnl2* is not consistent with the major (several fold) nascent protein reduction seen in figure 6a.

Response #24: Supplementary Fig. 14 is now Supplementary Fig. 16 in the revision. The differences are statistically significant. We have provided the information in Supplementary Fig. 16a.

It should be noted that accumulated total proteins are not only affected by translation rate, but also post-translation regulations like protein degradation. Thus, we think it is reasonable that the changed fold in translation rate (Fig. 6a) is not exactly the same as the changed fold of protein amount (Figure 6e and Supplementary Figure 16a).

Figure 7a, I do not think such a mild reduction in the polysomal fraction is real, especially that it is not accompanied by an increase in monosomal peaks. Different biological replicates of WT samples often show more variation than that. I suggest that the authors overlay three biological replicates each of WT and mutant to demonstrate that the mutant is, in fact, different from WT. State in the narrative how translational efficiency was calculated (currently the definition is buried in the methods).

Response #25: In the revision, we presented three biological replicates of polysome profiling results (Fig. 7a and Supplementary Fig. 17). All these three replicates showed that the 80S monosomal peak is slightly increased and the polysomal peaks are decreased.

The description of translation efficiency calculation is in lines 325-327: Translation efficiency (TE) of nuclear-encoded genes in Col-0 and *rlmnl3* were calculated via ribosome protected fragments divided by mRNA-seq FPKM (fragments per kilobase per million fragments) of corresponding genes

Lines 300-302, as written, it is unclear that the authors are talking about a ribosome footprinting experiment (aka Ribo-seq) when referring to sequencing data, since polysome profiling is mentioned in a prior sentence. Rephrase. Define FPKM.

Response #26: We have rephrased this part in the revision to “we performed three independent polysome profiling on Col-0 and *rlmnl3*. The results showed disruption of *RLMNL3* slightly increases 80S monosome peaks and reduces the polysome fractions (Fig. 7a and Supplementary Fig. 17). We also conducted ribosome footprinting sequencing together with mRNA sequencing on Col-0 and

rlmn13. Translation efficiency (TE) of nuclear-encoded genes in *Col-0* and *rlmn13* were calculated via ribosome protected fragments divided by mRNA-seq FPKM (fragments per kilobase per million fragments) of corresponding genes (Supplementary Fig. 18 and Supplementary Dataset 1)". (Lines 322-328)

Supplemental dataset 1 (and corresponding method description), in Ribo-seq quality metrics, the low number of reads in the UTRs looks good, but the 3nt periodicity of the dataset is not shown, so I am unable to reliably judge the validity of Figure 7 data and, accordingly, of the conclusions made. In the dataset table, I found the reference to "input" confusing. Please, rename "input" in the table as "RNA-seq" and change "Ribo" to "Ribo-seq". These terms would likely be more familiar to the reader and should probably be used in the narrative as well.

Response #27: In the revision, we have changed all the descriptions related to "input" and "ribo" into "mRNA-seq" and "Ribo-seq".

In the mapped reads of Ribo-seq, we observed the 12-nt offsets to the translation start codon and mild but obvious 3-nt periodicity.

The mild 3-nt periodicity may be due to the nuclease we used for polysome digestion. In published Ribo-seq methods, RNase I or micrococcal nuclease (MNase) was employed to generate ribosome-protected fragments (RPF). For P-site/A-site occupancy analysis, RNase I digestion is required since it has no 5'-end bias. MNase has a preference for cutting before A/U at the 5'-side so that it cannot be used when P-site/A-site occupancy is calculated. Here is the distribution of bases on the 5' read-ends

of our Ribo-seq.

Col-0 rep.1		Col-0 rep.2		rlmnl3 rep.1		rlmnl3 rep.2	
A	0.466829	A	0.465948	A	0.477429	A	0.468081
U	0.339068	U	0.344962	U	0.341311	U	0.342642
C	0.071807	C	0.070572	C	0.066285	C	0.068988
G	0.122296	G	0.118517	G	0.114975	G	0.120289

Due to the 5'-end bias, the Ribo-seq reads generated by MNase digestion may have a milder 3-nt periodicity, which is also observed in previous studies (See VanInsberghe et al. *Nature* **597**:561-565). However, the window for RNase I digestion is usually narrow. MNase digestion is much easier to handle than RNase I, so we eventually employed MNase to perform ribosome footprinting sequencing (see Methods). It should be noted that MNase is widely used in Ribo-seq (See McGlincy and Ingolia, *Methods* **126**:112-129; Kronja et al. *Cell Rep.* **7**:1495-1508; Dunn et al. *Elife* **2**:e01179-32; Wang et al. *Cell* **161**:1388-1399), and the conclusions we made are not dependent on P-site/A-site occupancy.

Supplemental figure 16, what is the cutoff for up- and down-regulated genes in RNA-seq data? The method section only talks about differentially expressed genes in Ribo-seq data. Would not one expect that a defect in translational efficiency (and hence protein levels) lead to secondary effects on transcription?

Response #28: Supplementary Fig. 16 is now Supplementary Fig. 19 in the revision. We have rephrased the “Input” of ribosome footprinting sequencing to “mRNA-seq”. The cutoff for up- and down-regulated genes is mentioned in lines 331-332 (cutoff criteria: FPKM fold change > 2; adjusted *p*-value < 0.05; *SNR* > 1), and the details are presented in the Methods section (Lines 709-714). Supplementary Fig. 19 shows the scatterplot of RNA-seq FPKM in wild-type and mutant plants. Up- and down-regulated genes (only a very small portion of total expressed genes) are illustrated in different colours, indicating that m²A has little effect on gene expression.

Line 338-341, these conclusions from Ribo-seq data need to be corroborated on a handful of genes in dual-luciferase assays.

Response #29: As suggested, we employed dual-luciferase assays in protoplasts isolated from 4-week-old leaves of wild-type Col-0, *rlmnl3*, *RLMNL3:RLMNL3/rlmnl3*, and *RLMNL3:RLMNL3m/rlmnl3* to further verify that tandem m²A-tRNA-dependent codons decoding was affected upon the absence of m²A. Three naturally existed tandem m²A-tRNA-dependent codons sequences in protein-coding genes (CAA-CGU-CGC-CGU from *AT2G18500*, CAG-CGU-CAA-CAA from *AT2G33640*, and CAA-CAG-CGU-CAA-CAA-CAG from *AT3G03460*) were inserted into dual-

luciferase reporter as shown in Fig. 6c (termed as S1, S2, and S3, respectively). A sequence without m²A-tRNA-dependent codons (GUG-GUG-ACA-ACA, termed as S4) was chosen as a negative control. The results showed that translation levels of all three reporters with S1, S2, and S3 are significantly decreased in *rlmnl3* protoplasts compared with Col-0, which can be recovered in *RLMNL3:RLMNL3/rlmnl3*, but not in *RLMNL3:RLMNL3m/rlmnl3* (Supplementary Fig. 22). We did not observe significant differences in the translation of the empty control reporter and the negative control reporter among different genotypic protoplasts. These results demonstrated that translation of tandem m²A-tRNA-dependent codons is reduced by m²A37 deficiency. See lines 349-362 for detail.

Lines 345 and 350, please, include the actual data in supplemental figures, even if the morphometric traits analyzed show no difference between WT and the three mutant genotypes. Stating that something does not have a phenotype and showing no data to corroborate that statement is not acceptable, in my opinion.

Response #30: In the revision, we have rephrased this paragraph. We have deleted “Disruption of *RLMNL1*, *RLMNL2*, or *RLMNL3* did not result in distinct phenotypes during growth under normal conditions.”, and directly described the hypersensitive phenotypes to aminoglycoside antibiotics. The germination ratio and root length of the genotypic lines including wild-type, mutant, and complement lines grown under the normal condition were provided (Lines 385-392). Details of phenotypic analysis were presented in the Methods section (Lines 492-509).

Supplemental Figure 20, why is the germination (and root length) in control media (no antibiotic) not shown? I worry that with only one allele of each mutant tested, it is unknown if the mild defect in germination seen in the presence of drugs is a consequence of the *rlmnl* gene disruptions or unrelated mutations in these genetic backgrounds (or even a seed batch variation). In the absence of additional *rlmnl* alleles on hand, complementation lines need to be examined side by side with the mutants.

Response #31: We re-performed the phenotypic analysis in the revision. Disruption of *RLMNL1*, *RLMNL2*, or *RLMNL3* lead to elevated sensitivity to paromomycin and G418 which are both aminoglycoside antibiotics and disturb protein translation. All three mutant lines showed delayed germination (Supplementary Fig. 24a) and suppressed early-stage vegetative growth (Supplementary Fig. 24b). The phenotypes can be recovered in *RLMNL1:RLMNL1/rlmnl1*, *RLMNL2:RLMNL2/rlmnl2*, and *RLMNL3:RLMNL3/rlmnl3* lines, but not in *RLMNL1:RLMNL1m/rlmnl1*, *RLMNL2:RLMNL2m/rlmnl2*, and *RLMNL3:RLMNL3m/rlmnl3* lines which only express enzymatically inactive mutants. Meanwhile, plants of these genotypes did not show significant differences in germination and root length without antibiotics. See lines 385-400 for detail.

Line 354, in situations like that I wonder whether WT is coming from an age-matched seed batch (ideally propagated side-by-side with the mutants).

Response #32: Yes, seeds of all genotypes are harvested from plants grown under the same condition at the same time. We mentioned it in lines 492-493

Line 379, what is ASL? Spell out.

Response #33: ASL is the abbreviation of anticodon stem-loop used in the literature 41 cited in the revised manuscript. In the revision, we deleted it and directly rephrase it to “synthesized *E. coli* tRNA^{Arg} anticodon stem-loops”. (Line 410)

Line 457, provide AtG gene numbers for ACT2 and TUB4 reference genes.

Response #34: We have provided this information. *ACTIN2* (AT3G18780) and *TUBULIN4* (AT5G44340) were used as reference genes for measuring the expression of *RLMNLI-3* (Line 539-540).

Lines 572 and 582, the reference to puromycin-labeled samples being dried is misleading. I think the authors mean pat-dried, not dried (e.g., lyophilized).

Response #35: Sorry for the misleading. We have described it in the method clearly. It should be “wiped dried with bibulous paper”. (Line 674)

Lines 584 and on, the description of Ribo-seq does not make it clear that RNA-seq is performed in parallel on the same lysates to enable quantification of translational efficiency. Please, amend.

Response #36: We have described the Ribo-seq method more clearly. For the ribosome footprinting assay, 20% of the lysate was saved as the input portion (saved for mRNA-seq to enable quantification of translational efficiency). The input portion was mixed with 5 ml TRIzol to further RNA extraction, mRNA purification, and fragmentation (~100 nt) for mRNA-seq. See line 687 and lines 690-692.

Grammar edits (this is not a comprehensive list and professional editing is recommended):

Lines 25 and 75, “Multi-types of RNA” sounds awkward. Rephrase to “different types of RNA”.

Line 31, change “location and transportation” to “localization and transport”, and “regulate” to “can regulate”.

Line 32, change “function” to “may function”.

Line 48, it is unclear what “it” refers to. Change “it” to “this modification”.

Line 52, change “chloroplasts” to “chloroplast”.

Line 53, change “fraction” to “frequency”.

Line 59, change “cytosol tRNA” to “cytosolic tRNA”.

Line 59-60, rephrase “tandem m2A-tRNA dependent codon dependent manner”.

Line 63, change “multi-species” to “many species”.

Line 64, change “isolated” to “purified”.

Line 88, add “that” before “two”.

Line 112, change “(t)RNA” to “tRNA”.

Line 125, change “23S RNA” to “23S rRNA”.

Line 139, “mechanically” or “mechanistically”? Mechanically does not make sense.

Line 151, “with deltaproteobacteria RlmN”?

Line 157, “relate to that of alphaproteobacterial”.

Line 158, change “widely spreading” to “wide distribution”.

Line 159, change “origination” to “origin”.

Line 161, change “separated” to “different”.

Line 167, change “indicating” to “suggesting”.

Line 227, change “parallelly” to “in parallel”.

Line 240, change “bacteria” to “bacterial”.

Line 246, change “instantaneous translation velocity” to “new protein synthesis”.

Line 262, 265 and 273, change “different genotypic protoplasts” to “protoplasts of different genotypes”.

Line 269, change “are reduced” to “is reduced”.

Line 299, rephrase this sentence.

Line 614, change “alternative expressed” to “differentially expressed”.

Response #37: All of these have been edited.

Reviewer #4 (Remarks to the Author):

In this thoroughly revised manuscript, authors have addressed the requested revisions by adding a number of new experiments. The most notable are:

- Experiments to show that tandem codons decoding by m² modified tRNA can affect protein translation
- Gel migration analysis that suggests a more open tRNA induced by modification, which could accelerate translation elongation and prevent translation pauses on tandem codons.
- Development of MBN-assisted LC-MS/MS detection method (Malc) to determine the location and fraction of m² A modification in rRNA and tRNA at single-nucleotide resolution

Overall, these revisions significantly strengthen the manuscript and its quality and impact.

I would like to suggest the following revisions:

A. Page 12 of the combined pdf:

1. We hypothesized that m² A³⁷ could play a role in tRNA conformation maintenance.

Change maintenance to regulation or selection.

Response #38: We thank this referee for the suggestions. We have changed it to “We hypothesized that m²A³⁷ could play a role in tRNA conformation regulation”. See line 234.

2. These results indicated a more compactly folded architecture of m² A deficient tRNAs, suggesting that m² A³⁷ helps maintain the proper conformation of tRNA

Change proper to more relaxed as that is all one can conclude from the gel migration

Response #39: Thanks for the suggestion. We have changed all “proper conformation” into “more relaxed conformation” in the revised manuscript.

B. Page 23: The mRNA was qualified using a 2100 Bioanalyzer (Agilent). Should read quantified.

Response #40: Here “qualify” should be used. We qualified the isolated mRNA using a 2100 Bioanalyzer (Agilent) to check whether the mRNA sample is degraded or contaminated with rRNA and tRNA. We mentioned it in the revision (Lines 546-547).

C. Page 38, Figure 4: Define eMet in figure legend.

Response #41: We have provided this information: “eMet stands for the elongator tRNA of methionine” in the revised Figure 4 legend.

D. Page 38, Figure 5 legend: RNA m² A modification helps maintain the proper conformation. Better to rephrase to: “m² A modification helps regulate RNA conformation”

Response #42: Thanks for the suggestion. We have rephrased the Figure 5 legend to: “m²A modification helps regulate tRNA conformation”.

Regarding gel migration data: Small migratory differences are indeed to be expected, given that this is a single carbon modification in otherwise very large tRNA molecule. However, given small migration differences, authors should: 1. provide at least two independent replicates of this experiment as a supplementary file to test if the finding is reproducible; 2. submit full gels (not only region of interest). This is important as this figure support main conclusions of the manuscript regarding the role of m²A in tRNA. Authors should also be careful not to overstate what they can reliably conclude from a gel-based migration assay.

Response #43: We have re-performed the gel migration assay. In both the old version and the revision, three replicates of WT or mutant samples run in parallel to avoid the misleading of occasional and unexpected band shifts. Please see Fig. 5a and 5b. The origin membranes of RNA-blot are provided in the Source Data file.

To provide more details on how m²A affects tRNA structure, we measure the melting curve of m²A-modified tRNA from Col-0 and m²A-deficient counterparts from mutants. The melting curves of tRNA usually show two transition points. The lower (T_{m1}) and higher (T_{m2}) transition points are generally regarded as tertiary and secondary structure dissolving temperatures, respectively. The results showed that deficiency of m²A elevated T_{m1} in chloroplast tRNA^{Arg}_{ACG} and cytosolic tRNA^{Gln}_{UUG} significantly, while T_{m2} in chloroplast elongator tRNA^{Met}_{CAU} and cytosolic tRNA^{Arg}_{ACG} are increased upon m²A defects (Fig. 5c). As negative controls, the non-m²A-modified chloroplast tRNA^{Gln}_{UUG} and cytosolic tRNA^{His}_{GUG} from Col-0 and *rlmnl2* or *rlmnl3* mutant have the same melting temperature (Fig. 5c). Since modifications in anticodon loop would have little effect on the L-shape tertiary structure of tRNA, a rational explanation is that m²A affects intra-loop base stacking (may contribute to T_{m1}) or intra-loop base pairing (may contribute to T_{m2}) in different tRNAs. It also coincided with the results of gel migration as changes in intra-loop base interaction influence RNA folding.

Reviewer #1 (Remarks to the Author):

My comments have been addressed. Still would prefer 3 reps, but understand the constraints (both time and money!).

Reviewer #5 (Remarks to the Author):

In the current manuscript, C2-methyladenosine is a ubiquitous RNA modification in plants that facilitates protein translation, the authors unequivocally show that AtRLNML1 is responsible for the m2A modifications in chloroplast rRNA, AtRLNML2 in the chloroplast tRNA, and AtRLNML3 in the cytosolic tRNA. Overall the work presented is of high interest and most parts of the manuscript show solid results, but the contribution of the characterized modifications to any biological aspects of plant development or the consequences caused by the absence of the modifications is still to be identified.

Despite the absence of a more detailed description of the origin and propagation of the different organisms used in the first part of the study, the results seem to be solid. However, once the authors try to identify the biological roles of rRNA m2A and tRNA m2A, some of the results required further interpretation.

It is difficult to understand how such large reductions in translation (shown in figure 6a and b) do not cause obvious phenotypes in the mutants.

What were the criteria to select NDHH and PSBA genes among the 14 genes with four or three m2A-tRNA-dependent codons in tandem? What about RPOC2? How is RPOC2 translation? Is it affected by m2A deficiency?

In the result section, "tRNA m2A37 promotes translation efficiency through decoding tandem m2A-tRNA-dependent codons in the cytosol", again it is hard to understand the small variation in the polysomal fraction, what is the explanation for the small increase in the monosomal peak? It is difficult to make a claim based on these small changes.

An excellent quality parameter for Ribo-seq data is the periodicity of the reads following the ribosome displacement codon by codon. Independently of the enzyme employed to generate ribosome-protected fragments, the periodicity in the reads should be maintained in high-quality Ribo-seq data.

For the mutants phenotypical characterization, are more alleles available in the different Arabidopsis mutant collections? Using only one allele is always problematic. In addition to that, the mutant used for the AtRLNML1 gene is intronic.

Other minor comments:

Line 243: Note that compactly folded RNA migrates faster than loosely folded.

The term relaxed used in other parts of the text is more appropriate.

Line 333: has little effect on gene expression

Perhaps, gene transcription is more suited.

Reviewer #1 (Remarks to the Author):

My comments have been addressed. Still would prefer 3 reps, but understand the constraints (both time and money!).

Response: We express our gratitude to the referee for diligently reviewing our manuscript and acknowledge the constraints involved. The bioinformatics analysis conducted indicates that our two replicated Ribo-seq and RNA-seq datasets exhibit high reliability. The presence of the 3-nt periodicity in our Ribo-seq data indicates the high-quality of our Ribo-seq data (Please see the response in Reviewer #5). The assertion that tRNA m²A modification enhances translation efficiency in both the chloroplast and cytosol by facilitating the decoding of tandem m²A-tRNA-dependent codons is not solely reliant on the sequencing data. We have complemented our findings with numerous additional experiments and single gene validations, reinforcing the robustness of our conclusions.

Given these supplementary validations, we believe that, at this stage, obtaining three replicates of sequencing may be excessive. The process of performing ribosome-profiling experiments with one more biological replicate would not only be resource-intensive but also potentially introduce variability due to growth conditions affecting plants harvested from different batches. Such variability could compromise the reliability of the sequencing data.

It is worth noting that the use of two replicates of Ribo-seq is a common practice in many relevant literature sources. We thank the referee appreciates the comprehensive nature of our experimental approach and the additional validations conducted, which collectively support the robustness of our conclusions.

Reviewer #5 (Remarks to the Author):

In the current manuscript, C2-methyladenosine is a ubiquitous RNA modification in plants that facilitates protein translation, the authors unequivocally show that AtRLNML1 is responsible for the m²A modifications in chloroplast rRNA, AtRLNML2 in the chloroplast tRNA, and AtRLNML3 in the cytosolic tRNA. Overall the work presented is of high interest and most parts of the manuscript show solid results, but the contribution of the characterized modifications to any biological aspects of plant development or the consequences caused by the absence of the modifications is still to be identified.

Response: We express our gratitude to the referee for providing very positive comments and conducting thorough reviews. Please find our point-by-point response below.

Despite the absence of a more detailed description of the origin and propagation of the different organisms used in the first part of the study, the results seem to be solid. However, once the authors try to identify the biological roles of rRNA m²A and

tRNA m²A, some of the results required further interpretation.

It is difficult to understand how such large reductions in translation (shown in figure 6a and b) do not cause obvious phenotypes in the mutants.

Response: We re-performed the phenotypic analysis in the revision. Disruption of *RLMNL1*, *RLMNL2*, or *RLMNL3* lead to elevated sensitivity to paromomycin and G418 which are both aminoglycoside antibiotics and disturb protein translation. All three mutant lines showed delayed germination (Supplementary Fig. 24a and Source data) and suppressed early-stage vegetative growth (Supplementary Fig. 24b and Source data). The phenotypes can be recovered in *RLMNL1:RLMNL1/rlmnl1*, *RLMNL2:RLMNL2/rlmnl2*, and *RLMNL3:RLMNL3/rlmnl3* lines, but not in *RLMNL1:RLMNL1m/rlmnl1*, *RLMNL2:RLMNL2m/rlmnl2*, and *RLMNL3:RLMNL3m/rlmnl3* lines which only express enzymatically inactive mutants. Meanwhile, plants of these genotypes did not show significant differences in germination and root length without antibiotics.

As to why the loss of m²A in tRNA and rRNA did not affect plant growth and development, we added a paragraph in Discussion (Please see Line 483-501). We checked the published results for other tRNA and rRNA modifications. Three methylation modifications (m¹G, m²G, and m⁷G) are well-conserved tRNA modifications, but only the loss of m²G exhibited an early-flowering phenotype⁴⁹. One interesting observation is that m²G modification is present at multiple positions of tRNA. Depletion of 2'-O-methylation modifications from position 32 and 34 of tRNA suppresses resistance to *Pseudomonas syringae* DC3000 in Arabidopsis⁵⁰. These results indicate that RNA modification at single site of tRNA might not be necessary for plant growth and development. The rRNA modification m^{6,6}₂A affects the translation of a subset of proteins and is involved in antibiotic resistance^{51,52}. However, the removal of m^{6,6}₂A from 18S rRNA of Arabidopsis mitochondria by depletion of its writer Dim1B was not reported to affect plant growth and development⁵³. Thus, we speculate that the loss of a single modification on tRNA and rRNA usually does not lead to severe defects in plant growth and development.

What were the criteria to select NDHH and PSBA genes among the 14 genes with four or three m²A-tRNA-dependent codons in tandem? What about RPOC2? How is RPOC2 translation? Is it affected by m²A deficiency?

Response: We randomly selected *NDHH* and *PSBA* genes for examination. We appreciate the suggestion to include RPOC2 in our analysis. In the revised manuscript, we extended our investigation to encompass both protein levels and gene expression of *RPOC2* gene. The results revealed that the protein level of RPOC2 was not significantly affected in *rlmnl2* mutant plants. However, the transcript level of *RPOC2* was notably elevated in *rlmnl2* compared to Col-0 (Supplementary Fig. 16b and 16c). Consequently, the apparent translation efficiency of RPOC2 was reduced in *rlmnl2* (Supplementary Fig. 16e), mirroring the observations made for NDHH and PSBA. For a more detailed presentation, please refer to the new Supplementary Fig. 16 and lines 319-329 in the revised manuscript.

In the result section, "tRNA m2A37 promotes translation efficiency through decoding tandem m2A-tRNA-dependent codons in the cytosol", again it is hard to understand the small variation in the polysomal fraction, what is the explanation for the small increase in the monosomal peak? It is difficult to make a claim based on these small changes.

Response: We attribute the small variation in the polysomal fraction to two factors: 1) the modification with m²A is limited to only two cytosolic tRNAs, tRNA^{Arg}_{ACG} and tRNA^{Gln}_{UUG}, and 2) m²A does not influence tRNA stability but rather selectively impacts the translation efficiency of a subset of proteins containing tandem m²A-dependent tRNA codons. We hypothesize that the observed decrease in the polysomal fraction may be accompanied by an increase in the monosomal peak.

Considering the limited number of cytosolic tRNAs modified with m²A, particularly tRNA^{Arg}_{ACG} and tRNA^{Gln}_{UUG}, we posit that this modest decrease in the polysomal fraction represents a substantial effect compared to the results observed for tRNA m¹A58 modification. The m¹A58 modification, which enhances the translation efficiency of certain proteins like MYC, is present in the majority of tRNAs. Despite the depletion of m¹A58 modification in the Trmt61a-KO sample, there was no significant decrease observed in the polysome peak, as shown in the figure below and referenced. In contrast, the m⁷G modification, present in 22 tRNAs, regulates both tRNA abundance and translation efficiency. Depletion of m⁷G, as demonstrated in the KO sample of the m⁷G writer, leads to a pronounced decrease in the polysome peak, as illustrated in the figure below and referenced. Thus, we ascribe the slight variation in the polysomal fraction to two key factors: 1) the modification with m²A is limited to only two cytosolic tRNAs, tRNA^{Arg}_{ACG} and tRNA^{Gln}_{UUG}, and 2) m²A does not influence tRNA stability but rather selectively impacts the translation efficiency of a subset of proteins containing tandem m²A-dependent tRNA codons.

Figure was from the Ref. *Nat Immunol.* 2022, 23(10):1433-1444.

Figure was from the Ref. *Mol Cell* 2018, 71(2):244-255.

An excellent quality parameter for Ribo-seq data is the periodicity of the reads following the ribosome displacement codon by codon. Independently of the enzyme employed to generate ribosome-protected fragments, the periodicity in the reads should be maintained in high-quality Ribo-seq data.

Response: Thanks for your inquiry. We have thoroughly analyzed our Ribo-seq data and have indeed identified 12-nt offsets to the translation start codon, which indicates a clear 3-nt periodicity in our Ribo-seq data (please refer to Response Figure 1 below). Either RNase I or micrococcal nuclease (MNase) was routinely employed to generate ribosome-protected fragments (RPF) in Ribo-seq procedure. Here we used MNase digestion. It's worth noting that MNase digestion in Ribo-seq commonly generates a 3-nt periodicity, a phenomenon consistently observed in previous studies (See VanInsberghe et al. *Nature* 597:561-565). Therefore, the presence of the 3-nt periodicity in our Ribo-seq data indicates the high-quality of our Ribo-seq data.

Response Figure 1. Analysis of the distance of 5'-end to the translation start codon in our Ribo-seq data.

For the mutants phenotypical characterization, are more alleles available in the different Arabidopsis mutant collections? Using only one allele is always problematic. In addition to that, the mutant used for the AtRLNML1 gene is intronic.

Response: Thanks for your inquiry. To illustrate the function of a gene, researchers typically employ either at least two alleles or a single allele in conjunction with a complementation assay. In our study, we aimed to illustrate the function of methyltransferases RLMNL1, RLMNL2, and RLMNL3. We employed a combination of one allele and complementation assays to demonstrate these three proteins are m²A methyltransferases and their functions are dependent on their activity of m²A methyltransferase. Additionally, the T-DNA insertion site in *rlmn11* is located within an intron. Despite this, our qPCR analysis (refer to Supplementary

Fig. 8b) confirmed the absence of full-length transcripts of RLMNL1 in *rlmnl1*. This observation might suggest that this intron region might play a regulatory role in the transcription of *RLMNL1*.

Specifically, we performed genetic complementation experiments in *rlmnl1*, *rlmnl2*, and *rlmnl3* mutants, generating two complemented lines for each mutant. The first set of lines, *RLMNL1:RLMNL1m/rlmnl1*, *RLMNL2:RLMNL2m/rlmnl2*, and *RLMNL3:RLMNL3m/rlmnl3*, expressed the catalytically inactive mutant proteins (RLMNL1 C398A, RLMNL2 C430A, and RLMNL3 C345A were termed as RLMNL1m, RLMNL2m, and RLMNL3m, respectively). The second set of lines, *RLMNL1:RLMNL1/rlmnl1*, *RLMNL2:RLMNL2/rlmnl2*, and *RLMNL3:RLMNL3/rlmnl3*, expressed wild-type methyltransferases (Supplementary Fig. 8b). We demonstrated that the loss of m²A in chloroplast rRNA in *rlmnl1* mutant, the loss of m²A in chloroplast tRNA in *rlmnl2* mutant, and the loss of m²A in cytosolic tRNA in *rlmnl3* mutant can be respectively restored in *RLMNL1:RLMNL1/rlmnl1*, *RLMNL2:RLMNL2/rlmnl2*, and *RLMNL3:RLMNL3/rlmnl3*. However, this restoration did not occur in lines expressing the catalytically inactive mutant proteins (Fig. 4b-d). This establishes RLMNL1, RLMNL2, and RLMNL3 as m²A methyltransferases responsible for modifying chloroplast rRNA, chloroplast tRNA, and cytosolic tRNA, respectively. Additionally, we conducted a dual-luciferase assay using protoplasts derived from 4-week-old leaves of wild-type Col-0, *rlmnl3*, *RLMNL3:RLMNL3/rlmnl3*, and *RLMNL3:RLMNL3m/rlmnl3* to demonstrate the impact of RLMNL3 on translation is dependent on its m²A methylation activity. Furthermore, we confirmed the hypersensitivity to aminoglycoside antibiotics in *rlmnl1*, *rlmnl2*, and *rlmnl3* can be respectively recovered by expressing wild-type RLMNL1, RLMNL2, and RLMNL3, but not by expression of the catalytically inactive mutant proteins (Supplementary Fig. 24).

Other minor comments:

Line 243: Note that compactly folded RNA migrates faster than loosely folded.

The term relaxed used in other parts of the text is more appropriate.

Line 333: has little effect on gene expression

Perhaps, gene transcription is more suited.

Response: We have carefully revised the descriptions in the manuscript. In the revised version of the manuscript, we have meticulously edited the text to ensure accuracy and clarity.

Reviewer #5 (Remarks to the Author):

The authors have addressed all my suggestions and concerns.